# Adam Reduces a Unique Form of Sharpness: Theoretical Insights Near the Minimizer Manifold

**Xinghan Li**[*]   **Haodong Wen**[*†]   **Kaifeng Lyu**[‡]
Institute for Interdisciplinary Information Sciences
Tsinghua University
{xh-li22,whd25}@mails.tsinghua.edu.cn   klyu@mail.tsinghua.edu.cn

## Abstract

Despite the popularity of the Adam optimizer in practice, most theoretical analyses study Stochastic Gradient Descent (SGD) as a proxy for Adam, and little is known about how the solutions found by Adam differ. In this paper, we show that Adam implicitly reduces a unique form of sharpness measure shaped by its adaptive updates, leading to qualitatively different solutions from SGD. More specifically, when the training loss is small, Adam wanders around the manifold of minimizers and takes semi-gradients to minimize this sharpness measure in an adaptive manner, a behavior we rigorously characterize through a continuous-time approximation using stochastic differential equations. We further demonstrate how this behavior differs from that of SGD in a well-studied setting: when training overparameterized models with label noise, SGD has been shown to minimize the trace of the Hessian matrix, $\mathrm{tr}(\boldsymbol{H})$, whereas we prove that Adam minimizes $\mathrm{tr}(\mathrm{Diag}(\boldsymbol{H})^{1/2})$ instead. In solving sparse linear regression with diagonal linear networks, this distinction enables Adam to achieve better sparsity and generalization than SGD. Finally, our analysis framework extends beyond Adam to a broad class of adaptive gradient methods, including RMSProp, Adam-mini, Adalayer and Shampoo, and provides a unified perspective on how these adaptive optimizers reduce sharpness, which we hope will offer insights for future optimizer design.

## 1 Introduction

Due to the non-convexity of the loss landscape, neural networks trained in different ways can perform very differently on the test set, even if they achieve the same training loss or accuracy [Zhang et al., 2017, Keskar et al., 2017, Liu et al., 2023, Saunshi et al., 2024]. To mathematically understand the generalization of neural networks, especially for over-parameterized models that admit many global minimizers, a key step is to understand the *implicit bias* of optimization methods [Neyshabur et al., 2014, Soudry et al., 2018]. That is, beyond just minimizing the training loss, *what kinds of solutions are different optimizers implicitly biased towards?*

Many theoretical works on implicit bias focus on (full-batch) gradient descent or its continuous variant, gradient flow. This includes the works on the implicit bias towards max-margin classifiers [Soudry et al., 2018, Nacson et al., 2019, Lyu and Li, 2020, Ji and Telgarsky, 2020], implicit bias towards min-norm solutions [Lyu et al., 2024], and equivalence to kernel methods [Jacot et al., 2018, Chizat et al., 2019]. However, these characterizations do not highlight the specific role of stochasticity in SGD, although it is more widely used in practice than gradient flow or full-batch gradient descent.

---

[*]Equal contribution; alphabet ordering

[†]Most work was done while Haodong was at Xi'an Jiaotong University.

[‡]Corresponding author.

39th Conference on Neural Information Processing Systems (NeurIPS 2025).

Another line of works [Blanc et al., 2020, Damian et al., 2021, Li et al., 2021b] demonstrate that the gradient noise in SGD induces an additional form of implicit bias that reduces the *sharpness* of the solutions, a generalization measure that has been long observed to correlate with generalization [Hochreiter and Schmidhuber, 1997, Keskar et al., 2017, Jiang et al., 2020, Foret et al., 2021]. More specifically, these works focus on the dynamics of SGD when the training loss is already small and the iterates are close to a manifold of minimizers. Li et al. [2021b] introduced a general framework to analyze the dynamics of SGD near the minimizer manifold, showing that SGD will not stop at arbitrary global minimizers, but drift and diffuse around the manifold, driving the iterates towards flatter regions of the loss landscape.

This behavior is mathematically characterized by a Stochastic Differential Equation (SDE), termed as *slow SDE* [Gu et al., 2023a]. This SDE accurately captures the projected dynamics of SGD near the minimizer manifold over a timescale of $\mathcal{O}(\eta^{-2})$, and reveals that SGD behaves like a (semi-)gradient method on the manifold, taking semi-gradients to minimize a specific sharpness measure determined by the Hessian and gradient noise. See Section 3 for more details.

However, SGD is rarely used directly in modern deep learning. Instead, Adaptive Gradient Methods (AGMs) have become the de facto standard for training neural networks. Among them, Adam [Kingma and Ba, 2014] innovatively combines the moving average of the first and second moments of gradients to determine an adaptive learning rate for each parameter. Across a wide range of domains, Adam provides faster convergence and better stability than SGD [Ashish, 2017, Dosovitskiy et al., 2020, Schulman et al., 2017, Zhang et al., 2024c].

Despite the popularity of Adam, little is known about its implicit bias, especially how it is different from SGD in terms of reducing sharpness. In the literature, Ma et al. [2023] made attempts to generalize the slow SDE framework from SGD to Adam, but their analysis is specific to a two-dimensional loss function and involves a quasistatic approximation that lacks full mathematical rigor. Other works, such as Liu et al. [2023] and Gu et al. [2024], leverage insights from the slow SDE developed for SGD to interpret empirical observations with Adam, but do not provide a theoretical analysis of Adam's own dynamics. A rigorous analysis of Adam's implicit bias in terms of sharpness remains an open problem.

**Our Contributions.** In this paper, we show that Adam biases the iteration towards flatter regions in a way that differs from SGD and implicitly reduces a unique form of sharpness. We formally prove separations between SGD and Adam under concrete theoretical settings:

1. In Section 4, we generalize the slow SDE for SGD to Adam. Our slow SDE approximates the dynamics of Adam near the minimizer manifold, and reveals that Adam behaves like an adaptive gradient method that minimizes a unique form of sharpness by taking semi-gradients on the manifold.

2. In Appendix B, we formally prove the generalization benefit of Adam when training overparameterized models with label noise. In this setting, we show that the implicit regularizer of Adam becomes $\mathrm{tr}(\mathrm{Diag}(\boldsymbol{H})^{1/2})$, where $\boldsymbol{H}$ denotes the Hessian matrix. Compared to the implicit regularizer $\mathrm{tr}(\boldsymbol{H})$ of SGD in the same setting, this unique form of sharpness reduction can yield sparser solutions when the model parameterization aligns with the underlying problem structure. We empirically verify this theoretical prediction in the setting of solving sparse linear regression with diagonal linear networks [Woodworth et al., 2020], where Adam recovers the sparse ground truth with less data. In Appendix C, we further present a failure case of Adam's generalization: Adam does not outperform SGD in deep matrix factorization, likely because minimizing $\mathrm{tr}(\mathrm{Diag}(\boldsymbol{H})^{1/2})$ does not favor low-rank solutions.

3. On the technical side, our analysis framework extends beyond Adam to a broad class of adaptive gradient methods (AGMs), including Adam, RMSProp, Adam-mini, Adalayer, and Shampoo. We develop several techniques that may be of independent interest, including a manifold projection operator tailored for AGMs, and a high-probability convergence analysis of AGMs under Polyak-Łojasiewicz conditions that directly bounds $\mathcal{L}(\boldsymbol{\theta}_k) - \mathcal{L}^*$.

## 2 Related Work

**Implicit Bias of SGD.** A line of works studies the implicit bias of SGD when training overparameterized models with label noise. Blanc et al. [2020] proved that with $\ell_2$ loss and label noise, with

high probability, SGD will move away from points on the minimizer manifold $\Gamma$ if and only if $\mathrm{tr}(\boldsymbol{H})$ is not locally minimal within $\Gamma$. HaoChen et al. [2021] further extended this local characterization to a global convergence result under the diagonal net setting [Woodworth et al., 2020]. However, HaoChen et al. [2021] relied on a manually designed, non-constant learning-rate schedule. Damian et al. [2021] overcame this issue by proving that the same implicit bias holds under constant learning rates and for any smooth loss satisfying the Kurdyka–Łojasiewicz condition. They also proved the same implicit bias for SGD with momentum (SGDM).

Up to this point, all analyses were not able to track the optimization over $\mathcal{O}(\eta^{-2})$ iterations, but this is necessary to capture the entire sharpness reduction process after converging to the minimizer manifold. Li et al. [2021b] were the first to tackle this problem, which provided an accurate SDE-based characterization of the long-term behavior of SGD over $\mathcal{O}(\eta^{-2})$ time. Their SDE characterization also goes beyond the label noise case and is able to capture the implicit bias induced by general forms of gradient noise. However, the SDE derivation in Li et al. [2021b] is specific to plain SGD and cannot be directly extended to other optimizers. Gu et al. [2023b] later termed this form of SDE *the slow SDE* and derived it for local SGD [Lin et al., 2018] in a way that is potentially generalizable to a broader class of optimizers. Wang et al. [2023] reinforced that momentum does not alter the implicit bias by showing that SGD and SGDM are characterized by the same slow SDE.

Another line of works characterizes the implicit bias of SGD through the lens of *Gradient Regularization (GR)*. Li et al. [2018] first derived a *stochastic modified equation* of SGD, showing that introducing an additional gradient norm regularizer to the drift term of SDE helps approximate SGD more accurately. Later Barrett and Dherin [2020] also discovered the same regularizer that can be added to gradient flow to approximate GD in a higher order, and termed this behavior the *Implicit Gradient Regularization (IGR)*. Smith et al. [2021] specifically studied and highlighted the IGR for SGD. Empirical studies [Geiping et al., 2022, Novack et al., 2022] showed that GR is not necessarily implicit by demonstrating that explicitly regularizing gradient norm for SGD with larger or even full batch size can recover the generalization performance of small-batch SGD. GR was also found by Karakida et al. [2023] to be connected with Sharpness-Aware Minimization (SAM) [Foret et al., 2021] and inspired improvements upon SAM in terms of generalization performance [Zhao et al., 2022]. The IGR method was later generalized to the analysis of as Adam and AdamW by Cattaneo et al. [2024] and Cattaneo and Shigida [2025]. We further discuss our relationship to Cattaneo et al. [2024] in the next paragraph.

**Implicit Bias of Adam.** On the theoretical side, the current literature still lacks a rigorous understanding of the implicit bias of Adam, although it has been used more widely than SGD in practical deep learning, especially for large language models. However, many efforts have been made on this problem. Qian and Qian [2019] and Xie and Li [2024] characterized the implicit biases of AdaGrad and AdamW, respectively. However, their methods do not extend to Adam. Wang et al. [2021] showed that Adam's implicit regularizer is identical to that of SGD, while their result requires that the gradient coordinates have magnitude at most $\epsilon$, which is typically not feasible in practice. Zhang et al. [2024a]'s analysis is also limited in scope, as it focuses on Adam's implicit bias on linearly separable data, a condition generally not met by real-world applications. A work using IGR to analyze the implicit bias of Adam, Cattaneo et al. [2024], seems similar at first glance to our study, but we are actually offering a different angle on the implicit bias of Adam. In particular, Cattaneo et al. [2024] argued that full-batch Adam with constant learning rate approximately follows an ODE that anti-regularizes sharpness when $\beta_1 < \beta_2$. Our work analyzes the dynamics of Adam for $\mathcal{O}(\eta^{-2})$ steps, a longer horizon than Cattaneo et al. [2024]. Our analysis shows that with gradient noise, Adam can be characterized by a slow SDE that regularizes sharpness in the long term, offering a complementary perspective.

**Approximation of Stochastic First-Order Optimizers with SDE.** Optimizers such as SGD and Adam take $\tilde{\mathcal{O}}(\eta^{-1})$ steps to converge onto the manifold, and then move along the manifold for $\mathcal{O}(\eta^{-2})$ steps, during which the optimizer will be dominated by a slower implicit regularization dynamics, different from the convergence dynamics earlier. Before slow SDE was introduced by Li et al. [2021b], the conventional SDE approximations [Li et al., 2018, 2021a, Cattaneo et al., 2024, Malladi et al., 2024] track the iteration during the convergence phase, but they struggle to bound the approximation error if extended to the manifold phase. Slow SDE tackles this problem by peeling the convergence dynamics off and approximating the iteration's projection on the manifold only. In this way, slow SDE manages to track iteration throughout the manifold phase for $\mathcal{O}(\eta^{-2})$ time. This idea will be made explicit in Section 3. In this work, to fill in the gaps and provide a theoretical analysis

that tracks iterations of Adam for a sufficiently long time, we adopt the tool of slow SDE as in the aforementioned Li et al. [2021b] and Gu et al. [2023a].

**Adaptive Gradient Methods.** As a test-of-time optimizer that has revolutionized the field of deep learning [Kingma and Ba, 2014], Adam innovatively combined the moving average of the first and second moments of gradients to determine an adaptive learning rate. Adam has also spawned a family of derivative optimizers such as AdamW [Loshchilov and Hutter, 2017], AdaFactor [Shazeer and Stern, 2018], Adam-mini [Zhang et al., 2025], Adalayer [Zhao et al., 2025] and AdaSGD [Wang and Wiens, 2020], maintaining significant advantages over SGD in terms of empirical use. Under a more general framework of adaptive gradient methods, many optimizers also get huge success as adaptive gradient methods, such as RMSprop [Hinton et al., 2012] and Adafactor [Shazeer and Stern, 2018].

**Convergence of Adam.** There have been many previous works discussing the convergence bound of Adam. For example, Reddi et al. [2018] and Dereich and Jentzen [2024] give convergence bounds under the convexity condition, Zou et al. [2019], Shi and Li [2021] and Zhang et al. [2022] focus on the cases where learning rates follow a $1/\sqrt{t}$ decay, and the bounds given by Zaheer et al. [2018], Zhang et al. [2022] and Wang et al. [2024b] do not decrease to $0$ as $\eta \to 0$. Also, most works [Défossez et al., 2020, Guo et al., 2025, Iiduka, 2022, Wang et al., 2024a, Zhang et al., 2024b, Hong and Lin, 2023] only establish an upper bound on the average of gradient norms over the time of iteration. In this work, we derive a high-probability convergence analysis that directly bound the loss term of the last step, $\mathcal{L}(\boldsymbol{\theta}_k) - \mathcal{L}^*$, to $o(1)$. Going beyond convex loss functions, we establish the bound on $\mu$-PL functions, and we focus on the constant learning rate schedule.

# 3 Preliminaries

**Notations.** Unless otherwise stated, for a square matrix $\boldsymbol{M}$, we use $\mathrm{diag}(\boldsymbol{M})$ to denote the vector consisting of its diagonal entries. The notation $\mathrm{Diag}$ (with capital "D") has two related usages: (1) for a vector $\boldsymbol{v}$, $\mathrm{Diag}(\boldsymbol{v})$ denotes the diagonal matrix with $\boldsymbol{v}$ on its diagonal; and (2) for a square matrix $\boldsymbol{M}$, $\mathrm{Diag}(\boldsymbol{M})$ denotes the diagonal matrix that keeps only the diagonal entries of $\boldsymbol{M}$ and zeros out the off-diagonal entries, i.e., $\mathrm{Diag}(\boldsymbol{M}) := \mathrm{Diag}(\mathrm{diag}(\boldsymbol{M}))$. For two vectors $\boldsymbol{u}, \boldsymbol{v}$ with the same dimension $d$, $\boldsymbol{u} \odot \boldsymbol{v}$ denotes their element-wise product $(u_1 v_1, \ldots, u_d v_d)$. For any exponent $p > 0$, $\boldsymbol{v}^{\odot p}$ denotes element-wise exponentiation, i.e., $\boldsymbol{v}^{\odot p} := (v_1^p, \ldots, v_d^p)$, and $\sqrt{\boldsymbol{v}}$ stands for $\boldsymbol{v}^{\odot 1/2}$. We use $\mathbb{R}_{\geq 0}^d$ to denote the set of $d$-dimensional vectors with non-negative entries, and $\mathbb{S}_{++}^d$ to denote the set of $d \times d$ symmetric positive definite matrices. Given a pint $\boldsymbol{\theta}$ and the manimizer manifold $\Gamma$, we let $\mathrm{dist}(\boldsymbol{\theta}, \Gamma) = \min_{\boldsymbol{v} \in \Gamma} \|\boldsymbol{\theta} - \boldsymbol{v}\|_2$ be the $\ell_2$ distance between $\boldsymbol{\theta}$ and the manifold $\Gamma$.

**Derivatives.** For any scalar-valued function $f : \mathbb{R}^d \to \mathbb{R}$, we write $\nabla f(\boldsymbol{\theta})$ for its gradient at $\boldsymbol{\theta}$, and $\nabla_\Gamma f(\boldsymbol{\theta})$ for the projection of this gradient onto the tangent space of a manifold $\Gamma$. For a vector-valued function $F : \mathbb{R}^d \to \mathbb{R}^d$, we denote its Jacobian at $\boldsymbol{\theta} \in \mathbb{R}^d$ as $\partial F(\boldsymbol{\theta}) \in \mathbb{R}^{d \times d}$, and its second-order derivative as $\partial^2 F(\boldsymbol{\theta})$, which is a third-order tensor. Given a matrix $\boldsymbol{M} \in \mathbb{R}^{d \times d}$, we define $\partial^2 F(\boldsymbol{\theta})[\boldsymbol{M}]$ as the second-order directional derivative of $F$ at $\boldsymbol{\theta}$ in the direction of $\boldsymbol{M}$, $\partial^2 F(\boldsymbol{\theta})[\boldsymbol{M}] := \sum_{i \in [d]} \left\langle \frac{\partial^2 F_i}{\partial \boldsymbol{\theta}^2}, \boldsymbol{M} \right\rangle \boldsymbol{e}_i$, where $F_i$ denotes the $i$-th coordinate of $F$, and $\boldsymbol{e}_i$ the $i$-th vector of the standard basis. When the context is clear, for any scalar-valued function $\mathcal{L} : \mathbb{R}^d \to \mathbb{R}$, we abbreviate $\partial^2 (\nabla \mathcal{L})(\boldsymbol{\theta})[\boldsymbol{M}]$ as $\nabla^3 \mathcal{L}(\boldsymbol{\theta})[\boldsymbol{M}]$.

**Loss Functions.** Define $\ell(\boldsymbol{\theta}; \xi)$ as the loss function for a data sample $\xi$ for a model with parameters $\boldsymbol{\theta}$. Define $\mathcal{L}(\boldsymbol{\theta}) := \mathbb{E}_{\xi \sim \mathcal{S}}[\ell(\boldsymbol{\theta}; \xi)]$ as the training loss function, where $\mathcal{S}$ is the training dataset and $\xi \sim \mathcal{S}$ means the data sample $\xi$ is drawn from $\mathcal{S}$ uniformly at random. Let $\mathcal{L}^* := \min_{\boldsymbol{\theta} \in \mathbb{R}^d} \mathcal{L}(\boldsymbol{\theta})$ be the minimum of training loss. Let $\mathcal{Z}(\boldsymbol{\theta})$ be the distribution of gradient noise $\nabla \ell(\boldsymbol{\theta}; \xi) - \nabla \mathcal{L}(\boldsymbol{\theta})$, which is a random variable that depends on $\boldsymbol{\theta}$. We define $\boldsymbol{\Sigma}(\boldsymbol{\theta}) := \mathbb{E}_{\boldsymbol{z} \sim \mathcal{Z}(\boldsymbol{\theta})}[\boldsymbol{z}\boldsymbol{z}^\top]$ as the noise covariance matrix of gradients at $\boldsymbol{\theta}$.

**Smoothness Assumptions.** We make the following smoothness assumptions on the loss function and the gradient noise distribution.

**Assumption 3.1.** *The loss function $\mathcal{L}$ and the matrix square root of the noise covariance $\boldsymbol{\Sigma}^{1/2}$ are $\mathcal{C}^5$-smooth on $\mathbb{R}^d$, i.e. all their partial derivatives up to order $5$ exist and are continuous.*

Assuming smoothness on the loss function is a common practice in optimization analysis. Here, we specifically assume the $\mathcal{C}^5$-smoothness, which we found to be a minimal smoothness requirement for our proof to hold for all $\mathcal{C}^3$-smooth test functions in Theorem 4.1.

Moreover, we assume that the smoothness constant of $\mathcal{L}$ and the gradient noise are globally bounded:

**Assumption 3.2.** *$\mathcal{L}$ is $\rho$-smooth on $\mathbb{R}^d$, i.e. $\forall \boldsymbol{\theta}_1, \boldsymbol{\theta}_2 \in \mathbb{R}^d$, $\|\nabla\mathcal{L}(\boldsymbol{\theta}_1) - \nabla\mathcal{L}(\boldsymbol{\theta}_2)\|_2 \leq \rho\|\boldsymbol{\theta}_1 - \boldsymbol{\theta}_2\|_2$ and $\mathcal{L}$ is bounded from below, i.e. $\mathcal{L}^* = \inf_{\boldsymbol{\theta}} \mathcal{L}(\boldsymbol{\theta}) > -\infty$.*

**Assumption 3.3.** *The noisy gradients are $\ell_2$-bounded, i.e., there exists some constant $R$ s.t. $\forall \boldsymbol{\theta} \in \mathbb{R}^d$, $\|\nabla\ell(\boldsymbol{\theta}; \xi)\|_2 \leq R$ almost surely for training data sample $\xi \sim \mathcal{S}_{\text{train}}$.*

**SGD and Adam.** SGD is an iterative method that starts from an initial point $\boldsymbol{\theta}_0$ and updates the parameters as $\boldsymbol{\theta}_{k+1} := \boldsymbol{\theta}_k - \eta\nabla\ell_k(\boldsymbol{\theta}_k)$ for all $k \geq 0$, where $\eta$ is the learning rate, $\ell_k(\boldsymbol{\theta})$ is the loss function for the data sample $\xi_k$ sampled at step $k$. Adam [Kingma and Ba, 2014] is a popular optimizer that updates the parameters as:

$$\boldsymbol{m}_{k+1} := \beta_1\boldsymbol{m}_k + (1 - \beta_1)\nabla\ell_k(\boldsymbol{\theta}_k)$$
$$\boldsymbol{v}_{k+1} := \beta_2\boldsymbol{v}_k + (1 - \beta_2)\nabla\ell_k(\boldsymbol{\theta}_k)^{\odot 2}$$
$$\theta_{k+1,i} := \theta_{k,i} - \eta\frac{m_{k+1,i}}{\sqrt{v_{k+1,i}} + \epsilon} \quad \text{for all } i \in [d].$$

Note that in practice, it is common to normalize $\boldsymbol{m}_{k+1}$ and $\boldsymbol{v}_{k+1}$ by $1 - \beta_1^{k+1}$ and $1 - \beta_2^{k+1}$ respectively before the division. However, this normalization quickly becomes neglectable when $k$ is large, so we ignore it for simplicity.

**SDE First-Order Approximation For SGD.** A *Stochastic Differential Equation* (SDE) is an extension of an ordinary differential equation that incorporates random perturbations, and is widely used to model systems under the influence of noise. An SDE on $\mathbb{R}^d$ takes the form $\mathrm{d}\boldsymbol{\theta}_t = b(\boldsymbol{\theta}_t)\mathrm{d}t + \sigma(\boldsymbol{\theta}_t)\mathrm{d}\boldsymbol{W}_t$ where $b : \mathbb{R}^d \to \mathbb{R}^d$ is the drift vector field, $\sigma : \mathbb{R}^d \to \mathbb{R}^{d \times m}$ is the diffusion matrix, and $\{\boldsymbol{W}_t\}_{t \geq 0}$ is an $m$-dimensional Wiener process. A line of works [Li et al., 2015, Jastrzębski et al., 2017, Li et al., 2017, Smith et al., 2020, Li et al., 2019, 2021a] used the following SDE to serve as a first-order approximation of SGD, which we refer to as the *conventional SDE*:

$$\mathrm{d}\boldsymbol{\theta}_t = -\nabla\mathcal{L}(\boldsymbol{\theta}_t)\mathrm{d}t + \sqrt{\eta}\boldsymbol{\Sigma}^{1/2}(\boldsymbol{\theta}_t)\mathrm{d}\boldsymbol{W}_t,$$

where the stochastic integral is taken in the Itô sense. For an introduction to Itô calculus, see Oksendal [2013]. Later, Malladi et al. [2024] extended this type of SDE to Adam. Besides these conventional SDEs, below we introduce another type of SDE, slow SDE, that can more explicitly capture the implicit bias of SGD near a manifold of minimizers.

**Manifold Assumption.** Before going into the slow SDE, we introduce the *manifold assumption*. Previous studies [Garipov et al., 2018, Kuditipudi et al., 2019] have found that low-loss solutions are in fact connected to each other, a phenomenon known as mode connectivity. Wen et al. [2024] provided empirical evidence that the training dynamics of language model training usually happen in a structure similar to a river valley, where many low-loss solutions lie in the bottom of the valley. Motivated by these observations, many previous works [Li et al., 2021b, Fehrman et al., 2020, Lyu and Li, 2020, Gu et al., 2023a] assumed that the minimizers of the training loss function are not isolated points but connected and form a manifold $\Gamma$:

**Assumption 3.4.** *$\Gamma$ is $\mathcal{C}^\infty$-smooth, $(d - m)$-dimensional compact submanifold of $\mathbb{R}^d$, where any $\boldsymbol{\zeta} \in \Gamma$ is a local minimizer of $\mathcal{L}$. For all $\boldsymbol{\zeta} \in \Gamma$, $rank(\nabla^2\mathcal{L}(\boldsymbol{\zeta})) = m$. Additionally, there exists an open neighborhood of $\Gamma$, denoted as $U$, such that $\Gamma = \arg\min_{\boldsymbol{\theta} \in U} \mathcal{L}(\boldsymbol{\theta})$.*

With this assumption, if an optimization process converges and the learning rate $\eta$ is sufficiently small, then the process will be trapped near some minimizer manifold which we denote by $\Gamma$.

**Slow SDE.** A line of works [Blanc et al., 2020, Damian et al., 2021, Li et al., 2021b] studied the dynamics of SGD near the manifold $\Gamma$ and showed that SGD has an implicit bias towards flatter minimizers on $\Gamma$. This effect cannot be directly seen from conventional SDEs, so Li et al. [2021b] derived a new type of SDE approximation, called slow SDE, that can explicitly capture this effect. See Appendix A for an illustration of the difference between conventional SDEs and slow SDEs. Here we introduce the slow SDE for SGD following the formulation in Gu et al. [2024]. For ease of presentation, we define the following projection operators $\Phi$, $P_{\boldsymbol{\zeta}}$ for points and differential forms respectively. Consider the gradient flow $\frac{\mathrm{d}\boldsymbol{x}(t)}{\mathrm{d}t} = -\nabla\mathcal{L}(\boldsymbol{x}(t))$ with $\boldsymbol{x}(0) = \boldsymbol{x}$, and fix some point $\boldsymbol{\theta}_{\text{null}} \notin \Gamma$, we define the gradient flow projection of any $\boldsymbol{x}$, $\Phi(\boldsymbol{x})$, as $\lim_{t \to +\infty} \boldsymbol{x}(t)$ if the limit exists and belongs to $\Gamma$, and $\boldsymbol{\theta}_{\text{null}}$ otherwise. It can be shown by simple calculus [Li et al., 2021b] that $\partial\Phi(\boldsymbol{\zeta})$

equals the projection matrix onto the tangent space of $\Gamma$ at $\boldsymbol{\zeta}$. We decompose the noise covariance $\boldsymbol{\Sigma}(\boldsymbol{\zeta})$ for $\boldsymbol{\zeta} \in \Gamma$ into two parts: the noise in the tangent space $\boldsymbol{\Sigma}_{\parallel}(\boldsymbol{\zeta}) := \partial\Phi(\boldsymbol{\zeta})\boldsymbol{\Sigma}(\boldsymbol{\zeta})\partial\Phi(\boldsymbol{\zeta})$ and the noise in the normal space $\boldsymbol{\Sigma}_{\Diamond}(\boldsymbol{\zeta}) := \boldsymbol{\Sigma}(\boldsymbol{\zeta}) - \boldsymbol{\Sigma}_{\parallel}(\boldsymbol{\zeta})$.

For any $\boldsymbol{\zeta} \in \Gamma$, matrix $\boldsymbol{A}$ and vector $\boldsymbol{b}$, we use $P_{\boldsymbol{\zeta}}(\boldsymbol{A}\mathrm{d}\boldsymbol{W}_t + \boldsymbol{b}\mathrm{d}t)$ to denote $\Phi(\boldsymbol{\zeta}+\boldsymbol{A}\mathrm{d}\boldsymbol{W}_t+\boldsymbol{b}\mathrm{d}t) - \Phi(\boldsymbol{\zeta})$, which equals $\partial\Phi(\boldsymbol{\zeta})\boldsymbol{A}\mathrm{d}\boldsymbol{W}_t + \left(\partial\Phi(\boldsymbol{\zeta})\boldsymbol{b} + \frac{1}{2}\partial^2\Phi(\boldsymbol{\zeta})[\boldsymbol{A}\boldsymbol{A}^{\top}]\right)\mathrm{d}t$ by Itô calculus. $P_{\boldsymbol{\zeta}}$ can be interpreted as projecting an infinitesimal step from $\boldsymbol{\zeta}$, so that $\boldsymbol{\zeta}$ after taking the projected step does not leave the manifold $\Gamma$. Now we are ready to state the slow SDE for SGD.

**Definition 3.1** (Slow SDE for SGD). *Given $\eta > 0$ and $\boldsymbol{\zeta}_0 \in \Gamma$, define $\boldsymbol{\zeta}(t)$ as the solution of the following SDE with initial condition $\boldsymbol{\zeta}(0) = \boldsymbol{\zeta}_0$:*

$$\mathrm{d}\boldsymbol{\zeta}(t) = P_{\boldsymbol{\zeta}}\Big(\underbrace{\boldsymbol{\Sigma}_{\parallel}^{1/2}(\boldsymbol{\zeta})\mathrm{d}\boldsymbol{W}_t}_{(a)\ diffusion} - \underbrace{\frac{1}{2}\nabla^3\mathcal{L}(\boldsymbol{\zeta})\big[\widehat{\boldsymbol{\Sigma}}_{\Diamond}(\boldsymbol{\zeta})\big]\mathrm{d}t}_{(b)\ drift}\Big). \tag{1}$$

*Here $\widehat{\boldsymbol{\Sigma}}_{\Diamond}(\boldsymbol{\zeta})$ is defined as $\sum_{i,j:\ \lambda_i \neq 0 \vee \lambda_j \neq 0} \frac{1}{\lambda_i + \lambda_j}\langle\boldsymbol{\Sigma}_{\Diamond}(\boldsymbol{\zeta}), \boldsymbol{v}_i\boldsymbol{v}_j^{\top}\rangle\boldsymbol{v}_i\boldsymbol{v}_j^{\top}$, where $\{\boldsymbol{v}_i\}_{i=1}^d$ is an orthonormal eigenbasis of $\nabla^2\mathcal{L}(\boldsymbol{\zeta})$ with corresponding eigenvalues $\lambda_1,\ldots,\lambda_d$.*

**Interpretation of the Slow SDE for SGD: Semi-gradient Descent**    This SDE on the minimizer manifold $\Gamma$ splits naturally into a *diffusion* term $P_{\boldsymbol{\zeta}}\big(\boldsymbol{\Sigma}_{\parallel}^{1/2}(\boldsymbol{\zeta})\,\mathrm{d}\boldsymbol{W}_t\big)$ injecting noise in the tangent space, and a *drift* term $-\frac{1}{2}P_{\boldsymbol{\zeta}}\big(\nabla^3\mathcal{L}(\boldsymbol{\zeta})\big[\widehat{\boldsymbol{\Sigma}}_{\Diamond}(\boldsymbol{\zeta})\big]\mathrm{d}t\big)$ that can be seen as the negative *semi-gradient* of the following sharpness measure:

$$\mu(\boldsymbol{\zeta}) := \left\langle\nabla^2\mathcal{L}(\boldsymbol{\zeta}), \widehat{\boldsymbol{\Sigma}}_{\Diamond}(\boldsymbol{\zeta})\right\rangle.$$

Here we use the word "semi-gradient" [Mnih et al., 2015, Brandfonbrener and Bruna, 2019] because it is not exactly the gradient of $\mu(\boldsymbol{\zeta})$ but only the gradient with respect to the first argument of the inner product. More specifically, define $\mu(\boldsymbol{\zeta}_1, \boldsymbol{\zeta}_2) := \left\langle\nabla^2\mathcal{L}(\boldsymbol{\zeta}_1), \widehat{\boldsymbol{\Sigma}}_{\Diamond}(\boldsymbol{\zeta}_2)\right\rangle$, then the drift term is essentially $-\frac{1}{2}\left.\nabla_{\boldsymbol{\zeta}_1}\mu(\boldsymbol{\zeta}_1, \boldsymbol{\zeta}_2)\right|_{\boldsymbol{\zeta}_1=\boldsymbol{\zeta}, \boldsymbol{\zeta}_2=\boldsymbol{\zeta}}$ after projecting onto the tangent space of $\Gamma$ at $\boldsymbol{\zeta}$. In other words, SGD near manifold takes semi-gradients to minimize the implicit regularizer $\langle\nabla^2\mathcal{L}(\boldsymbol{\zeta}), \widehat{\boldsymbol{\Sigma}}_{\Diamond}(\boldsymbol{\zeta})\rangle$ but pretend $\widehat{\boldsymbol{\Sigma}}_{\Diamond}(\boldsymbol{\zeta})$ to be fixed, i.e. ignore the dependency of $\widehat{\boldsymbol{\Sigma}}_{\Diamond}(\boldsymbol{\zeta})$ on $\boldsymbol{\zeta}$.

**Example: Noisy Ellipse.**    We provide a toy example to illustrate the phenomenon described by the slow SDE for SGD: there are two parameters $x, y$ and an elliptical loss with label noise $\mathcal{L}(x, y) = \frac{1}{2}\left(\frac{(x+y)^2}{2a^2} + \frac{(y-x)^2}{2b^2} - 1 - \delta\right)^2$. The label noise $\delta$ is sampled uniformly from $\{-0.5, 0.5\}$ at every step. As depicted in Fig. 1, SGD moves towards flatter minimizers after reaching the manifold. The same phenomenon can be observed for Adam, but Adam converges to a different minimizer that is closer to the axis (or, "sparser" in the parameter space). Understanding the difference between SGD and Adam is the main focus of this paper.

# 4 Theoretical Analysis of Adam

In this section, we generalize the slow SDE for SGD to a general class of adaptive gradient methods (AGMs), including Adam. We first present our novel slow SDE for a general class of AGMs, including Adam, and give an intuitive explanation for our results. Then, we discuss the difficulty of directly applying the slow SDE framework to Adam and other AGMs and how we resolve the problems.

**A General Class of Adaptive Gradient Methods.**    We define a general class of AGMs as follows:

$$\boldsymbol{m}_{k+1} := \beta_1\boldsymbol{m}_k + (1 - \beta_1)\nabla\ell_k(\boldsymbol{\theta}_k)$$
$$\boldsymbol{v}_{k+1} := \beta_2\boldsymbol{v}_k + (1 - \beta_2)V\big(\nabla\ell_k(\boldsymbol{\theta}_k)\nabla\ell_k(\boldsymbol{\theta}_k)^{\top}\big)$$
$$\boldsymbol{\theta}_{k+1} := \boldsymbol{\theta}_k - \eta S(\boldsymbol{v}_{k+1})\boldsymbol{m}_{k+1},$$

where $d$ is the dimension of the parameter $\boldsymbol{\theta}$ and the first order momentum $\boldsymbol{m}$, and $D$ is the dimension of the vector $\boldsymbol{v}$ that encodes information related to second order momentum. For all optimizers but Shampoo in Table 1, we set $D = d$, while for Shampoo we have $D = d_1^2 + d_2^2$ if the matrix-like parameters have shape $(d_1, d_2)$. See Appendix J for a detailed explanation.

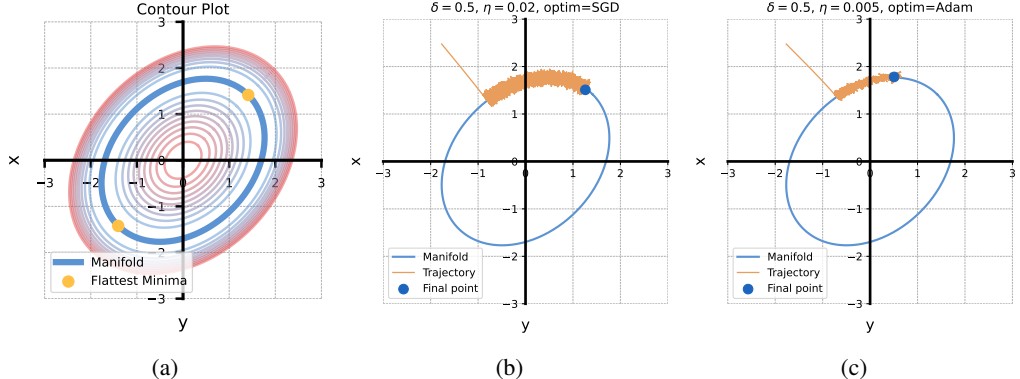

Figure 1: **(a)**: Coutour of the elliptical loss, from which we can see the two tips as the flattest minima. **(b)**: SGD implicitly minimizes $\text{tr}(\boldsymbol{H})$ and converges to the flattest minima. **(c)**: Adam reduces sharpness too but converges to a different and sparser minimizer.

The function $S : \mathbb{R}^D_{\geq 0} \longrightarrow \mathbb{S}^d_{++}$ is a $\rho_s$-smooth function for some $\rho_s > 0$, which maps a non-negative vector $\boldsymbol{v} \in \mathbb{R}^D_{\geq 0}$ to a symmetric positive definite matrix $S(\boldsymbol{v}) \in \mathbb{S}^d_{++}$. In addition, we require $S$ to satisfy $S(\boldsymbol{v}) \succeq \frac{1}{R_0}\boldsymbol{I}$ for some $R_0 > 0$ and any $\boldsymbol{v} \in \mathbb{R}^D_{\geq 0}$. We also require $V : \mathbb{R}^{d \times d} \longrightarrow \mathbb{R}^D$ to be a linear function that satisfies $V(\boldsymbol{g}\boldsymbol{g}^\top) \in \mathbb{R}^D_{\geq 0}$ for all $\boldsymbol{g} \in \mathbb{R}^d$, i.e., it always maps vector outer products to non-negative vectors.

A number of currently used optimization algorithms, such as RMSProp, Adam, Adam-mini, Adafactor[1], Adalayer, AdaSGD, and Shampoo[2], all fit this framework. Note that we do not consider weight decays or bias corrections in these optimizers. Some examples of $V$ and $S$ functions are listed in Table 1, including the AdamE-$\lambda$ optimizer that will be introduced in Appendix B as a tool to tune the implicit bias of Adam.

Prior to the results, we introduce two technical assumptions: $S$ satisfies a mild smoothness condition, and $1 - \beta_1$ is of constant order.

**Assumption 4.1.** *The function $S$ is $\mathcal{C}^4$-smooth on $\mathbb{R}^D_{\geq 0}$.*

**Assumption 4.2.** $\beta_1 \leq 0.9$.

**Remark 4.1.** *The threshold $0.9$ in Assumption 4.2 can also be replaced by any constant below $1$, and the approximation rate in our result will remain unaffected. So we actually consider the regime where $b_1 := 1 - \beta_1$ is of constant order. For real-world Natural Language Processing (NLP) models, BERT [Devlin et al., 2019], Transformer [Vaswani et al., 2017] and GPT [Radford et al., 2018] all use $\beta_1 = 0.9$. In computer vision (CV), pix2pix [Isola et al., 2017] uses $\beta_1 = 0.5$, while U-Net [Ronneberger et al., 2015] and ViT [Dosovitskiy et al., 2020] use $\beta_1 = 0.9$. Thus, assuming $\beta_1 \leq 0.9$ is consistent with standard practice across multiple aspects.*

### 4.1 Slow SDE Analysis for AGMs

Our SDE for AGMs characterizes the training dynamics near the manifold $\Gamma$. First we rigorously define the preconditioned projection mapping $\Phi_{\boldsymbol{S}}$ and the SDE projection formula as an extension to the $\Phi$ and $\boldsymbol{P}_\zeta$ mentioned in Section 3, after which we present the SDE for AGMs we derived.

**Definition 4.1** (Preconditioner Flow Projection). *Fix a point $\theta_{null} \notin \Gamma$. Given a Positive Semi-Definite matrix $\boldsymbol{S}$, for $x \in \mathbb{R}^d$, consider the preconditioner flow $\frac{\mathrm{d}x(t)}{\mathrm{d}t} = -\boldsymbol{S}\nabla\mathcal{L}(x(t))$ with $x(0) = x$. We denote the preconditioner flow projection of $x$ as $\Phi_{\boldsymbol{S}}(x)$, i.e. $\Phi_{\boldsymbol{S}}(x) := \lim_{t \to +\infty} x(t)$ if the limit exists and belongs to $\Gamma$, and $\Phi_{\boldsymbol{S}}(x) = \theta_{null}$ otherwise.*

---

[1]We ignore update clipping, i.e. we adopt the Algorithm 2 in Shazeer and Stern [2018].

[2]In practice, the Shampoo optimizer is often equipped with the exponential moving average (EMA) on the calculation of pre-conditioner [Morwani et al., 2024]. Here we adopt this practical version of Shampoo instead of the original one [Gupta et al., 2018].

**Definition 4.2.** *For any $\boldsymbol{\zeta} \in \Gamma$ and any differential form $\boldsymbol{A}\mathrm{d}\boldsymbol{W}_t + \boldsymbol{b}\mathrm{d}t$ in Itô calculus, where $\boldsymbol{A} \in \mathbb{R}^{d \times d}$, and $b \in \mathbb{R}^d$, we use $\boldsymbol{P}_{\boldsymbol{\zeta},\boldsymbol{S}}(\boldsymbol{A}\mathrm{d}\boldsymbol{W}_t + \boldsymbol{b}\mathrm{d}t)$ as a shorthand for the differential form $\partial\Phi_{\boldsymbol{S}}(\boldsymbol{\zeta})\boldsymbol{A}\mathrm{d}\boldsymbol{W}_t + \boldsymbol{S}\left(\partial\Phi_{\boldsymbol{S}}(\boldsymbol{\zeta})\boldsymbol{b} + \frac{1}{2}\partial^2\Phi_{\boldsymbol{S}}(\boldsymbol{\zeta})[\boldsymbol{A}\boldsymbol{A}^\top]\right)\mathrm{d}t.$*

**Definition 4.3** (Slow SDE for AGMs). *Given learning rate $\eta$ and the initial state $\boldsymbol{\zeta}_0 \in \Gamma$, $\boldsymbol{v}_0 \in \mathbb{R}^D_{\geq 0}$, let $c := \frac{1-\beta_2}{\eta^2}$ be a constant and denote $\boldsymbol{S}_t := S(\boldsymbol{v}(t))$, we define $\boldsymbol{\zeta}(t)$ as the solution of the following SDE with initial condition $(\boldsymbol{\zeta}(0), \boldsymbol{v}(0)) = (\boldsymbol{\zeta}_0, \boldsymbol{v}_0)$:*

$$
\begin{cases}
\mathrm{d}\boldsymbol{\zeta}(t) = P_{\boldsymbol{\zeta}(t),\boldsymbol{S}(t)}\left(\underbrace{\boldsymbol{\Sigma}_{\parallel}^{1/2}(\boldsymbol{\zeta}(t);\boldsymbol{S}(t))\mathrm{d}\boldsymbol{W}_t}_{\text{diffusion}} \underbrace{-\frac{1}{2}\boldsymbol{S}(t)\nabla^3\mathcal{L}(\boldsymbol{\zeta})\left[\boldsymbol{\Sigma}_{\diamond}(\boldsymbol{\zeta}(t);\boldsymbol{S}(t))\right]\mathrm{d}t}_{\text{drift}}\right), \\
\mathrm{d}\boldsymbol{v}(t) = \underbrace{c\left(V(\boldsymbol{\Sigma}(\boldsymbol{\zeta})) - \boldsymbol{v}\right)\mathrm{d}t}_{\text{preconditioner drift}}.
\end{cases}
\tag{2}
$$

$\boldsymbol{\Sigma}_{\diamond}(\boldsymbol{\zeta};\boldsymbol{S}) = \boldsymbol{S}\boldsymbol{\Sigma}(\boldsymbol{\zeta})\boldsymbol{S} - \boldsymbol{\Sigma}_{\parallel}(\boldsymbol{\zeta};\boldsymbol{S})$, $\boldsymbol{\Sigma}_{\parallel}(\boldsymbol{\zeta};\boldsymbol{S}) = \partial\Phi_{\boldsymbol{S}}(\boldsymbol{\zeta})\boldsymbol{S}\boldsymbol{\Sigma}(\boldsymbol{\zeta})\boldsymbol{S}\partial\Phi_{\boldsymbol{S}}(\boldsymbol{\zeta})$.

Note that the drift term in $\mathrm{d}\boldsymbol{\zeta}(t)$ can be interpreted as an *adaptive semi-gradient descent* process, in that this term drives the dynamics towards optimizing an adaptive loss function

$$
\mu(\boldsymbol{\zeta}, \boldsymbol{v}) = \langle\nabla^2\mathcal{L}(\boldsymbol{\zeta}), \boldsymbol{\Sigma}_{\diamond}(\boldsymbol{\zeta}(t);\boldsymbol{S}(t))\rangle
$$

as if $\boldsymbol{\Sigma}_{\diamond}(\boldsymbol{\zeta}(t);\boldsymbol{S}(t))$ has no dependence on $\boldsymbol{\zeta}$; also this gradient flow is preconditioned by a positive definite matrix $\boldsymbol{S}(t)$. Recall that the drift term in the slow SDE for SGD can be seen as a semi-gradient descent. In the AGM framework, it takes $\Theta(\eta^{-2})$ time for the preconditioner $\boldsymbol{S}(t)$ to make a significant (i.e. $\Theta(1)$) change, which coincides with the moving speed of the slow SDE of $\boldsymbol{\zeta}$. Therefore, compared to that of SGD, our SDE includes a new formula that tracks the motion of the preconditioner and injects adaptiveness accordingly in the semi-gradient descent process.

We prove that $\boldsymbol{\zeta}(t)$ always stays on the manifold $\Gamma$. And next, we present our main theorem showing that the above SDE in Equation (2) tracks the trajectory of Adam in a weak approximation sense.

**Theorem 4.1.** *Let $T > 0$ and suppose Assumption 3.1–4.2 hold. There exist constant $\epsilon_0, C > 0$ such that the following statement holds for all sufficiently small learning rates $\eta$. Define step $K_0 := \lfloor C\log(1/\eta)\rfloor$ and $K := \lfloor T\eta^{-2}\rfloor$. Run an AGM for $K_0 + K$ iterations and obtain $\{(\boldsymbol{\theta}_k, \boldsymbol{v}_k)\}_{k=0}^{K_0+K}$, where the initial parameter $\boldsymbol{\theta}_0$ is an arbitrary point that satisfies $\mathrm{dist}(\boldsymbol{\theta}_0, \Gamma) \leq \epsilon_0$. Let $\boldsymbol{X}(t) = (\boldsymbol{\zeta}(t), \boldsymbol{v}(t))$ be the solution to (2) on $t \in [0, T]$ with initial condition*

$$
\boldsymbol{X}(0) = \boldsymbol{X}_0 \in \Gamma \times \mathbb{R}^D_{\geq 0}, \quad \boldsymbol{X}_0 := (\Phi_{\boldsymbol{S}(\boldsymbol{v}_{K_0})}(\boldsymbol{\theta}_{K_0}), \boldsymbol{v}_{K_0}).
$$

*For any $k \in \{0, 1, \ldots, K\}$, let $\bar{\boldsymbol{X}}_k := \left(\Phi_{\boldsymbol{S}(\boldsymbol{v}_{K_0+k})}(\boldsymbol{\theta}_{K_0+k}), \boldsymbol{v}_{K_0+k}\right)$ be the AGM state at step $K_0 + k$ projected onto $\Gamma$. Then for any $C^3$ function $g(\boldsymbol{\theta})$,*

$$
\max_{0 \leq k \leq K}\left|\mathbb{E}\left[g\left(\bar{\boldsymbol{X}}_k\right)\big|\boldsymbol{X}_0\right] - \mathbb{E}\left[g\left(\boldsymbol{X}(k\eta^2)\right)\big|\boldsymbol{X}(0) = \boldsymbol{X}_0\right]\right| = \widetilde{\mathcal{O}}\left(\eta^{0.25}\right),
$$

*where $\widetilde{\mathcal{O}}(\cdot)$ hides logarithmic factors and constants independent of $\eta$ (but possibly depending on $g$).*

Theorem 4.1 shows that with a small $\eta$, once AGM approaches the minimizer manifold, its long-horizon behavior within $\widetilde{\mathcal{O}}(\eta^{-2})$ steps is captured by the SDE in Equation (2). A more explicit version of Theorem 4.1 can be found in Appendix D.

### 4.2 Interpretation of The Slow SDEs for AGMs

**Adaptive Projection Operator.** Equation (1) employs a fixed projection operator $P_{\boldsymbol{\zeta}}$ to constrain the SDE to the manifold. As a comparison, the slow SDE for AGM uses an adaptive projection $P_{\boldsymbol{\zeta},\boldsymbol{S}(t)}$ that depends on the current preconditioner $S(\boldsymbol{v}(t))$. In other words, SGD's projection is state-independent, but AGM's projection is state-dependent. This adaptive projection alters the way the stochastic trajectory evolves on the manifold, giving rise to a different implicit bias in AGMs versus SGD.

Table 1: Examples of optimizers in the AGM Framework. See Appendix I for derivations of their implicit regularizers under label noise.

| Optimizer | Functions $V/S$ | Implicit Regularizer (Label Noise, $\epsilon = 0$) | Remarks |
|---|---|---|---|
| SGD | **V:** $V(\boldsymbol{M}) = \mathbf{1_d}$ 
 **S:** $S(\boldsymbol{v}) = \boldsymbol{I}_d$ | $\mathrm{tr}(\boldsymbol{H})$ [Blanc et al., 2020] | |
| Adam | **V:** $V(\boldsymbol{M}) = \mathrm{diag}(\boldsymbol{M})$ 
 **S:** $S(\boldsymbol{v}) = \mathrm{Diag}(1/(\sqrt{\boldsymbol{v}} + \epsilon))$ | $\mathrm{tr}\left(\mathrm{Diag}(\boldsymbol{H})^{1/2}\right)$ | |
| RMSProp | **V:** $V(\boldsymbol{M}) = \mathrm{diag}(\boldsymbol{M})$ 
 **S:** $S(\boldsymbol{v}) = \mathrm{Diag}(1/(\sqrt{\boldsymbol{v}} + \epsilon))$ | $\mathrm{tr}\left(\mathrm{Diag}(\boldsymbol{H})^{1/2}\right)$ | Adam with $\beta_1 = 0$. |
| Adam-mini | **V:** $V(\boldsymbol{M})_i = \frac{1}{|B_k|} \sum_{j \in B_k} M_{jj}$ for all $i \in B_k$ 
 **S:** $S(\boldsymbol{v}) = \mathrm{Diag}(1/(\sqrt{\boldsymbol{v}} + \epsilon))$ | $\sum_{i \in [N]} \sqrt{|B_i| \cdot \mathrm{tr}\left(\boldsymbol{H}_{B_i}\right)}$ | Parameters are partitioned into blocks $\{B_1, B_2, \cdots, B_N\}$. |
| Adalayer | **V:** $V(\boldsymbol{M})_i = \frac{1}{|L_k|} \sum_{j \in L_k} M_{jj}$ for all $i \in L_k$ 
 **S:** $S(\boldsymbol{v}) = \mathrm{Diag}(1/(\sqrt{\boldsymbol{v}} + \epsilon))$ | $\sum_{i \in [N]} \sqrt{|L_i| \cdot \mathrm{tr}\left(\boldsymbol{H}_{L_i}\right)}$ | Parameters are grouped by layers $\{L_1, L_2, \cdots, L_N\}$. |
| AdamE-$\lambda$ | **V:** $V(\boldsymbol{M}) = \mathrm{diag}(\boldsymbol{M})$ 
 **S:** $S(\boldsymbol{v}) = \mathrm{Diag}(1/(\boldsymbol{v}^{\odot\lambda} + \epsilon))$ | $\mathrm{tr}\left(\mathrm{Diag}(\boldsymbol{H})^{1-\lambda}\right)$ | See Section B.1 for discussion. |
| Shampoo | **V:** $V(\boldsymbol{M}) = (V_L(\boldsymbol{M}), V_R(\boldsymbol{M}))$ 
 $[V_L(\boldsymbol{M})]_{i,j} = \sum_k M_{i,k,k\cdot j}$, 
 $[V_R(\boldsymbol{M})]_{i,j} = \sum_k M_{k,i,j,k}$ 
 **S:** $S(\boldsymbol{V}_L, \boldsymbol{V}_R) = ((\boldsymbol{V}_R + \epsilon\boldsymbol{I})^\top \otimes (\boldsymbol{V}_L + \epsilon\boldsymbol{I}))^{-1/2}$ | No explicit form | For a single matrix parameter. See Appendix J for discussion. |

**Effect of the Preconditioner on the Gradient Noise Covariance.** Near the manifold, as the gradient of loss vanishes ($\nabla\mathcal{L}(\boldsymbol{\theta}) \to 0$), SGD's wandering around becomes noise-driven. For AGMs, the situation is more subtle.

First, one can show that the momentum term does not affect the implicit bias, consistent with prior theory [Wang et al., 2023]. The reason why $\beta_1$ does not affect the implicit bias is that, after the iteration approaches the manifold, the difference between the current gradient $g_t$ and momentum $M_t$ becomes negligible in expectation.

Second, the AGM trajectory is influenced by its preconditioner. Concretely, the gradient-noise covariance matrix $\boldsymbol{\Sigma}$ is filtered through the preconditioner $\boldsymbol{S}(t)$ into $\boldsymbol{S}(t)\boldsymbol{\Sigma}\boldsymbol{S}(t)$ and then contributes to the SDE. Over a long time horizon, this modified noise term alters the deterministic drift direction, further distinguishing AGM's dynamics from those of vanilla SGD.

## 4.3 Technical Difficulties and Proof Insights

### 4.3.1 Convergence Guarantee of AGMs

The core of our study is to consider the behavior of Adam's implicit bias around the minimizer manifold. However, to make our study self-contained, we first need to show that Adam can actually converge to the neighborhood of the minimizer manifold under our setting. This is non-trivial since Adam cannot provably converge to the minimizer manifold without any constraints. In fact, the convergence issue of Adam has been debated from its birth. Reddi et al. [2018] show that Adam does not converge to the optimal solution even in some simple convex settings. Recent work [Dereich and Jentzen, 2024] gives Adam's ODE and shows that this ODE does not necessarily converge to the absorbing point of the gradient flow.

The magnitude of $1 - \beta_2$ has been found to be an important factor influencing whether Adam converges. When $1 - \beta_2$ is too large such that $\beta_2 < \beta_1^2$, Adam may not converge at all [Reddi et al., 2018, Zhang et al., 2024b]. In our analysis, we assume that $1 - \beta_2 = \Theta(\eta^2)$, i.e., $1 - \beta_2$ goes to zero as $\eta \to 0$ with a rate of $\eta^2$, a configuration we call the 2-*scheme*.

As stated in Section 2, there have been works proving the convergence of Adam using various kinds of assumptions and giving different forms of bounds. However, previous results cannot be simply applied in our analysis. Specifically, our analysis near manifold requires a high-probability bound on the optimality gap directly, and we have the 2-scheme assumption. Directly applying previous results in our setting yields bounds that are either loose or hold only in expectation, failing to meet

the requirements of our analysis. Moreover, we expect a bound that holds for all AGMs, instead of only Adam.

Therefore, we present the following statement of AGMs' convergence as a preparation for our subsequent study into their behavior near the manifold.

**Theorem 4.2** (Convergence Bound of AGMs, Stated Informally). *Let Assumptions 3.2, 3.3 and 4.2 hold, $1 - \beta_2 = \Theta(\eta^2)$, and $\mathcal{L}$ satisfy the Polyak-Łojasiewicz condition. With a small learning rate $\eta$, it holds with high probability for some $K = \mathcal{O}(\frac{1}{\eta} \log \frac{1}{\eta})$ that $\mathcal{L}(\boldsymbol{\theta}_K) - \mathcal{L}^* = \tilde{\mathcal{O}}(\eta)$. See Theorem D.2 for a formal statement.*

### 4.3.2 Key Insights in the Derivation of Slow SDEs for AGMs

After the AGMs reach the neighborhood of the minimizer manifold, we can derive an analysis similar to the one in the local SGD paper [Gu et al., 2023a]. Specifically, we use SDEs to approximate the AGMs after they reach the manifold neighborhood. However, unlike the usual SDE approximation, the SDEs we use here can track the AGMs for a much longer period of time, up to $\mathcal{O}(\eta^{-2})$ rather than the $\tilde{\mathcal{O}}(\eta^{-1})$, which is more common in the previous papers. This type of SDE is termed "*slow SDE*" by Gu et al. [2023a].

There are two obstacles preventing us from directly applying the analysis of slow SDEs from SGDs to AGMs. First, the obtaining of slow SDEs requires an accurate calculation of the variation of the first-order and second-order moments of the parameters over a relatively large number of steps (a "*giant step*" in the notation of Gu et al. [2023a]), and in the case of SGD, due to the nature of its rotational equivariance, we can always consider its Hessian matrix as a diagonal array, as well as its corresponding minimizer manifold as a space extended by some full-space standard bases, which greatly simplifies the computation. However, it is not the case for AGMs. Due to the effect of preconditioners $S(\boldsymbol{v}_k)$, the rotation equivariance is not satisfied here.

To resolve this, we generalize the gradient flow projection in Gu et al. [2023a], Li et al. [2021b] into a varying preconditioner flow projection. Based on this definition, reparameterizing to the original space lets us reuse the simple formulas employed previously [Gu et al., 2023a, Li et al., 2021b].

The second reason is that when $\beta_2$ is far from 1, the preconditioner changes too quickly, making the evolution of the moments hard to characterize. Conversely, when $\beta_2$ is extremely close to 1, the preconditioner changes so little as to be impractical. Accordingly, we focus on the 2-scheme: $1 - \beta_2 = \mathcal{O}(\eta^2)$. The key point is that this regime does not make the preconditioner's evolution negligible; rather, its slow but nontrivial drift shapes the SDE and can be tracked analytically.

## 5 Conclusions

In this work, we have shown that Adam implicitly minimizes a distinctive sharpness measure $\text{tr}(\text{Diag}(\boldsymbol{H})^{1/2})$, and that this bias leads to different solutions and generalization behavior compared to SGD. Our slow SDE framework not only rigorously characterizes Adam's adaptive semi-gradient drift near the minimizer manifold, but also recovers concrete separations in sparse linear regression and deep matrix factorization settings.

Despite these advances, several important avenues remain open. First, we have focused on the "2-scheme" regime (where $1 - \beta_2 = O(\eta^2)$) in order to track Adam's preconditioner over a long timescale; extending our analysis to the intermediate 1.5-scheme or other scalings of $1 - \beta_2$ is left for future work. Second, our derivations assume that the iterates remain close to a smooth minimizer manifold; understanding Adam's implicit bias once the trajectory ventures beyond this local neighborhood may require restarting the analysis from the SGD dynamics. Finally, our approach cannot cover weight-decay or decoupled decay terms such as the $W$-term in AdamW; characterizing how weight decay alters the effective sharpness regularizer is an important direction for follow-on studies.

## 6 Acknowledgement

We would like to thank Xinran Gu, Kaiyue Wen, Yushun Zhang, Jiaye Teng, Yuhang Cai, and the anonymous NeurIPS reviewers for their insightful comments and feedback.

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

# Contents

## A  Illustration of the Difference between Conventional SDE and Slow SDE

In this section, we illustrate the difference between conventional SDE and slow SDE. In Figure 2, let $\Gamma$ denotes a 1D manifold, then the discrete iteration of the optimization process can be seen as successive steps (orange, Figure 2a) that starts from $A$, first converge to some point $B$ in $\Gamma$ and then move along $\Gamma$ to $C$.

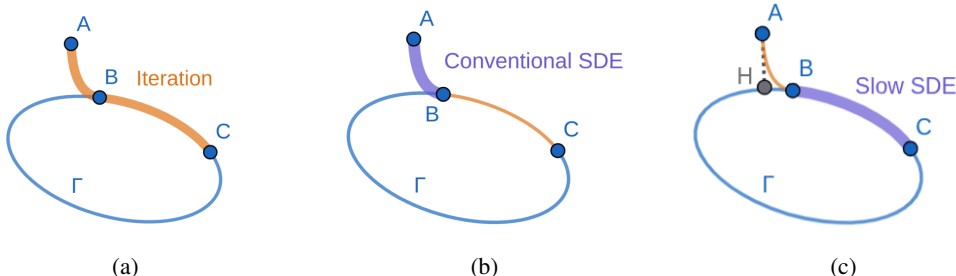

(a)                          (b)                          (c)

Figure 2: Comparison of conventional SDE and slow SDE.

The main intuition behind slow SDE is that the whole process $A \to B \to C$ can actually be decomposed into two motions: a convergence motion $A \to H$ (dashed, Figure 2c) and an implicit regularization motion $H \to B \to C$. The convergence motion is fast and dominates the dymanics during the convergence phase, but it fades out as soon as convergence phase ends; meanwhile the slow, implicit regularization motion starts to dominate.

The conventional SDE approximates the convergence phase only, whose unit time corresponds to $\tilde{O}(\eta^{-1})$ steps (Figure 2b). In contrast, slow SDE manages to separate the slow implicit regularization motion from the fast convergence, and approximate the implicit regularization near manifold only (Figure 2c).

**Remark A.1.** *The projection method (which projects $A \to B \to C$ to $H \to B \to C$) varies in the analysis of different optimizers. Intuitively, the projection should reflect the converging direction driven by a clean (without noise) and continuous version of the optimizer. In SGD the projection is gradient flow; but in Adam we need to consider the preconditioning effect caused by $1/\left(\sqrt{v} + \epsilon\right)$, so we add an SDE to track the preconditioner, and define a preconditioned gradient flow for projection.*

## B  Adam's Provable Generalization Benefit with Label Noise

In this section, we prove that with the label noise condition, the implicit regularizer of Adam reduces to a simpler form that aligns better with sparsity regularizations, and then verify experimentally.

### B.1  Implicit Regularizer of Adam under Label Noise

**Label Noise.**  By *label noise* we refer to the condition that for all $\boldsymbol{\theta} \in \Gamma$, the covariance matrix $\boldsymbol{\Sigma}$ is a constant multiple of the Hessian: $\boldsymbol{\Sigma}(\boldsymbol{\theta}) = \alpha \nabla^2 \mathcal{L}(\boldsymbol{\theta})$ for some constant $\alpha$ [Blanc et al., 2020]. This condition typically arises when training a model that is overparameterized enough to fit the training data perfectly, while fresh noise is added to the target label at each step of training.

To see how this label noise condition holds, imagine a simple regression problem with training data $\{(\boldsymbol{x}^{(i)}, y^{(i)})\}_{i=1}^{n}$ and model $h(\boldsymbol{\theta}; \boldsymbol{x})$. The loss function for a single data sample is defined as $\ell(\boldsymbol{\theta}; \boldsymbol{x}, y) = \frac{1}{2}(h(\boldsymbol{\theta}; \boldsymbol{x}) - y)^2$. With label noise, at each step $t$ of training, we randomly take a random data sample $(\boldsymbol{x}_t, y_t)$ as well as an independent noise perturbation $\zeta_t$ from a distribution with zero mean and constant variance $\delta > 0$. Then we take one gradient step on $\ell(\boldsymbol{\theta}; \boldsymbol{x}_t, y_t + \zeta_t)$ with the noisy label $y_t + \zeta_t$.

The expected training loss becomes $\mathcal{L}(\boldsymbol{\theta}) = \mathbb{E}[\ell(\boldsymbol{\theta}; \boldsymbol{x}_t, y_t + \zeta_t)]$. When the model is sufficiently overparameterized so that $h(\boldsymbol{\theta}; \boldsymbol{x}) = y$ can be simultaneously attained for all training data points in its parameter space, the minimizer manifold is just $\Gamma = \left\{\boldsymbol{\theta} : \mathcal{L}(\boldsymbol{\theta}) = \frac{1}{2}\delta^2\right\}$, where $\frac{1}{2}\delta^2$ is entirely due to the presence of label noise. On this manifold, we can see how the label noise condition

$\boldsymbol{\Sigma}(\boldsymbol{\theta}) = \alpha \nabla^2 \mathcal{L}(\boldsymbol{\theta})$ holds for $\alpha = \delta^2$:

$$\boldsymbol{\Sigma}(\boldsymbol{\theta}) := \mathbb{E}\left[\nabla \ell(\boldsymbol{\theta}; \boldsymbol{x}_t, y_t + \zeta_t)\nabla \ell(\boldsymbol{\theta}; \boldsymbol{x}_t, y_t + \zeta_t)^\top\right] = \mathbb{E}\left[\zeta_t^2 \nabla h(\boldsymbol{\theta}; \boldsymbol{x})\nabla h(\boldsymbol{\theta}; \boldsymbol{x})^\top\right] = \delta^2 \nabla \mathcal{L}(\boldsymbol{\theta}).$$

Beyond this simple setting, a similar analysis can be applied to establish the label noise condition $\boldsymbol{\Sigma}(\boldsymbol{\theta}) = \alpha \nabla^2 \mathcal{L}(\boldsymbol{\theta})$ for mini-batch training and other common losses such as cross-entropy loss. This proportional relationship between Hessian and noise covariance greatly simplifies the analysis of training dynamics and has been widely used to study the implicit bias of SGD and related optimizers [Blanc et al., 2020, Damian et al., 2021, Li et al., 2021b, Gu et al., 2023a].

Under the label noise condition, Li et al. [2021b] proved that slow SDE for SGD reduces to an ODE. In the following theorem, we show that the slow SDE for AGM also reduces to an ODE, but the resulting ODE takes a different form:

**Theorem B.1** (Slow ODE for AGMs with Label Noise). *Under the label noise condition, the Slow SDE for AGMs in Equation* (2) *becomes the following ODE:*

$$\begin{cases} \mathrm{d}\boldsymbol{\zeta}(t) = -\frac{\alpha}{2}\boldsymbol{S}_t \partial \Phi_{\boldsymbol{S}_t}(\boldsymbol{\zeta})\boldsymbol{S}_t \nabla^3 \mathcal{L}(\boldsymbol{\zeta})[\boldsymbol{S}_t]\mathrm{d}t, \\ \mathrm{d}\boldsymbol{v}(t) = c\left(V(\boldsymbol{\Sigma}(\boldsymbol{\zeta})) - \boldsymbol{v}\right)\mathrm{d}t, \end{cases} \quad (3)$$

*where* $\boldsymbol{S}_t := \boldsymbol{S}(\boldsymbol{v}(t))$.

See Appendix H for the proof. We then derive the implicit bias of Adam with label noise.

**Lemma B.1** (Adam's Implicit Bias under Label Noise). *Under the label noise condition, every fixed point of Equation* (3) *for Adam satisfies* $\nabla_\Gamma tr\left(\mathrm{Diag}(\boldsymbol{H})^{1/2}\right) = 0$.

**Proof Sketch.** At the fixed point of Equation (3), we have

$$\boldsymbol{v} = V(\boldsymbol{\Sigma}(\boldsymbol{\zeta})), \quad S(\boldsymbol{v})\partial\Phi_{S(\boldsymbol{v})}(\boldsymbol{\zeta})S(\boldsymbol{v})\nabla^3\mathcal{L}(\boldsymbol{\zeta})[S(\boldsymbol{v})] = 0.$$

With label noise, $\boldsymbol{H} \stackrel{\text{def}}{=} \nabla^2 \mathcal{L}(\boldsymbol{\zeta}) = \boldsymbol{\Sigma}(\boldsymbol{\zeta})/\alpha$. In the Adam case, $\boldsymbol{v} = \mathrm{diag}(\boldsymbol{\Sigma}) = \alpha\mathrm{diag}(\boldsymbol{H})$, and $S(\boldsymbol{v}) = \mathrm{Diag}(1/\sqrt{\boldsymbol{v}})$ with $\epsilon = 0$. Then $S(\boldsymbol{v}) = \mathrm{Diag}(\frac{1}{(\alpha\mathrm{diag}(\boldsymbol{H}))^{1/2}})$, and we can simplify the regularizer term into

$$\nabla^3\mathcal{L}(\boldsymbol{\zeta})[S(\boldsymbol{v})] = \sum_{j=1}^d \frac{1}{(\alpha H_{jj})^{1/2}}\nabla(H_{jj}) = \frac{2}{\sqrt{\alpha}}\sum_{j=1}^d \nabla((H_{jj})^{1/2}) = \frac{2}{\sqrt{\alpha}}\nabla\mathrm{tr}\left(\mathrm{Diag}(\boldsymbol{H})^{1/2}\right).$$

which implies that the stationary point of Adam satisfies

$$S(\boldsymbol{v})\partial\Phi_{S(\boldsymbol{v})}(\boldsymbol{\zeta})S(\boldsymbol{v})\nabla\mathrm{tr}\left(\mathrm{Diag}(\boldsymbol{H})^{1/2}\right) = 0.$$

The preceding matrices have some clean properties. From Lemma G.2, $\partial\Phi_{S(\boldsymbol{v})}(\boldsymbol{\zeta})S(\boldsymbol{v})$ acts as an invertible linear map on the tangent space of the manifold $\Gamma$, and vanishes on its normal space. Together with the invertibility of $S(\boldsymbol{v})$, this gives

$$\nabla_\Gamma\mathrm{tr}\left(\mathrm{Diag}(\boldsymbol{H})^{1/2}\right) = 0$$

for any fixed point of Adam's slow ODE. See Appendix H for more details. $\square$

The above proof can be generalized to the case of $\epsilon > 0$. When $\epsilon > 0$, the implicit bias of Adam changes to

$$\nabla_\Gamma\mathrm{tr}\left(\mathrm{Diag}(\boldsymbol{H}^*)^{1/2} - \frac{\epsilon}{\sqrt{\alpha}}\ln\left(\frac{\sqrt{\alpha}}{\epsilon}\mathrm{Diag}(\boldsymbol{H}^*)^{1/2} + \boldsymbol{I}\right)\right) = \boldsymbol{0}.$$

See Lemma H.2 for more details. Note that this trace term is always non-negative, since $x - \frac{\epsilon}{\sqrt{\alpha}}\ln(\frac{\sqrt{\alpha}}{\epsilon}x + 1) \geq x - \frac{\epsilon}{\sqrt{\alpha}} \cdot \frac{\sqrt{\alpha}}{\epsilon}x = 0$ for any $x \geq 0$.

**A Simple Way to Tune Adam's Implicit Bias: AdamE.** The proof of Lemma B.1 inspires the following simple variant of Adam: We define *AdamE* as an optimizer class that, is identical to Adam except that $S(\boldsymbol{v}) = \mathrm{Diag}(1/(\boldsymbol{v}^{\odot\lambda} + \epsilon))$ for a tunable parameter $\lambda \in [0, 1)$. For any $\lambda_0 \in [0, 1)$ we also use the term *AdamE with* $\lambda = \lambda_0$, or simply *AdamE-$\lambda_0$*. Note that AdamE with $\lambda = \frac{1}{2}$ coincides with Adam, and that all AdamE optimizers lie within the AGM framework. Applying the same method as in Lemma B.1 yields the implicit bias of AdamE under label noise.

**Lemma B.2** (AdamE's Implicit Bias with Label Noise). *Under the label noise condition, the fixed point of Equation (3) for AdamE-$\lambda$ with $\lambda \in [0, 1)$ and $\epsilon = 0$ satisfies $\nabla_\Gamma tr \left( \mathrm{Diag}(\boldsymbol{H})^{1-\lambda} \right) = 0$.*

Lemma B.2 indicates that tuning the exponent of the second-order moment in Adam exactly results in tuning the exponent of $\mathrm{diag}(\nabla^2 \mathcal{L}(\boldsymbol{\zeta}))$ in the implicit bias. When $\lambda = 0$, the implicit bias reduces to that of SGD, and AdamE also gets rid of the effect of second-order moments and reduces to SGD with momentum, which coincides perfectly. Next, we relate the implicit bias to sparsity and compare the performance of Adam, AdamE, and SGD in a simple experimental setup.

## B.2 Example: Sparse Linear Regression with Diagonal Net

In this section, we adopt the *diagonal linear network* (diagonal net) setting proposed by Woodworth et al. [2020] as an experimental setting, which is also used by Li et al. [2021b] to study the implicit bias of SGD.

**Setting (Diagonal Net with Label Noise):** Let $\boldsymbol{w}^* \in \mathbb{R}^d$ be an unknown $\kappa$-sparse ground truth vector. Let $\{(\boldsymbol{z}_i, y_i)\}_{i \in [n]}$ be the training dataset where each $\boldsymbol{z}_i \overset{\text{i.i.d.}}{\sim} \mathrm{Unif} \{\pm 1\}^d$, and each $y_i$ is generated by $\langle \boldsymbol{z}_i, \boldsymbol{w}^* \rangle$. Our parameter is defined as $\boldsymbol{\theta} = \begin{pmatrix} \boldsymbol{u} \\ \boldsymbol{v} \end{pmatrix} \in \mathbb{R}^{2d}$. For any function $g$ defined on $\mathbb{R}^{2d}$, we write $g(\boldsymbol{\theta})$ and $g(\boldsymbol{u}, \boldsymbol{v})$ interchangeably. The loss function is defined as:

$$\mathcal{L}(\boldsymbol{\theta}) = \frac{1}{n} \sum_{i=1}^{n} \mathcal{L}_i(\boldsymbol{\theta}), \quad \text{where } \mathcal{L}_i(\boldsymbol{\theta}) = \frac{1}{2} \left( \langle \boldsymbol{z}_i, \boldsymbol{u}^{\odot 2} - \boldsymbol{v}^{\odot 2} \rangle - y_i \right)^2$$

where a label noise is added to the true label $y$ during training. This setting can be viewed as using estimation $\widehat{\boldsymbol{w}} = \boldsymbol{u}^{\odot 2} - \boldsymbol{v}^{\odot 2}$ to approximate the ground truth vector $\boldsymbol{w}^*$ of a linear regression task. Note that $d \gg n$ here so the model is highly overparameterized: Theoretically, Li et al. [2021b] proved that $n = \mathcal{O}(\kappa \ln d)$ is enough for SGD to recover ground truth, and we will later show experimentally that less than 1000 training pairs is required for both Adam and SGD to achieve a low test loss when $d = 10000$. The manifold is defined as wherever zero train loss is achieved, i.e. $\Gamma = \left\{ \boldsymbol{\theta} | \langle \boldsymbol{z}_i, \boldsymbol{u}^{\odot 2} - \boldsymbol{v}^{\odot 2} \rangle = y_i, \forall i \in [n] \right\}$.

This setting allows us to relate the implicit bias directly to the sparsity of the estimated ground truth.

**Lemma B.3.** *Let $\boldsymbol{\theta}^*$ be an optimal parameter minimizing the loss function $\mathcal{L}$, i.e. $\boldsymbol{\theta}^* \in \Gamma$. For each $\boldsymbol{\theta} = \begin{pmatrix} \boldsymbol{u} \\ \boldsymbol{v} \end{pmatrix} \in \Gamma$, denote $\widehat{\boldsymbol{w}} := \boldsymbol{u}^{\odot 2} - \boldsymbol{v}^{\odot 2}$ and $\boldsymbol{H} := \nabla^2 \mathcal{L}(\boldsymbol{\theta})$. We have the following:*

- *If $\boldsymbol{\theta}^* \in \arg\min_{\boldsymbol{\theta} \in \Gamma} tr(\mathrm{Diag}(\boldsymbol{H})^{0.5})$, then we also have $\boldsymbol{\theta}^* \in \arg\min_{\boldsymbol{\theta} \in \Gamma} \|\widehat{\boldsymbol{w}}\|_{0.5}$.*

- *Furthermore, for any $e_0 \in (0, 1]$, if $\boldsymbol{\theta}^* \in \arg\min_{\boldsymbol{\theta} \in \Gamma} tr(\mathrm{Diag}(\boldsymbol{H})^{e_0})$, then we also have $\boldsymbol{\theta}^* \in \arg\min_{\boldsymbol{\theta} \in \Gamma} \|\widehat{\boldsymbol{w}}\|_{e_0}$.*

The main idea of the proof is that the training loss depends only on the combined quantity $\widehat{\boldsymbol{w}} = \boldsymbol{u}^{\odot 2} - \boldsymbol{v}^{\odot 2}$. Hence, if for some index $i$ both $u_i$ and $v_i$ are nonzero, we can reduce the magnitudes of $u_i$ and $v_i$ while keeping $u_i^2 - v_i^2$ fixed, obtaining another minimizer with strictly smaller $tr(\mathrm{Diag}(\boldsymbol{H})^{e_0})$. Therefore, at any optimum we must have $u_i = 0$ or $v_i = 0$ for every $i$. Under this condition, $tr(\mathrm{Diag}(\boldsymbol{H})^{e_0})$ can be identified with $\|\widehat{\boldsymbol{w}}\|_{e_0}$. We provide the detailed derivation in Appendix H.

Lemma B.3 gives the following insights: Implicitly regularizing $tr(\mathrm{Diag}(\boldsymbol{H})^{e_0})$ is equivalent to regularizing the $\ell_{e_0}$-norm of the estimated ground truth $\widehat{\boldsymbol{w}} = \boldsymbol{u}^{\odot 2} - \boldsymbol{v}^{\odot 2}$: Adam corresponds to $\ell_{0.5}$, SGD to $\ell_1$, and AdamE-$\lambda$ to $\ell_{1-\lambda}$. Just as lasso ($\ell_1$) is preferable to ridge ($\ell_2$) for sparse ground-truth recovery, we therefore expect Adam and AdamE (with $\lambda > 0$) to recover sparse ground truth more efficiently than SGD. We verify this prediction below.

### B.2.1 Result: Adam's Implicit Regularizer Facilitates Sparse Ground-truth Recovery

Figure 3 shows the results of the experiment. We gradually increase the number of training points and train Adam, SGD, and AdamE under several configurations until convergence. We consider a configuration to have *recovered the ground truth* if the test loss falls below 1. As illustrated in Figure 3a, Adam's test loss plunges towards zero at approximately $n_{\text{train}} = 420$, whereas SGD's test loss decreases more gradually as the training set grows. To interpolate between different implicit biases, we evaluate AdamE for several values of $\lambda$. Figure 3b indicates that AdamE, even with a small positive value of $\lambda$, exhibits the same sudden recovery behavior as Adam. This suggests that Adam's influence on the implicit bias upon SGD arises from its preconditioning mechanism.

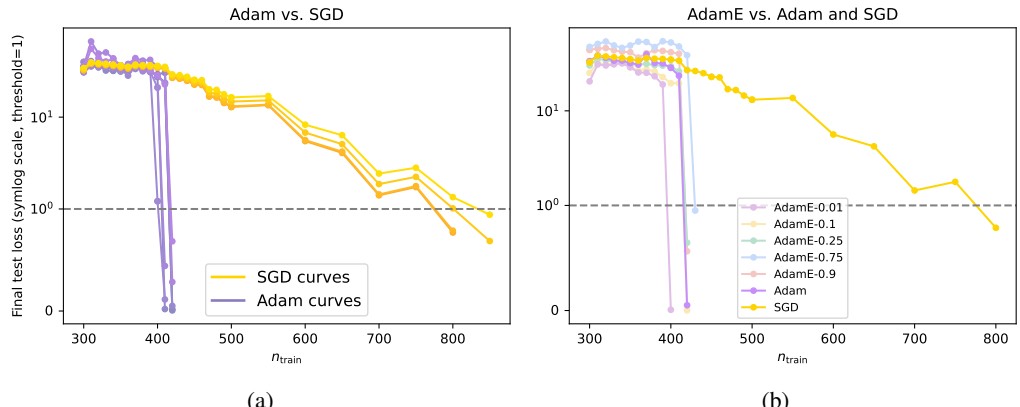

(a)                          (b)

Figure 3: Final test loss as a function of the training data size with $d = 10000$, $\kappa = 50$. Each plotted point is the final test loss after both the training and test losses have converged; its x-coordinate is the training data size and the curve denotes the optimizer and configuration. **(a)** Loss comparison between SGD with different learning rates, and Adam with different learning rates and $\beta_2$ values. **(b)** Loss comparison between AdamE with $\lambda = 0.01, 0.1, 0.25, 0.75, 0.9$, Adam, and SGD.

**Takeaway.** Adam's unique implicit bias may help to recover a sparse ground truth.

However, we should also keep in mind that a clear interpretation of Adam's unique implicit bias, $\mathrm{tr}(\mathrm{Diag}(\boldsymbol{H})^{1/2})$ relies heavily on the condition that $\boldsymbol{H}$ is diagonal. Only with this condition can we claim Adam as minimizing $\|\boldsymbol{h}\|_{0.5}$ instead of SGD's $\|\boldsymbol{h}\|_1$ where $\boldsymbol{h}$ is the vector consisting of all eigenvalues of $\boldsymbol{H}$. In other words, Adam's optimization on the implicit bias upon SGD only makes sense when $\boldsymbol{H}$ is diagonal. In the diagonal net setting this is indeed the case in expectation, but we will see in the next chapter that Adam's unique implicit bias may even lead to worse generalization when $\boldsymbol{H}$ is no longer diagonal.

## C  Matrix Factorization: Adam Implicitly Regularizes Sharpness Differently

The diagonal net experiments in Section B showed that Adam's implicit bias towards *sparsity* improves generalization relative to SGD. We now turn to supply the potentially negative impact of Adam's implicit bias in another controlled setting: **deep matrix factorization with label noise**, where the relevant implicit regularizers are analytically tractable.

In this task, Adam is expected to minimize $\mathrm{tr}(\mathrm{Diag}(\boldsymbol{H})^{1/2})$ rather than $\mathrm{tr}(\boldsymbol{H})$. Leveraging existing theory, we therefore predict that (i) Adam will converge to a solution with larger $\mathrm{tr}(\boldsymbol{H})$, whose $\mathrm{tr}(\mathrm{Diag}(\boldsymbol{H})^{1/2})$, however, smaller than SGD's solution, and (ii) once Adam reaches a solution with larger $\mathrm{tr}(\boldsymbol{H})$, supported by Gatmiry et al. [2023], it will *generalize worse* than vanilla SGD in the presence of label noise. Our experiments confirm both predictions (Figure 4 and Figure 5).

### C.1  Problem setup

We consider an $L$-layer linear network with parameters $\boldsymbol{W} = (\boldsymbol{W}_1, \ldots, \boldsymbol{W}_L)$, where $\boldsymbol{W}_i \in \mathbb{R}^{d_i \times d_{i-1}}$ and $d_i \geq \min\{d_0, d_L\}$ for all $i$. We let $\boldsymbol{M}^* \in \mathbb{R}^{d_L \times d_0}$ be a rank-$r$ ground truth matrix and observe the $n$ i.i.d. linear measurements $\{(\boldsymbol{A}_i, b_i)\}_{i=1}^n$ generated by $b_i = \langle \boldsymbol{A}_i, \boldsymbol{M}^* \rangle$. With label noise and mini-batch size $B$, the empirical loss at step $t$ is:

$$\mathcal{L}_t(\boldsymbol{W}) = \frac{1}{B} \sum_{i \in \mathcal{B}_t} \big( \langle \boldsymbol{A}_i, \boldsymbol{W}_L \cdots \boldsymbol{W}_1 \rangle - b_i + \xi_{t,i} \big)^2,$$

where $\mathcal{B}_t$ stands for a fresh batch of size $B$, and the label noise $\xi_{t,i} \sim \mathcal{N}(0, \sigma^2)$ are independent to each other for both different $t$ and $i$.

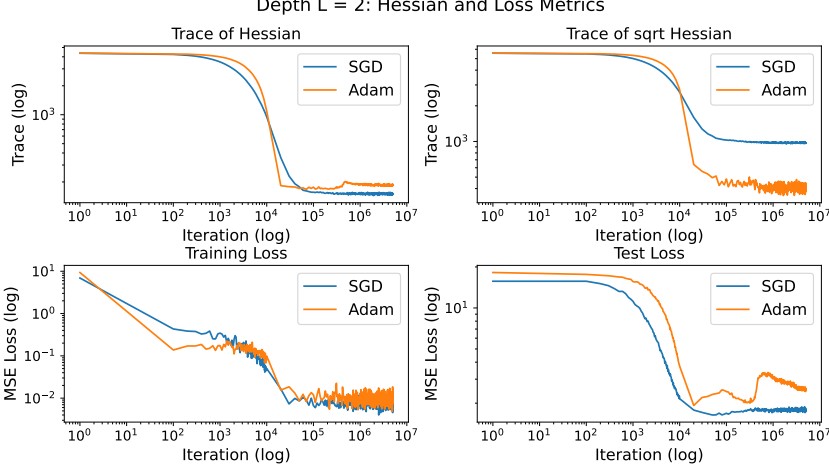

Figure 4: **Deep matrix factorization with label noise with deepth** $L = 2$. Adam and SGD are trained on identical data and noise realizations. *Top:* evolution of $\mathrm{tr}(\boldsymbol{H})$ and $\mathrm{tr}(\mathrm{Diag}(\boldsymbol{H})^{1/2})$. *Bottom:* training and test MSE. Adam converges to a point with larger $\mathrm{tr}(\boldsymbol{H})$ but smaller $\mathrm{tr}(\mathrm{Diag}(\boldsymbol{H})^{1/2})$, and exhibits higher test error.

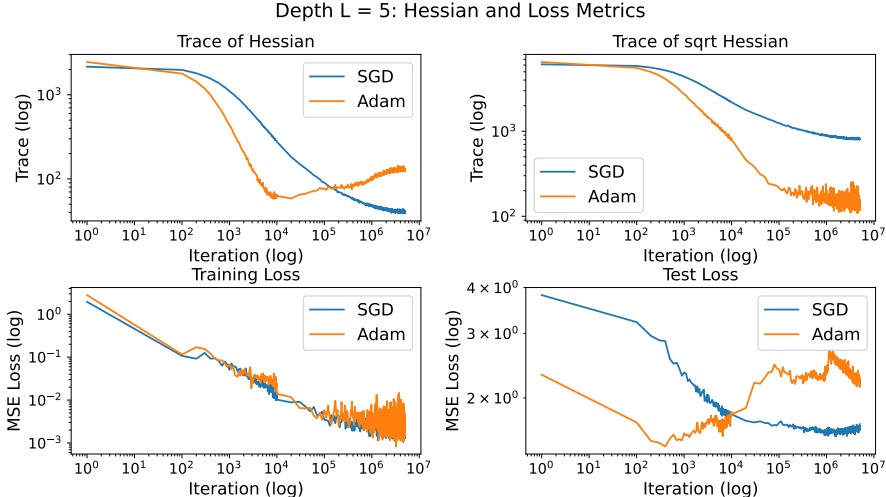

Figure 5: **Deep matrix factorization with label noise with deepth** $L = 5$.

## C.2 Results

Our setups for SGD optimizer follows Section 7 of Gatmiry et al. [2023]. For Adam, we use the standard hyperparameters $\beta_1 = 0.9$, $\beta_2 = 0.999$, and learning rate $10^{-3}$; all other settings are identical to SGD.

Figure 4 and Figure 5 (top rows in the figures) show the evolution of sharpness metrics and the training/test losses for layer depth $L = 2, 5$. Adam drives $\mathrm{tr}(\mathrm{Diag}(\boldsymbol{H})^{1/2})$ sharply downward while $\mathrm{tr}(\boldsymbol{H})$ remains high and even non-monotone, confirming that Adam does *not* target Hessian trace. Correspondingly, the bottom rows in these two figures show that Adam attains a higher test loss despite similar training error, which is an evidence that Adam's implicit bias is detrimental in this setting.

Recall that SGD with label noise implicitly regularizes $\mathrm{tr}(\boldsymbol{H})$ on the minimizer manifold. On this matrix factorization task, Gatmiry et al. [2023] proved that minimizing $\mathrm{tr}(\boldsymbol{H})$ is roughly equivalent to minimizing the nuclear norm of $\boldsymbol{W}^*$, which is connected to generalization improvement when $\boldsymbol{M}^*$

possesses the low-rank property. In contrast, our theoretical analysis suggests that Adam implicitly regularizes $\mathrm{tr}(\mathrm{diag}(\boldsymbol{H})^{1/2})$ rather than $\mathrm{tr}(\boldsymbol{H})$, inducing a different bias that does not necessarily favor low-rank solutions. Consequently, Adam is expected to converge to solutions with smaller $\mathrm{tr}(\mathrm{diag}(\boldsymbol{H})^{1/2})$, larger $\mathrm{tr}(\boldsymbol{H})$ and worse generalization compared to SGD.

**Takeaway.** In deep matrix factorization with label noise, Adam's tendency to minimize $\mathrm{tr}(\mathrm{Diag}(\boldsymbol{H})^{1/2})$ drives it towards solutions different from those found by SGD and exhibits worse recovery of the low-rank ground truth. This highlights that Adam's implicit regularization can be detrimental for certain tasks.

# D    Restatements of the Main Results

In this section, to make our results more organized and easy to understand, we give the restatements of the main results in Section 4, where we present the two main theorems.

1. The AGM iterates converge to a neighborhood of the manifold (Theorem 4.2);

2. Moreover, once the iterates enter this neighborhood, their dynamics over $\mathcal{O}(\eta^{-2})$ discrete steps can be accurately tracked by a slow SDE (Theorem 4.1).

Recall that in the AGM framework, the transition from $\boldsymbol{\theta}_k$ to $\boldsymbol{\theta}_{k+1}$ is defined as:

$$\boldsymbol{m}_{k+1} := \beta_1 \boldsymbol{m}_k + (1 - \beta_1)\nabla \ell_k(\boldsymbol{\theta}_k)$$
$$\boldsymbol{v}_{k+1} := \beta_2 \boldsymbol{v}_k + (1 - \beta_2)V\big(\nabla \ell_k(\boldsymbol{\theta}_k)\nabla \ell_k(\boldsymbol{\theta}_k)^\top\big)$$
$$\boldsymbol{\theta}_{k+1} := \boldsymbol{\theta}_k - \eta S(\boldsymbol{v}_{k+1})\boldsymbol{m}_{k+1},$$

under the following conditions:

1. $S : \mathbb{R}^D_{\geq 0} \longrightarrow \mathbb{S}^d_{++}$ is $\rho_s$-smooth, where $\mathbb{R}^D_{\geq 0}$ denotes the subset of $\mathbb{R}^D$ that has non-negative entries, and $\mathbb{S}^d_{++}$ denotes the space of $\mathbb{R}^{d \times d}$ positive definite matrices.

2. $S(\boldsymbol{v}) \succeq \frac{1}{R_0}\boldsymbol{I}$ for some $R_0 > 0$ and any $\boldsymbol{v} \in \mathbb{R}^D_{\geq 0}$.

3. $V : \mathbb{R}^{d \times d} \longrightarrow \mathbb{R}^D$ is linear, and $V(\boldsymbol{g}\boldsymbol{g}^\top) \in \mathbb{R}^D_{\geq 0}$ for all $\boldsymbol{g} \in \mathbb{R}^d$.

## D.1    Slow SDE for AGMs

**Theorem D.1.** *Let Assumptions 3.2, 3.3 and 4.2 be satisfied. Let $\Gamma$ denote a local minimizer manifold, $\eta$ be a sufficiently small learning rate of an AGM, and $1 - \beta_2 = \Theta(\eta^2)$. Then we have the following conclusions:*

1. *(Convergence to a near–manifold neighborhood) There exists a constant $\epsilon_0 > 0$ such that for any initial point $\boldsymbol{\theta}_0$ whose $\ell_2$ distance from $\Gamma$ does not exceed $\epsilon_0$, and any $\delta \in (\eta^{200}, 1)$,[3] with probability at least $1 - \delta$, the following holds for some $K_0 = \mathcal{O}(\frac{1}{\eta}\log\frac{1}{\eta})$:*

$$\mathcal{L}(\boldsymbol{\theta}_{K_0}) - \mathcal{L}^* = \mathcal{O}\left(\eta \log \frac{1}{\eta\delta}\right),$$

$$\|\boldsymbol{\theta}_{K_0} - \Phi_{\boldsymbol{S}_{K_0}}(\boldsymbol{\theta}_{K_0})\|_2 = \mathcal{O}\left(\sqrt{\eta \log \frac{1}{\eta\delta}}\right).$$

2. *(Slow SDE tracks AGM's trajectory in a weak approximation sense) Moreover, let $T > 0$, $K := \lfloor T\eta^{-2}\rfloor$, and suppose Assumptions 3.4, 3.1 and 4.1 hold. Continue running an AGM for $K$ iterations after reaching the final state $(\boldsymbol{\theta}_{K_0}, \boldsymbol{v}_{K_0})$ in conclusion 1 and totally obtain $\{(\boldsymbol{\theta}_k, \boldsymbol{v}_k)\}_{k=0}^{K_0+K}$. Let $\boldsymbol{X}(t) = (\boldsymbol{\zeta}(t), \boldsymbol{v}(t))$ be the solution to (2) with initial condition*

$$\boldsymbol{X}(0) = \boldsymbol{X}_0 \in \Gamma \times \mathbb{R}^D_{\geq 0}, \quad \boldsymbol{X}_0 := (\Phi_{\boldsymbol{S}(\boldsymbol{v}_{K_0})}(\boldsymbol{\theta}_{K_0}), \boldsymbol{v}_{K_0}).$$

---

[3]The exponent here, along with the exponents related to the $\delta$-goodness in Definition G.1, can be arbitrary large constant, which does not affect the order of following derivations.

*For any $k \in \{0, 1, \ldots, K\}$, let $\bar{X}_k := \left( \Phi_{S(v_{K_0+k})}(\theta_{K_0+k}), v_{K_0+k} \right)$ be the AGM state at step $K_0 + k$ projected onto $\Gamma$. Then for any $C^3$ function $g(\theta)$,*

$$\max_{0 \leq k \leq K} \left| \mathbb{E}\left[ g\left(\bar{X}_k\right) \middle| X_0 \right] - \mathbb{E}\left[ g\left(X(k\eta^2)\right) \middle| X(0) = X_0 \right] \right| = \widetilde{\mathcal{O}}\left( \eta^{0.25} \right),$$

*where $\widetilde{\mathcal{O}}(\cdot)$ hides logarithmic factors and constants independent of $\eta$ (but possibly depending on $g$).*

Notice that in Theorem D.1 we give a more explicit statement for our SDE approximation results in Theorem 4.1. One can see that Theorem 4.1 is a direct corollary of Theorem D.1.

*Proof of Theorem D.1.* For the convergence part of Theorem D.1, we first construct the working zones near the manifold in Appendix E to ensure some properties that are crucial to our analysis such as the $\mu$-PL condition. After that, we give the proof of convergence in Appendix F. We finish our proof in Appendix F.1 by combining the previous results in working zone and convergence analysis.

And the proof of the SDE approximation part is finished in Appendix G.4 through the moment calculation and the subsequent weak approximation analysis in Appendix G. □

### D.2 Convergence Guarantee of AGMs

In the proof, the first part of Theorem D.1 is done by first proving a convergence result with global $\mu$-PL condition, and then arguing that AGM starting near enough to the manifold will stick to the manifold with high probability. As mentioned in Section 4.3.1, the convergence under $\mu$-PL condition can be seen as a separate technical contribution of our paper, which is stated below.

**Definition D.1** (Polyak-Łojasiewicz Condition). *For some $\mu, \bar{L} > 0$, we say some function $\mathcal{L} : \mathbb{R}^d \to \mathbb{R}$ is $(\mu, \bar{L})$-Polyak-Łojasiewicz (abbreviated as $(\mu, \bar{L})$-PL), if and only if for all $\theta \in \mathbb{R}^d$ such that $\mathcal{L}(\theta) \leq \bar{L}$:*

$$2\mu(\mathcal{L}(\theta) - \mathcal{L}^*) \leq \|\nabla\mathcal{L}(\theta)\|_2^2.$$

*When $\bar{L} = +\infty$, we call this condition the $\mu$-Polyak-Łojasiewicz ($\mu$-PL) condition.*

**Theorem D.2** (Formal restatement of Theorem 4.2). *Let Assumptions 3.2, 3.3 and 4.2 be satisfied, $\mathcal{L}$ be a function satisfying the $\mu$-PL condition, and $\eta$ be a sufficiently small learning rate of an AGM. Let $1 - \beta_2 = \Theta(\eta^2)$. For any $\delta \in (0, 1)$, with probability at least $1 - \delta$, the following holds for some $K = \mathcal{O}(\frac{1}{\eta} \log \frac{1}{\eta})$:*

$$\mathcal{L}(\theta_K) - \mathcal{L}^* = \mathcal{O}\left( \eta \log \frac{1}{\eta\delta} \right),$$

$$\|\theta_K - \Phi_{S_K}(\theta_K)\|_2 = \mathcal{O}\left( \sqrt{\eta \log \frac{1}{\eta\delta}} \right).$$

**Remark.** Note that Theorem D.2 is different from Part 1 of Theorem D.1 in the sense that Theorem D.2 requires the $\mu$-PL condition to hold globally, while Part 1 of Theorem D.1 does not. Actually, the latter requires the iteration to start from some neighborhood of $\Gamma$. Later on, we will find from Lemma E.3 that $\mu$-PL provably exists in a neighborhood of $\Gamma$, and we prove that an iteration of AGMs starting within that neighborhood stays within that neighborhood with high probability.

There have been many previous works discussing the convergence bound of Adam. For example, Reddi et al. [2018] and Dereich and Jentzen [2024] give convergence bounds under the convexity condition, Zou et al. [2019], Shi and Li [2021] and Zhang et al. [2022] focus on the cases where learning rates follow a $1/\sqrt{t}$ decay, and the bounds given by Zaheer et al. [2018], Zhang et al. [2022] and Wang et al. [2024b] do not decrease to 0 as $\eta \to 0$. Also, most works [Défossez et al., 2020, Guo et al., 2025, Iiduka, 2022, Wang et al., 2024a, Zhang et al., 2024b, Hong and Lin, 2023] only establish an upper bound on the average of gradient norms over the time of iteration. In contrast, we directly bound the loss term of the last step to $o(1)$. Going beyond convex loss functions, we establish the bound on $\mu$-PL functions, and we focus on the constant learning rate schedule.

# E  Constructing the Working Zones

Note that it is generally hard to ensure some properties that are crucial to the feasibility of our analysis, such as the $\mu$-PL condition or the well-definedness of preconditioned gradient projections. However, this becomes possible when we constrain the discussion inside some local neighborhood of a manifold. So in this subsection, we construct "working zones" around any local minimizer manifold $\Gamma$ such that iterations inside the working zones will be captured by the manifold and obtain certain properties that support the analysis of slow SDE.

For any $\Gamma \subset \mathbb{R}^n$ being a nonempty set and $\boldsymbol{\theta} \in \mathbb{R}^n$, let $\|\cdot\|_2$ denote the Euclidean norm. We denote the distance from $\boldsymbol{\theta}$ to $\Gamma$ as $\mathrm{dist}(\boldsymbol{\theta}, \Gamma) := \inf_{\boldsymbol{\zeta} \in \Gamma} \|\boldsymbol{\theta} - \boldsymbol{\zeta}\|_2$. Note that when $\Gamma$ is closed, the infimum is attained (i.e., there exists $\boldsymbol{\zeta}^\star \in \Gamma$ with $\|\boldsymbol{\theta} - \boldsymbol{\zeta}^\star\|_2 = \mathrm{dist}(\boldsymbol{\theta}, \Gamma)$).

**Definition E.1** (Neighborhood of a Manifold). *For any manifold $\Gamma$ and positive constant $\epsilon$, the $\epsilon$-neighborhood of $\Gamma$, denoted by $\Gamma^\epsilon$ is defined as the set of points $\boldsymbol{\theta}$ such that*

$$\mathrm{dist}(\boldsymbol{\theta}, \Gamma) \leq \epsilon.$$

**Definition E.2** (Preconditioned gradient flow, restatement of Definition 4.1). *For any differentiable function $\mathcal{L}$ and any matrix $\mathbf{S} \in \mathbb{R}^{d \times d}$, the $\mathbf{S}$-preconditioned gradient flow of $\mathcal{L}$ is the ordinary differential equation*

$$\frac{\mathrm{d}\boldsymbol{\theta}(t)}{\mathrm{d}t} = -\mathbf{S}\,\nabla\mathcal{L}\big(\boldsymbol{\theta}(t)\big).$$

*When the objective $\mathcal{L}$ (or the time dependence of $\boldsymbol{\theta}$) is clear from context, we may omit it in the notation and simply refer to the system as the $\mathbf{S}$-preconditioned gradient flow or the gradient flow preconditioned by $\mathbf{S}$.*

**Lemma E.1.** *Let $C_1, C_2 > 0$ with $C_1 \leq C_2$, and let $\mathcal{L} : \mathbb{R}^d \to \mathbb{R}$ be $\rho$-smooth and satisfy the $\mu$-PL condition. For any symmetric matrix $\boldsymbol{S}$ with $C_1\boldsymbol{I} \preceq \boldsymbol{S} \preceq C_2\boldsymbol{I}$, consider the $\boldsymbol{S}$-preconditioned gradient flow of $\mathcal{L}$ starting at $\boldsymbol{\theta}(0) = \boldsymbol{\theta}_0$. Then for any $T > 0$,*

$$\|\boldsymbol{\theta}(T) - \boldsymbol{\theta}_0\|_2 \ \leq \ \frac{2C_2}{C_1\sqrt{2\mu}}\,\sqrt{\mathcal{L}(\boldsymbol{\theta}_0) - \mathcal{L}^*},$$

*where $\mathcal{L}^* = \inf_{\boldsymbol{\theta}} \mathcal{L}(\boldsymbol{\theta})$.*

*Proof.* Since $C_1\boldsymbol{I} \preceq \boldsymbol{S} \preceq C_2\boldsymbol{I}$, we have $\|\boldsymbol{S}\nabla\mathcal{L}(\boldsymbol{\theta})\|_2 \leq C_2\,\|\nabla\mathcal{L}(\boldsymbol{\theta})\|_2$ and $\langle\nabla\mathcal{L}(\boldsymbol{\theta}), \boldsymbol{S}\nabla\mathcal{L}(\boldsymbol{\theta})\rangle \geq C_1\,\|\nabla\mathcal{L}(\boldsymbol{\theta})\|_2^2$ for any $\boldsymbol{\theta}$, which implies

$$\langle\nabla\mathcal{L}(\boldsymbol{\theta}), \boldsymbol{S}\nabla\mathcal{L}(\boldsymbol{\theta})\rangle \geq \frac{C_1}{C_2}\,\|\nabla\mathcal{L}(\boldsymbol{\theta})\|_2\,\|\boldsymbol{S}\nabla\mathcal{L}(\boldsymbol{\theta})\|_2.$$

Then plugging in the above equation gives that, for any $t < T$

$$
\begin{aligned}
\frac{\mathrm{d}}{\mathrm{d}t}\sqrt{\mathcal{L}(\boldsymbol{\theta}(t)) - \mathcal{L}^*} &= \frac{1}{2}\,(\mathcal{L}(\boldsymbol{\theta}(t)) - \mathcal{L}^*)^{-\frac{1}{2}} \cdot \left\langle\nabla\mathcal{L}(\boldsymbol{\theta}(t)), \frac{\mathrm{d}\boldsymbol{\theta}(t)}{\mathrm{d}t}\right\rangle \\
&\leq -\frac{C_1}{2C_2}\,(\mathcal{L}(\boldsymbol{\theta}(t)) - \mathcal{L}^*)^{-\frac{1}{2}} \cdot \|\nabla\mathcal{L}(\boldsymbol{\theta}(t))\|_2\left\|\frac{\mathrm{d}\boldsymbol{\theta}(t)}{\mathrm{d}t}\right\|_2 \\
&\leq -\frac{C_1}{2C_2}\,(\mathcal{L}(\boldsymbol{\theta}(t)) - \mathcal{L}^*)^{-\frac{1}{2}} \cdot \sqrt{2\mu(\mathcal{L}(\boldsymbol{\theta}(t)) - \mathcal{L}^*)}\left\|\frac{\mathrm{d}\boldsymbol{\theta}(t)}{\mathrm{d}t}\right\|_2 \\
&= -\frac{\sqrt{2\mu}C_1}{2C_2}\left\|\frac{\mathrm{d}\boldsymbol{\theta}(t)}{\mathrm{d}t}\right\|_2.
\end{aligned}
$$

Integrating both sides gives us

$$
\begin{aligned}
\sqrt{\mathcal{L}(\boldsymbol{\theta}_0) - \mathcal{L}^*} &\geq \frac{\sqrt{2\mu}C_1}{2C_2}\int_0^T\left\|\frac{\mathrm{d}\boldsymbol{\theta}(t)}{\mathrm{d}t}\right\|_2 \\
&\geq \frac{\sqrt{2\mu}C_1}{2C_2}\,\|\boldsymbol{\theta}_0 - \boldsymbol{\theta}(T)\|_2.
\end{aligned}
$$

The above equations complete the proof. $\qquad\square$

To avoid ambiguity, all comparisons between vectors and scalars are interpreted componentwise. Specifically, for $\boldsymbol{v} \in \mathbb{R}^D$ and a scalar $c \in \mathbb{R}$ we write $\boldsymbol{v} \leq c$ (resp. $\boldsymbol{v} < c$) iff $v_i \leq c$ (resp. $v_i < c$) for every coordinate $i$. Equivalently $\boldsymbol{v} \geq c$ (resp. $\boldsymbol{v} > c$) means $v_i \geq c$ (resp. $v_i > c$) for all $i$. In particular the notation $0 \leq \boldsymbol{v} \leq c$ means $0 \leq v_i \leq c$ for every $i$, which is the convention used in the sequel. Recall that $\mathbb{R}_{\geq 0}^D$ means $\{\boldsymbol{v} \mid \boldsymbol{v} \in \mathbb{R}^D, \boldsymbol{v} \geq 0\}$. In the sequel, we slightly abuse this notation such that for any subset $I \subseteq \mathbb{R}$, $\mathbb{R}_I^D$ means $\{\boldsymbol{v} \mid \boldsymbol{v} \in R^D, v_i \in I \text{ for all } i \in [D]\}$.

**Lemma E.2.** *There exist constants $R_1, R_2 > 0$ such that for all $k \geq 0$, $0 \leq \boldsymbol{v}_k \leq R_1$ and $S(\boldsymbol{v}) \preceq R_2 \boldsymbol{I}$ almost surely. Moreover, $S$ is Lipschitz on $\mathbb{R}_{[0,R_1]}^D$.*

*Proof.* From Assumption 3.3, all noisy gradients $\nabla \ell_k(\boldsymbol{\theta}_k)$ are uniformly bounded by a constant $R$. Hence $V(\nabla \ell_k(\boldsymbol{\theta}_k) \nabla \ell_k(\boldsymbol{\theta}_k)^\top)$ is also bounded. Combining with the condition that $V(\boldsymbol{g}\boldsymbol{g}^\top) \geq 0$ for all $\boldsymbol{g}$, we have $V(\nabla \ell_k(\boldsymbol{\theta}_k) \nabla \ell_k(\boldsymbol{\theta}_k)^\top) \in \mathbb{R}_{[0,R_1]}^D$ for some constant $R_1$. Since $\boldsymbol{v}_k$ is an exponential moving average of previous $V(\nabla \ell_t(\boldsymbol{\theta}_t) \nabla \ell_t(\boldsymbol{\theta}_t)^\top)$ terms and $\mathbb{R}_{[0,R_1]}^D$ is convex, we have $\boldsymbol{v}_k \in \mathbb{R}_{[0,R_1]}^D$ for all $k \geq 0$.

From Assumption 4.1, $S$ is $\rho_s$-smooth, hence both $S$ and $\nabla S$ are continuous, thus are bounded on the compact set $\mathbb{R}_{[0,R_1]}^D$. The boundedness of $S$ gives the existence of $R_2$, while the boundedness of $\nabla S$ gives the Lipschitzness of $S$. $\qquad\qquad\qquad \square$

We continue to use the notations $R_1$ and $R_2$ throughout the following part of the paper. Note that for all optimizers listed in Table 1, setting $R_1 = R^2$ is sufficient. Another thing to clarify is the relationship between the $R_2$ here and the stabilizing constant $\epsilon$ used by optimizers in Table 1. We will call it $\epsilon_{\text{optim}}$ here, so as to distinguish from the $\epsilon$ notations that represent a distance (for instance, the $\epsilon$ in Definition E.1 or Lemma E.3).

**Remark E.1** (Relationship between $R_2$ and $\epsilon_{\text{optim}}$). *Setting $R_2 := 1/\epsilon_{\text{optim}}$ here is theoretically enough for the requirement in Lemma E.2 to hold, but will introduce a large constant to the proof since $\epsilon_{\text{optim}}$ is very small in practice; However, in practice the gradient noise is also very likely to keep $\boldsymbol{v}$ away from zero, thus the operational $R_0$ that governs empirical convergence is usually much smaller than the worst-case $1/\epsilon_{\text{optim}}$.*

Now we are ready to construct working zones in which nice properties are ensured to benefit our analysis. For all $\epsilon > 0$, define $\mathcal{X}^\epsilon := \Gamma^\epsilon \times \mathbb{R}_{[0,R_1]}^D$ as the set of AGM states $(\boldsymbol{\theta}, \boldsymbol{v})$ where $\boldsymbol{\theta}$ lies in $\Gamma^\epsilon$ and $0 \leq \boldsymbol{v} \leq R_1$.

We construct nested working zones $(\Gamma^{\epsilon_1}, \Gamma^{\epsilon_2}, \Gamma^{\epsilon_3})$ in the following way:

**Lemma E.3** (Working Zone Lemma). *We denote the minimal distance of $\Gamma$ and any other local minimizer manifold as $\epsilon_4$. There exist positive constants $\epsilon_1, \epsilon_2, \epsilon_3$ such that $\epsilon_1 < \epsilon_2 < \epsilon_3 < \epsilon_4$ and $\Gamma^{\epsilon_1}, \Gamma^{\epsilon_2}, \Gamma^{\epsilon_3}$ satisfy the following properties:*

1. *$\mathcal{L}$ is $\mu$-PL in $\Gamma^{\epsilon_3}$ for some constant $\mu > 0$.*

2. *For all matrices $\boldsymbol{S} \in \mathbb{R}^{d \times d}$ with $\frac{1}{R_0} \boldsymbol{I} \preceq \boldsymbol{S} \preceq \frac{1}{\epsilon} \boldsymbol{I}$, starting from any initial point $\boldsymbol{\theta}_0 \in \Gamma^{\epsilon_2}$, the gradient flow preconditioned by $\boldsymbol{S}$ converges to a point in $\Gamma$.*

3. *Under Assumption 3.1 and Assumption 4.1 the function $F : \mathcal{X}^{\epsilon_2} \to \mathbb{R}^d, (\boldsymbol{\theta}, \boldsymbol{v}) \mapsto \Phi_{S(\boldsymbol{v})}(\boldsymbol{\theta})$ is $C^4$ on $\mathcal{X}^{\epsilon_1}$.*

*Proof.* By Lemma H.3 in Lyu et al. [2022], there exists an $\epsilon_3$-neighborhood of $\Gamma$ where $\mathcal{L}$ is $\mu$-PL for some $\mu > 0$. WLOG let $\epsilon_3 < \epsilon_4$.

We prove the second property by contradiction. Let $C_1 = 1/R_0$ and $C_2 = 1/\epsilon$. Let $\epsilon_2$ be some constant such that $\epsilon_2 + \sqrt{\frac{\rho}{\mu}} \cdot \frac{C_2}{C_1} \epsilon_2 < \epsilon_3$. For any starting point $\boldsymbol{\theta}_0 \in \Gamma^{\epsilon_2}$, and any preconditioning matrix $\boldsymbol{S}$ satisfying $C_1 \boldsymbol{I} \preceq \boldsymbol{S} \preceq C_2 \boldsymbol{I}$, assume on the contrary that the preconditioned gradient flow starting from $\boldsymbol{\theta}(0) = \boldsymbol{\theta}_0$ will leave $\Gamma^{\epsilon_3}$ at some finite time. Then let $T = \inf \{t : \boldsymbol{\theta}(t) \notin \Gamma^{\epsilon_3}\} < \infty$.

Using Lemma E.1 and combining the $\mu$-PL condition, we conclude that

$$\|\boldsymbol{\theta}_0 - \boldsymbol{\theta}(T)\|_2 \leq \frac{2C_2}{\sqrt{2\mu C_1}} \sqrt{\mathcal{L}(\boldsymbol{\theta}_0) - \mathcal{L}^*} \leq \frac{2C_2}{\sqrt{2\mu C_1}} \cdot \sqrt{\frac{\mu}{2}} \|\boldsymbol{\theta}_0 - \boldsymbol{\theta}^*\|_2 = \sqrt{\frac{\rho}{\mu}} \cdot \frac{C_2}{C_1} \|\boldsymbol{\theta}_0 - \boldsymbol{\theta}^*\|_2$$

for any $\boldsymbol{\theta}^* \in \Gamma$. Hence $\boldsymbol{\theta}(T) \in \Gamma^{\epsilon_3}$, which is a contradition.

Next we begin the construction of $\Gamma^{\epsilon_1}$ with Assumptions 3.1 and 4.1. Define a function $f(\boldsymbol{\theta}, \boldsymbol{v})$ : $\mathbb{R}^{d+D} \to \mathbb{R}^{d+D}$ as

$$f(\boldsymbol{\theta}, \boldsymbol{v}) := (-S(\boldsymbol{v})\nabla\mathcal{L}(\boldsymbol{\theta}), \boldsymbol{v}),$$

then $f$ is $\mathcal{C}^4$ on $\mathbb{R}^d \times \mathbb{R}^D_{[0,R_1]}$. Let $\tilde{r}$ be a constant such that $\tilde{r} > \epsilon_2$. Substituting $f_0 = f$, $r = \sqrt{\tilde{r}^2 + d \cdot R_1^2}$, $x_0 = (\boldsymbol{\theta}_0, \boldsymbol{v}_0)$ such that each entry of $\boldsymbol{v}_0$ is $R_1/2$ and $\boldsymbol{\theta}_0$ be arbitrary point in $\Gamma$, and $B = \mathcal{X}^{\tilde{r}}$ into Lemma B.4 in Duistermaat and Kolk [2012], we conclude that there exists some constant $\delta$ such that the mapping $\gamma_\delta(\boldsymbol{\theta}, \boldsymbol{v})$ defined by:

$$\boldsymbol{\theta}(0) = \boldsymbol{\theta}, \quad \frac{d\boldsymbol{\theta}(t)}{dt} = -S(\boldsymbol{v})\nabla\mathcal{L}(\boldsymbol{\theta}(t)), \quad \gamma_\delta(\boldsymbol{\theta}, \boldsymbol{v}) = \boldsymbol{\theta}(\delta)$$

is well-defined and $\mathcal{C}^4$ on $\mathcal{X}^{\tilde{r}}$. Note that we require a slight modification of the original proof since $B$ is now a factorization of a ball and a hypercube instead of a ball, but the convexity of $B$ is preserved, hence the modification is trivial.

Note that the constant $\delta$ can be independent with $\boldsymbol{\theta}_0$ to fulfill the requirements of Lemma B.4 in Duistermaat and Kolk [2012] since $\|\nabla\mathcal{L}\|_2$ and $\|\nabla^2\mathcal{L}\|_2$ can be uniformly bounded. Take $\epsilon_1 = 0.9\epsilon_2$, then for any $\boldsymbol{\theta} \in \Gamma^{\epsilon_1}$, a small open neighborbood of $\boldsymbol{\theta}$ stays in the $\epsilon_2$-neighborhoods of two different points on $\Gamma$. Taking union of all $\boldsymbol{\theta}_0 \in \Gamma$, we conclude that $\gamma_\delta$ is $\mathcal{C}^4$ on $\mathcal{X}^{\epsilon_1}$. Finally, we use Theorem 6.4 in Falconer [1983] to conclude that $F(\boldsymbol{\theta}, \boldsymbol{v}) := \Phi_{S(\boldsymbol{v})}(\boldsymbol{\theta})$ is $\mathcal{C}^4$ on $\mathcal{X}^{\epsilon_1}$. $\qquad\square$

# F  Proof of the Convergence of AGMs

In this section, we aim to prove Theorem D.2 and the first part of Theorem D.1. Specifically, for some constant $\gamma = 1 - \Theta(\eta)$, we prove that the loss value of AGM converges to $\tilde{\mathcal{O}}(\gamma^K + \eta)$ within $K$ steps with high probability. If we substitute $K = \mathcal{O}\left(\frac{1}{\eta}\log\frac{1}{\eta}\right)$, this will recover the first part of Theorem D.1; However, this convergence analysis works for any $K = \mathcal{O}(\text{poly}(1/\eta))$, and substituting $K = \mathcal{O}(\eta^{-2})$ will give us a high probability guarantee that the iteration stays near manifold in the whole scope of our analysis, which helps the proof of the second part too.

First, we introduce some additional notations that will be used in our proof. In the AGM framework, an algorithm starts from initial state $\boldsymbol{\theta}_0$, and we set $\boldsymbol{m}_0 = \mathbf{0} \in \mathbb{R}^\mathbf{d}$, $\boldsymbol{v}_0 = \mathbf{0} \in \mathbb{R}^\mathbf{D}$. For every $k \geq 0$, we use **step** $k + 1$ to refer to the process of obtaining the noisy gradient $\nabla\ell_k(\boldsymbol{\theta}_k)$ and then $\boldsymbol{m}_{k+1}, \boldsymbol{v}_{k+1}$ and $\boldsymbol{\theta}_{k+1}$.

For any $k \geq 0$, to simplify the notation, we denote that

$$\boldsymbol{g}_k := \nabla\ell_k(\boldsymbol{\theta}_k), \quad \boldsymbol{z}_k := \ell_k(\boldsymbol{\theta}_k) - \mathcal{L}(\boldsymbol{\theta}_k) \sim \mathcal{Z}(\boldsymbol{\theta}_k), \quad \boldsymbol{S}_k := S(\boldsymbol{v}_k),$$
$$\boldsymbol{U}_{k+1} := S(\boldsymbol{v}_k)\boldsymbol{g}_k, \quad \boldsymbol{u}_{k+1} := S(\boldsymbol{v}_k)\boldsymbol{m}_{k+1}, \quad \phi_k := \Phi_{\boldsymbol{S}_k}(\boldsymbol{\theta}_k).$$

We use **time** $k$ to refer to the time right before step $k + 1$ happens, i.e. the time right after we get $\boldsymbol{\theta}_k$. We also define $\{\mathcal{F}_k\}$ as the natural filtration generated by the history of optimization, where each $\mathcal{F}_k = \sigma(\boldsymbol{\theta}_0, \boldsymbol{z}_0, \cdots, \boldsymbol{z}_{k-1})$ can be interpreted as "all the information available up to time $k$". We use the notation $\mathbb{E}_k$ to denote the expectation conditioned on $\mathcal{F}_k$.

To start with, we prove that the descent direction of each step does not veer off the direction of a preconditioned gradient descent, and the mismatch term can also be constrained by a list of martingales. After that, we can ensure a decay in the loss function every step, with some small perturbations that can be dealt with using Azuma-Hoeffding's inequality.

From Lemma F.1 throughout Lemma F.5, we will assume that the loss function $\mathcal{L}$ satisfies $\mu$-PL condition at each iteration step, which is automatically satisfied in the setting of Theorem D.2. follows directly from the result. After that, we argue that if the loss function satisfies $\mu$-PL only within some local neighborhood, an AGM starting near enough to the manifold will stick to the manifold with high probability, which leads to the first part of Theorem D.1.

**Lemma F.1.** *Let $\mathcal{L}$ satisfy Assumption 3.2. Define $\tilde{\boldsymbol{v}}_k := \beta_2\boldsymbol{v}_{k-1} + (1-\beta_2)\mathbb{E}_{k-1}[V(\boldsymbol{g}_{k-1}\boldsymbol{g}_{k-1}^\top)]$. There exist a constant $C_{1a}$ and a constant $C_{1b}$ independent of $\mathcal{L}$, such that for any $k \geq 1$,*

$$\langle \nabla\mathcal{L}(\boldsymbol{\theta}_{k-1}), \boldsymbol{U}_k \rangle = \nabla\mathcal{L}(\boldsymbol{\theta}_{k-1})^\top S(\tilde{\boldsymbol{v}}_k)\nabla\mathcal{L}(\boldsymbol{\theta}_{k-1}) - Y_k - X_k,$$

*where $Y_k$ and $X_k$ are two $\mathcal{F}_k$-measurable random variables such that:*

1. $|Y_k| \leq C_{1a} \|\nabla \mathcal{L}(\boldsymbol{\theta}_{k-1})\|_2 \cdot \eta^2$ a.s.

2. $|X_k| \leq C_{1b} \|\nabla \mathcal{L}(\boldsymbol{\theta}_{k-1})\|_2$ a.s., and $\mathbb{E}_{k-1}[X_k] = 0$.

*Proof.* We first peel the $S(\tilde{\boldsymbol{v}}_k)$ part off the $S(\boldsymbol{v}_k)$ term:

$$\langle \nabla \mathcal{L}(\boldsymbol{\theta}_{k-1}), \boldsymbol{U}_k \rangle = \langle \nabla \mathcal{L}(\boldsymbol{\theta}_{k-1}), S(\boldsymbol{v}_k)\boldsymbol{g}_{k-1} \rangle$$
$$= \langle \nabla \mathcal{L}(\boldsymbol{\theta}_{k-1}), S(\tilde{\boldsymbol{v}}_k)\boldsymbol{g}_{k-1} \rangle + \langle \nabla \mathcal{L}(\boldsymbol{\theta}_{k-1}), (S(\boldsymbol{v}_k) - S(\tilde{\boldsymbol{v}}_k)) \boldsymbol{g}_{k-1} \rangle.$$

Define $Y_k$ as $Y_k = -\langle \nabla \mathcal{L}(\boldsymbol{\theta}_{k-1}), (S(\boldsymbol{v}_k) - S(\tilde{\boldsymbol{v}}_k)) \boldsymbol{g}_{k-1} \rangle$, then it holds almost surely that

$$|Y_k| \leq \|\nabla \mathcal{L}(\boldsymbol{\theta}_{k-1})\|_2 \|(S(\boldsymbol{v}_k) - S(\tilde{\boldsymbol{v}}_k)) \boldsymbol{g}_{k-1}\|_2.$$

Since $S$ is Lipscitz, $V$ is linear and

$$\|\tilde{\boldsymbol{v}}_k - \boldsymbol{v}_k\|_2 = (1 - \beta_2) \left\|\mathbb{E}_{k-1}\left[V\left(\boldsymbol{g}_{k-1}\boldsymbol{g}_{k-1}^\top\right)\right] - V\left(\boldsymbol{g}_{k-1}\boldsymbol{g}_{k-1}^\top\right)\right\|_2,$$

we conclude that $|Y_k| \leq C_{1a} \|\nabla \mathcal{L}(\boldsymbol{\theta}_{k-1})\|_2 \cdot \eta^2$ a.s. for some constant $C_{1a}$. The rest term $\langle \nabla \mathcal{L}(\boldsymbol{\theta}_{k-1}), S(\tilde{\boldsymbol{v}}_k)\boldsymbol{g}_{k-1} \rangle$ can also be decomposed into a deterministic part and a random part as:

$$\langle \nabla \mathcal{L}(\boldsymbol{\theta}_{k-1}), S(\tilde{\boldsymbol{v}}_k)\boldsymbol{g}_{k-1} \rangle = \langle \nabla \mathcal{L}(\boldsymbol{\theta}_{k-1}), S(\tilde{\boldsymbol{v}}_k)(\nabla \mathcal{L}(\boldsymbol{\theta}_{k-1}) + \boldsymbol{z}_{k-1}) \rangle$$
$$= \nabla \mathcal{L}(\boldsymbol{\theta}_{k-1})^\top S(\tilde{\boldsymbol{v}}_k)\nabla \mathcal{L}(\boldsymbol{\theta}_{k-1}) + \langle \boldsymbol{z}_{k-1}, S(\tilde{\boldsymbol{v}}_k)^\top \nabla \mathcal{L}(\boldsymbol{\theta}_{k-1}) \rangle.$$

Now we only need to let $X_k = \langle \boldsymbol{z}_{k-1}, S(\tilde{\boldsymbol{v}}_k)^\top \nabla \mathcal{L}(\boldsymbol{\theta}_{k-1}) \rangle$. It's easy to see that $\mathbb{E}_{k-1}[X_k] = 0$ and $|X_k| \leq C_{1b} \|\nabla \mathcal{L}(\boldsymbol{\theta}_{k-1})\|_2$ a.s. for some constant $C_{1b}$. Finally, note that $C_{1b}$ is the multiplication of a constant bounding the magnitude of $\boldsymbol{z}$ and $R_2$ which bounds $\|S\|_2$, which is independent of $\mathcal{L}$. This completes the proof. $\square$

**Lemma F.2** (Descent Lemma of the AGM Framework). *Let $\mathcal{L}$ satisfy Assumption 3.2. For any $k \geq 1$ it holds that*

$$\mathcal{L}(\boldsymbol{\theta}_k) - \mathcal{L}(\boldsymbol{\theta}_{k-1}) \leq C_2 \eta^2 - \eta(1 - \beta_1) \sum_{i=1}^{k} \beta_1^{k-i} \langle \nabla \mathcal{L}(\boldsymbol{\theta}_{i-1}), \boldsymbol{U}_i \rangle$$

*for some constant $C_2$.*

*Proof.* From the smoothness of $\mathcal{L}$ we have

$$\mathcal{L}(\boldsymbol{\theta}_k) - \mathcal{L}(\boldsymbol{\theta}_{k-1}) \leq -\langle \nabla \mathcal{L}(\boldsymbol{\theta}_{k-1}), \eta \boldsymbol{u}_k \rangle + \frac{\rho \eta^2}{2} \|\boldsymbol{u}_k\|_2^2.$$

If $k = 1$, then $\boldsymbol{m}_k = (1 - \beta_1)\boldsymbol{g}_{k-1}$, so $\boldsymbol{u}_k = (1 - \beta_1)\boldsymbol{U}_k$, and the statement trivially holds as long as $C_2 \geq \frac{\rho}{2} \|\boldsymbol{u}_k\|_2^2$. If $k > 1$, then the $-\langle \nabla \mathcal{L}(\boldsymbol{\theta}_{k-1}), \boldsymbol{u}_k \rangle$ term can be expanded as

$$-\langle \nabla \mathcal{L}(\boldsymbol{\theta}_{k-1}), \boldsymbol{u}_k \rangle = -\langle \nabla \mathcal{L}(\boldsymbol{\theta}_{k-1}), S(\boldsymbol{v}_k)\boldsymbol{m}_k \rangle$$
$$= -\langle \nabla \mathcal{L}(\boldsymbol{\theta}_{k-1}), S(\boldsymbol{v}_k)(\beta_1 \boldsymbol{m}_{k-1} + (1 - \beta_1)\boldsymbol{g}_{k-1}) \rangle$$
$$= -\beta_1 \langle \nabla \mathcal{L}(\boldsymbol{\theta}_{k-1}), S(\boldsymbol{v}_k)\boldsymbol{m}_{k-1} \rangle - (1 - \beta_1) \langle \nabla \mathcal{L}(\boldsymbol{\theta}_{k-1}), S(\boldsymbol{v}_k)\boldsymbol{g}_{k-1} \rangle$$
$$= -\beta_1 \langle \nabla \mathcal{L}(\boldsymbol{\theta}_{k-2}), S(\boldsymbol{v}_{k-1})\boldsymbol{m}_{k-1} \rangle - (1 - \beta_1) \langle \nabla \mathcal{L}(\boldsymbol{\theta}_{k-1}), \boldsymbol{U}_k \rangle$$
$$\quad - \beta_1 \langle \nabla \mathcal{L}(\boldsymbol{\theta}_{k-1}) - \nabla \mathcal{L}(\boldsymbol{\theta}_{k-2}), S(\boldsymbol{v}_{k-1})\boldsymbol{m}_{k-1} \rangle$$
$$\quad - \beta_1 \langle \nabla \mathcal{L}(\boldsymbol{\theta}_{k-1}), (S(\boldsymbol{v}_k) - S(\boldsymbol{v}_{k-1})) \boldsymbol{m}_{k-1} \rangle$$
$$\leq -\beta_1 \langle \nabla \mathcal{L}(\boldsymbol{\theta}_{k-2}), S(\boldsymbol{v}_{k-1})\boldsymbol{m}_{k-1} \rangle - (1 - \beta_1) \langle \nabla \mathcal{L}(\boldsymbol{\theta}_{k-1}), \boldsymbol{U}_k \rangle$$
$$\quad + \beta_1 \|\nabla \mathcal{L}(\boldsymbol{\theta}_{k-1}) - \nabla \mathcal{L}(\boldsymbol{\theta}_{k-2})\|_2 \|S(\boldsymbol{v}_{k-1})\boldsymbol{m}_{k-1}\|_2$$
$$\quad + \beta_1 \|\nabla \mathcal{L}(\boldsymbol{\theta}_{k-1})\|_2 \|(S(\boldsymbol{v}_k) - S(\boldsymbol{v}_{k-1})) \boldsymbol{m}_{k-1}\|_2.$$

Note that a single step of update on $\boldsymbol{\theta}$ and $\boldsymbol{v}$ is small since

$$\boldsymbol{\theta}_k - \boldsymbol{\theta}_{k-1} = \eta \boldsymbol{u}_k,$$
$$\boldsymbol{v}_k - \boldsymbol{v}_{k-1} = \beta_2 \boldsymbol{v}_{k-1} + (1 - \beta_2)V\left(\boldsymbol{g}_{k-1}\boldsymbol{g}_{k-1}^\top\right) - \boldsymbol{v}_{k-1}$$
$$= (1 - \beta_2)\left(V\left(\boldsymbol{g}_{k-1}\boldsymbol{g}_{k-1}^\top\right) - \boldsymbol{v}_{k-1}\right),$$

which implies that $\|\boldsymbol{\theta}_k - \boldsymbol{\theta}_{k-1}\|_2 = \mathcal{O}(\eta)$ and $\|\boldsymbol{v}_k - \boldsymbol{v}_{k-1}\|_2 = \mathcal{O}(\eta^2)$. We then leverage the smoothness of $\nabla \mathcal{L}$ and $S$ to conclude that there exists some constant $\tilde{C}_2$ such that

$$-\langle \nabla \mathcal{L}(\boldsymbol{\theta}_{k-1}), \boldsymbol{u}_k \rangle \leq -\beta_1 \langle \nabla \mathcal{L}(\boldsymbol{\theta}_{k-2}), \boldsymbol{u}_{k-1} \rangle - (1 - \beta_1) \langle \nabla \mathcal{L}(\boldsymbol{\theta}_{k-1}), \boldsymbol{U}_k \rangle + \beta_1 \tilde{C}_2 \eta.$$

Giving that $\boldsymbol{u}_0 = \boldsymbol{0}$, we can expand this formula iteratively as

$$
\begin{aligned}
-\langle \nabla \mathcal{L}(\boldsymbol{\theta}_{k-1}), \boldsymbol{u}_k \rangle &\leq -\beta_1 \langle \nabla \mathcal{L}(\boldsymbol{\theta}_{k-2}), \boldsymbol{u}_{k-1} \rangle - (1 - \beta_1) \langle \nabla \mathcal{L}(\boldsymbol{\theta}_{k-1}), \boldsymbol{U}_k \rangle + \beta_1 \tilde{C}_2 \eta \\
&\leq -\beta_1^2 \langle \nabla \mathcal{L}(\boldsymbol{\theta}_{k-3}), \boldsymbol{u}_{k-2} \rangle - \beta_1(1 - \beta_1) \langle \nabla \mathcal{L}(\boldsymbol{\theta}_{k-2}), \boldsymbol{U}_{k-1} \rangle \\
&\quad - (1 - \beta_1) \langle \nabla \mathcal{L}(\boldsymbol{\theta}_{k-1}), \boldsymbol{U}_k \rangle + \beta_1 \tilde{C}_2 \eta + \beta_1^2 \tilde{C}_2 \eta \\
&\leq \cdots \\
&\leq -(1 - \beta_1) \sum_{i=1}^k \beta_1^{k-i} \langle \nabla \mathcal{L}(\boldsymbol{\theta}_{i-1}), \boldsymbol{U}_i \rangle + \beta_1^{k-i+1} \tilde{C}_2 \eta \\
&\leq \frac{\beta_1}{1 - \beta_1} \tilde{C}_2 \eta - (1 - \beta_1) \sum_{i=1}^k \beta_1^{k-i} \langle \nabla \mathcal{L}(\boldsymbol{\theta}_{i-1}), \boldsymbol{U}_i \rangle.
\end{aligned}
$$

Plugging in, we get

$$
\begin{aligned}
\mathcal{L}(\boldsymbol{\theta}_k) - \mathcal{L}(\boldsymbol{\theta}_{k-1}) &\leq \frac{\beta_1}{1 - \beta_1} \tilde{C}_2 \eta^2 + \frac{\rho \eta^2}{2} \|\boldsymbol{u}_k\|_2^2 - \eta(1 - \beta_1) \sum_{i=1}^k \beta_1^{k-i} \langle \nabla \mathcal{L}(\boldsymbol{\theta}_{i-1}), \boldsymbol{U}_i \rangle \\
&\leq C_2 \eta^2 - \eta(1 - \beta_1) \sum_{i=1}^k \beta_1^{k-i} \langle \nabla \mathcal{L}(\boldsymbol{\theta}_{i-1}), \boldsymbol{U}_i \rangle,
\end{aligned}
$$

for some constant $C_2$. $\qquad \square$

**Lemma F.3.** *Let $\mathcal{L}$ satisfy Assumption 3.2, and assume that $\mu$-PL condition is satisfied at all $\boldsymbol{\theta}_k$ where $k \geq 0$. Define $\gamma := 1 - \frac{2\eta\mu(1-\beta_1)}{R_0}$. For any $k \geq 0$, we have*

$$\mathcal{L}(\boldsymbol{\theta}_k) - \mathcal{L}^* \leq \gamma^k (\mathcal{L}(\boldsymbol{\theta}_0) - \mathcal{L}^*) + \eta(1 - \beta_1) \sum_{i=1}^k X_i \sum_{j=i}^k \gamma^{k-j} \beta_1^{j-i} + C_3 \eta$$

*for some constant $C_3$.*

*Proof.* We start from Lemma F.2 and plug in Lemma F.1:

$$
\begin{aligned}
\mathcal{L}(\boldsymbol{\theta}_k) - \mathcal{L}(\boldsymbol{\theta}_{k-1}) &\leq C_2 \eta^2 - \eta(1 - \beta_1) \sum_{i=1}^k \beta_1^{k-i} \langle \nabla \mathcal{L}(\boldsymbol{\theta}_{i-1}), \boldsymbol{U}_i \rangle \\
&= C_2 \eta^2 - \eta(1 - \beta_1) \sum_{i=1}^k \beta_1^{k-i} \left( \nabla \mathcal{L}(\boldsymbol{\theta}_{i-1})^\top S(\tilde{\boldsymbol{v}}_i) \nabla \mathcal{L}(\boldsymbol{\theta}_{i-1}) - Y_i - X_i \right).
\end{aligned}
$$

Since $|Y_i| \leq C_{1a} \|\nabla \mathcal{L}(\boldsymbol{\theta}_{i-1})\|_2 \cdot \eta^2$ for every $i$, the effect of $Y$ is negligible:

$$\left| \eta(1 - \beta_1) \sum_{i=1}^k \beta_1^{k-i} Y_i \right| \leq C_{1a} \eta^3 \cdot \max_{i=0}^k \{\|\nabla \mathcal{L}(\boldsymbol{\theta}_{i-1})\|_2\} = o(\eta^2),$$

and we can absorb it into the $C_2 \eta^2$ term to write out that

$$\mathcal{L}(\boldsymbol{\theta}_k) - \mathcal{L}(\boldsymbol{\theta}_{k-1}) \leq \tilde{C}_3 \eta^2 - \eta(1 - \beta_1) \sum_{i=1}^k \beta_1^{k-i} \left( \nabla \mathcal{L}(\boldsymbol{\theta}_{i-1})^\top S(\tilde{\boldsymbol{v}}_i) \nabla \mathcal{L}(\boldsymbol{\theta}_{i-1}) - X_i \right)$$

for some constant $\tilde{C}_3$. Note that $S(\tilde{\boldsymbol{v}}_i) \succeq \frac{1}{R_0}$, so $\nabla \mathcal{L}(\boldsymbol{\theta}_{i-1})^\top S(\tilde{\boldsymbol{v}}_i) \nabla \mathcal{L}(\boldsymbol{\theta}_{i-1}) \geq \frac{1}{R_0} \|\nabla \mathcal{L}(\boldsymbol{\theta}_{i-1})\|_2^2$ for any $i$, hence

$$\mathcal{L}(\boldsymbol{\theta}_k) - \mathcal{L}(\boldsymbol{\theta}_{k-1}) \leq \tilde{C}_3 \eta^2 - \eta(1 - \beta_1) \sum_{i=1}^k \beta_1^{k-i} \left( \frac{1}{R_0} \|\nabla \mathcal{L}(\boldsymbol{\theta}_{i-1})\|_2^2 - X_i \right). \qquad (4)$$

Combining with the $\mu$-PL property $\|\nabla \mathcal{L}(\boldsymbol{\theta}_{i-1})\|_2^2 \geq 2\mu(\mathcal{L}(\boldsymbol{\theta}_{i-1}) - \mathcal{L}^*)$, we have

$$\mathcal{L}(\boldsymbol{\theta}_k) - \mathcal{L}^* \leq \tilde{C}_3\eta^2 + \mathcal{L}(\boldsymbol{\theta}_{k-1}) - \mathcal{L}^* - \frac{2\eta\mu(1-\beta_1)}{R_0}\sum_{i=1}^k \beta_1^{k-i}(\mathcal{L}(\boldsymbol{\theta}_{i-1}) - \mathcal{L}^*)$$

$$+ \eta(1-\beta_1)\sum_{i=1}^k \beta_1^{k-i}X_i$$

$$\leq \tilde{C}_3\eta^2 + \left(1 - \frac{2\eta\mu(1-\beta_1)}{R_0}\right)(\mathcal{L}(\boldsymbol{\theta}_{k-1}) - \mathcal{L}^*) + \eta(1-\beta_1)\sum_{i=1}^k \beta_1^{k-i}X_i$$

$$= \tilde{C}_3\eta^2 + \gamma(\mathcal{L}(\boldsymbol{\theta}_{k-1}) - \mathcal{L}^*) + \eta(1-\beta_1)\sum_{i=1}^k \beta_1^{k-i}X_i.$$

Note that we can expand the $\mathcal{L}(\boldsymbol{\theta}_{k-1}) - \mathcal{L}^*$ term iteratively to obtain a generic formula for $\mathcal{L}(\boldsymbol{\theta}_k) - \mathcal{L}^*$:

$$\mathcal{L}(\boldsymbol{\theta}_k) - \mathcal{L}^* \leq \gamma(\mathcal{L}(\boldsymbol{\theta}_{k-1}) - \mathcal{L}^*) + \eta(1-\beta_1)\sum_{i=1}^k \beta_1^{k-i}X_i + \tilde{C}_3\eta^2$$

$$\leq \gamma^k(\mathcal{L}(\boldsymbol{\theta}_0) - \mathcal{L}^*) + \eta(1-\beta_1)\sum_{j=1}^k \gamma^{k-j}\sum_{i=1}^j \beta_1^{j-i}X_i + \sum_{j=1}^k \gamma^{k-j}\tilde{C}_3\eta^2$$

$$\leq \gamma^k(\mathcal{L}(\boldsymbol{\theta}_0) - \mathcal{L}^*) + \eta(1-\beta_1)\sum_{i=1}^k X_i \sum_{j=i}^k \gamma^{k-j}\beta_1^{j-i} + C_3\eta,$$

where $C_3 = \tilde{C}_3 \cdot \frac{R_0}{2\mu(1-\beta_1)}$. $\qquad\square$

**Corollary F.1.** *Let $\mathcal{L}$ satisfy Assumption 3.2. There exists a constant $\bar{C}$ independent of $\mathcal{L}$, such that $\forall k > 0$, if*

$$\|\nabla \mathcal{L}(\boldsymbol{\theta}_{k-1})\|_2 > \bar{C},$$

*then with sufficiently small $\eta$, we have*

$$\mathcal{L}(\boldsymbol{\theta}_k) < \mathcal{L}(\boldsymbol{\theta}_{k-1}).$$

*Proof.* Note that Equation (4) can be obtained without PL condition:

$$\mathcal{L}(\boldsymbol{\theta}_k) - \mathcal{L}(\boldsymbol{\theta}_{k-1}) \leq \tilde{C}_3\eta^2 - \eta(1-\beta_1)\sum_{i=1}^k \beta_1^{k-i}\left(\frac{1}{R_0}\|\nabla\mathcal{L}(\boldsymbol{\theta}_{i-1})\|_2^2 - X_i\right).$$

From Lemma F.1, $|X_i| \leq C_{1b}\|\nabla\mathcal{L}(\boldsymbol{\theta}_{i-1})\|_2, \forall i \leq k$ where $C_{1b}$ is a constant independent of $\mathcal{L}$. So

$$\frac{1}{R_0}\|\nabla\mathcal{L}(\boldsymbol{\theta}_{i-1})\|_2^2 - X_i \geq \frac{1}{R_0}\|\nabla\mathcal{L}(\boldsymbol{\theta}_{i-1})\|_2^2 - C_{1b}\|\nabla\mathcal{L}(\boldsymbol{\theta}_{i-1})\|_2 \geq -\frac{R_0C_{1b}^2}{4},$$

where the last inequality uses $a^2 - 2ab \geq -b^2$ with $a = \|\nabla\mathcal{L}(\boldsymbol{\theta}_{i-1})\|_2/\sqrt{R_0}$ and $b = C_{1b}\sqrt{R_0}/2$. Let $G_{i-1} := \|\nabla\mathcal{L}(\boldsymbol{\theta}_{i-1})\|_2$, then

$$\sum_{i=1}^k \beta_1^{k-i}\left(\frac{1}{R_0}G_{i-1}^2 - X_i\right) \geq \left(\frac{1}{R_0}G_{k-1}^2 - C_{1b}G_{k-1}\right) - \frac{R_0C_{1b}^2}{4}\sum_{i=1}^{k-1}\beta_1^{k-i}$$

$$\geq \frac{1}{R_0}G_{k-1}^2 - C_{1b}G_{k-1} - \frac{R_0C_{1b}^2}{4}\cdot\frac{\beta_1}{1-\beta_1}.$$

As long as $\frac{1}{R_0}G_{k-1}^2 - C_{1b}G_{k-1} - \frac{R_0C_{1b}^2}{4}\cdot\frac{\beta_1}{1-\beta_1} > 0$, a small $\eta$ can ensure the loss strictly decreases at this step $k$. Set

$$\bar{C} := \frac{R_0C_{1b}}{2}\left(1 + \sqrt{\frac{1}{1-\beta_1}}\right),$$

then any $G_{k-1} > \bar{C}$ meets this requirement. Moreover, $\bar{C}$ depends only on $(R_0, C_{1b}, \beta_1)$ and is independent of $\mathcal{L}$. This proves the corollary. $\qquad\square$

**Lemma F.4.** *Let $\mathcal{L}$ satisfy Assumption 3.2, and assume that $\mu$-PL condition is satisfied at all $\boldsymbol{\theta}_k$ where $k \geq 0$. Let $k \leq K = \mathcal{O}(\text{poly}(1/\eta))$ and let*

$$\psi : (\{0, 1, \cdots, k-1\} \times (0, 1)) \longrightarrow \mathbb{R}^+$$

*be a function. Let $\{X_i\}_{i=1}^k$ be any martingale difference sequence such that:*

1. *$X_i$ is $\mathcal{F}_i$-measurable and $\mathbb{E}_{i-1}[X_i] = 0$;*

2. *$|X_i| \leq C_{1b} \|\nabla \mathcal{L}(\boldsymbol{\theta}_{i-1})\|_2$ a.s.*

*for any $i \in [k]$. If for any $i \in [k]$ and $\delta \in (0, 1)$, it holds with probability $1 - \delta$ that*

$$\mathcal{L}(\boldsymbol{\theta}_{i-1}) - \mathcal{L}^* \leq \psi(i, \delta),$$

*then $\forall \delta \in (0, 1)$, with probability $1 - \delta$, we have $\mathcal{L}(\boldsymbol{\theta}_{i-1}) - \mathcal{L}^* \leq \psi\left(i, \frac{\delta}{2k}\right)$ for all $i \in [k]$, and that*

$$\left| \sum_{i=1}^k \gamma^{k-i} X_i \right| \leq C_4 \sqrt{\sum_{i=1}^k \gamma^{2k-2i} \psi\left(i, \frac{\delta}{2k}\right) \log \frac{4}{\delta}}$$

*for some constant $C_4$.*

**Remark F.1.** *The $\{X_i\}$ here may not necessarily equal the $\{X_i\}$ defined in Lemma F.1; we just make it general to benefit future steps. In fact, when we leverage this lemma later, we will multiply that of Lemma F.1 by some scalar $\in (0, 1)$.*

*Proof.* Note that $\sum_{i=1}^k \gamma^{k-i} X_i$ is a sum of martingale differences. Moreover, since $\mathcal{L}$ is $\rho$-smooth and $\exists C_{1b}$ s.t. every $|X_i|$ is bounded by $C_{1b} \|\nabla \mathcal{L}(\boldsymbol{\theta}_{i-1})\|_2$ (Lemma F.1), we have

$$
\begin{aligned}
|X_i| &\leq C_{1b} \|\nabla \mathcal{L}(\boldsymbol{\theta}_{i-1})\|_2 \\
&\leq C_{1b} \sqrt{2\rho \left(\mathcal{L}(\boldsymbol{\theta}_{i-1}) - \mathcal{L}^*\right)} \\
&\leq C_{1b} \sqrt{2\rho \psi(i, \delta')} \quad \text{if } \mathcal{L}(\boldsymbol{\theta}_{i-1}) - \mathcal{L}^* \leq \psi(i, \delta').
\end{aligned}
$$

Since $\mathcal{L}(\boldsymbol{\theta}_{i-1}) - \mathcal{L}^* \leq \psi(i, \delta')$ holds with probability $1 - \delta'$ instead of probability $1$, we create a new martingale difference sequence that masks out all the positions that exceed the bound. Specifically, we define $X'_{i, \delta'}$ as:

$$
X'_{i, \delta'} = \begin{cases} X_i & \text{if } \mathcal{L}(\boldsymbol{\theta}_{i-1}) - \mathcal{L}^* \leq \psi(i, \delta'), \\ 0 & \text{else.} \end{cases}
$$

This ensures that $\left| X'_{i, \delta'} \right| \leq C_{1b} \sqrt{2\rho \psi(i, \delta')}$ a.s. Then Azuma-Hoeffding's inequality gives us that for any $\epsilon'$,

$$\mathbb{P}\left[ \left| \sum_{i=1}^k \gamma^{k-i} X'_{i, \delta'} \right| \geq \epsilon' \right] \leq 2 \exp\left( \frac{-\epsilon'^2}{4 \sum_{i=1}^k C_{1b}^2 \gamma^{2k-2i} \rho \psi(i, \delta')} \right),$$

denoting the right hand side as $\frac{\delta}{2}$ gives that for any $\delta$, with probability $1 - \frac{\delta}{2}$,

$$\left| \sum_{i=1}^k \gamma^{k-i} X'_{i, \delta'} \right| \leq \sqrt{4 \sum_{i=1}^k C_{1b}^2 \gamma^{2k-2i} \rho \psi(i, \delta') \log \frac{4}{\delta}}.$$

Let $\delta' = \frac{\delta}{2k}$, by union bound, $\mathcal{L}(\boldsymbol{\theta}_{i-1}) - \mathcal{L}^* \leq \psi\left(i, \frac{\delta}{2k}\right)$ for all $i \in [k]$ with probability $1 - \frac{\delta}{2}$, which also implies $X'_{i, \delta'} = X_i$ for all $i \in [k]$. So with probabilty $1 - \delta$, the following two statements hold simultaneously for all $i \in [k]$:

$$\mathcal{L}(\boldsymbol{\theta}_{i-1}) - \mathcal{L}^* \leq \psi\left(i, \frac{\delta}{2k}\right)$$

and

$$\left| \sum_{i=1}^{k} \gamma^{k-i} X_i \right| \leq \sqrt{4 \sum_{i=1}^{k} C_{1b}^2 \gamma^{2k-2i} \rho \psi(i, \delta') \log \frac{4}{\delta}}$$

$$= C_4 \sqrt{\sum_{i=1}^{k} \gamma^{2k-2i} \psi\left(i, \frac{\delta}{2k}\right) \log \frac{4}{\delta}},$$

where $C_4 = 2C_{1b}\sqrt{\rho}$. $\square$

**Lemma F.5** (Convergence Bound of the AGM Framework). *Let $\mathcal{L}$ satisfy Assumption 3.2, and assume that $\mu$-PL condition is satisfied at all $\boldsymbol{\theta}_k$ where $k \geq 0$. Let $\eta$ be a small learning rate satisfying $\frac{\beta_1}{\gamma} = \beta_1/(1 - \frac{2\eta\mu(1-\beta_1)}{R_0}) \leq 0.95$. Let $K = \mathcal{O}(\text{poly}(1/\eta))$. Under mild restrictions on $K$, for any $k \leq K$, $\delta \in (0, 1)$, it holds with probability at least $1 - \delta$ that*

$$\mathcal{L}(\boldsymbol{\theta}_k) - \mathcal{L}^* \leq C_{5a} \cdot \gamma^k \left(\mathcal{L}(\boldsymbol{\theta}_0) - \mathcal{L}^*\right) + C_{5b} \cdot \eta \log \frac{K}{\delta}$$

*for some constants $C_{5a}$, $C_{5b}$.*

*Proof.* Denote $D_0 := \mathcal{L}(\boldsymbol{\theta}_0) - \mathcal{L}^*$, and denote the bound with $1 - \delta$ probability as $\psi(k, \delta) := \gamma^k C_{5a} D_0 + C_{5b}\eta \log \frac{K}{\delta}$, where the constants $C_{5a}, C_{5b}$ will be specified by us later. We prove by induction. When $k = 0$, the inequality

$$C_{5a} D_0 + C_{5b}\eta \log \frac{K}{\delta} \geq D_0$$

holds trivially as long as $C_{5a} \geq 1$. Now assume that the statement holds for $0, 1, \cdots, k-1$. From Lemma F.3, we have

$$\mathcal{L}(\boldsymbol{\theta}_k) - \mathcal{L}^* \leq \gamma^k \left(\mathcal{L}(\boldsymbol{\theta}_0) - \mathcal{L}^*\right) + \eta(1 - \beta_1) \sum_{i=1}^{k} X_i \sum_{j=i}^{k} \gamma^{k-j} \beta_1^{j-i} + C_3 \eta.$$

We can bound the coefficients by

$$\sum_{j=i}^{k} \gamma^{k-j} \beta_1^{j-i} = \gamma^{k-i} \sum_{j=0}^{k-i} \left(\frac{\beta_1}{\gamma}\right)^j$$

$$\leq \gamma^{k-i} \cdot \frac{1}{1 - \frac{\beta_1}{\gamma}}$$

$$\leq 20\gamma^{k-i},$$

where the last inequality is due to the assumption $\frac{\beta_1}{\gamma} \leq 0.95$ in the statement. Let $\tilde{X}_i := \frac{\sum_{j=i}^{k} \gamma^{k-j} \beta_1^{j-i}}{20\gamma^{k-i}} X_i$, then $\left\{\tilde{X}_i\right\}_{i=1}^{k}$ is also a martingale difference sequence and

$$\left| \tilde{X}_i \right| \leq |X_i| \leq C_{1b} \|\nabla \mathcal{L}(\boldsymbol{\theta}_i)\|_2 \text{ a.s.}$$

From Lemma F.4, with probability $1 - \delta$, $\mathcal{L}(\boldsymbol{\theta}_{i-1}) - \mathcal{L}^* \leq \psi\left(i, \frac{\delta}{2k}\right)$ holds for all $i \in [k]$ and it also holds that

$$\left| \sum_{i=1}^{k} \gamma^{k-i} \tilde{X}_i \right| \leq C_4 \sqrt{\sum_{i=1}^{k} \gamma^{2k-2i} \psi\left(i, \frac{\delta}{2k}\right) \log \frac{4}{\delta}}.$$

The above arguments give

$$\eta(1-\beta_1)\sum_{i=1}^{k} X_i \sum_{j=i}^{k} \gamma^{k-j}\beta_1^{j-i}$$

$$\leq 20 C_4 \eta (1-\beta_1)\sqrt{\sum_{i=1}^{k} \gamma^{2k-2i}\psi\left(i,\frac{\delta}{2k}\right)\log\frac{4}{\delta}}$$

$$\leq 20 C_4 \eta (1-\beta_1)\sqrt{\sum_{i=1}^{k} \gamma^{2k-2i}\left(\gamma^i C_{5a}D_0 + C_{5b}\eta\log\frac{2kK}{\delta}\right)\log\frac{4}{\delta}}$$

$$\leq 20 C_4 \eta (1-\beta_1)\sqrt{\sum_{i=1}^{k}\gamma^{2k-i}C_{5a}D_0 + \sum_{i=1}^{k}\gamma^{2k-2i}C_{5b}\eta\log\frac{2K^2}{\delta}}\cdot\sqrt{\log\frac{4}{\delta}}$$

$$\leq 20 C_4 \eta (1-\beta_1)\sqrt{\frac{\gamma^k C_{5a}D_0}{1-\gamma} + \frac{C_{5b}\eta}{1-\gamma^2}\log\frac{2K^2}{\delta}}\cdot\sqrt{\log\frac{4}{\delta}}$$

$$\leq 20 C_4 \eta (1-\beta_1)\left(\sqrt{\frac{\gamma^k C_{5a}D_0}{1-\gamma}} + \sqrt{\frac{C_{5b}\eta}{1-\gamma^2}\log\frac{2K^2}{\delta}}\right)\cdot\sqrt{\log\frac{4}{\delta}}.$$

As long as $K \geq \max\{2\delta^2,4\}$ (which is a mild restriction on $K$), we have $\log\frac{2K^2}{\delta}\log\frac{4}{\delta} \leq 3\log^2\frac{K}{\delta}$ and $\log\frac{4}{\delta} \leq \log\frac{K}{\delta}$. Plugging in $\frac{1}{1-\gamma} = \frac{R_0}{2\mu(1-\beta_1)}\cdot\frac{1}{\eta}$, we have

$$\eta(1-\beta_1)\sum_{i=1}^{k} X_i \sum_{j=i}^{k} \gamma^{k-j}\beta_1^{j-i}$$

$$\leq 20 C_4 (1-\beta_1)\left(\sqrt{\frac{C_{5a}R_0}{2\mu(1-\beta_1)}}\cdot\sqrt{\eta\gamma^k D_0\log\frac{K}{\delta}} + \eta\sqrt{\frac{3C_{5b}R_0}{2\mu(1-\beta_1)}}\log^2\frac{K}{\delta}\right)$$

$$\leq 10 C_4\sqrt{\frac{2C_{5a}R_0(1-\beta_1)}{\mu}}\cdot\sqrt{\eta\gamma^k D_0\log\frac{K}{\delta}} + 10 C_4\sqrt{\frac{6C_{5b}R_0(1-\beta_1)}{\mu}}\cdot\eta\log\frac{K}{\delta}$$

$$\leq C_{5c}\gamma^k D_0 + C_{5d}\eta\log\frac{K}{\delta},$$

where

$$C_{5c} = 5 C_4\sqrt{2C_{5a}R_0(1-\beta_1)/\mu},$$
$$C_{5d} = 5 C_4\sqrt{2C_{5a}R_0(1-\beta_1)/\mu} + 10 C_4\sqrt{6C_{5b}R_0(1-\beta_1)/\mu}.$$

Now as long as $K \geq e\delta$ (so that $\log\frac{K}{\delta}\geq 1$), we have

$$\mathcal{L}(\boldsymbol{\theta}_k) - \mathcal{L}^* \leq \gamma^k\left(\mathcal{L}(\boldsymbol{\theta}_0) - \mathcal{L}^*\right) + \eta(1-\beta_1)\sum_{i=1}^{k} X_i \sum_{j=i}^{k} \gamma^{k-j}\beta_1^{j-i} + C_3\eta$$

$$\leq \gamma^k D_0 + C_{5c}\gamma^k D_0 + C_{5d}\eta\log\frac{K}{\delta} + C_3\eta$$

$$\leq (C_{5c}+1)\gamma^k D_0 + (C_{5d}+C_3)\eta\log\frac{K}{\delta}.$$

To complete the induction, we need $C_{5a},C_{5b}$ satisfy

$$\begin{cases} C_{5a} \geq C_{5c}+1 & = 5 C_4\sqrt{\frac{2C_{5a}D_0 R_0(1-\beta_1)}{\mu}}+1, \\ C_{5b} \geq C_{5d}+C_3 & = 5 C_4\sqrt{\frac{2C_{5a}D_0 R_0(1-\beta_1)}{\mu}}+10 C_4\sqrt{\frac{6C_{5b}R_0(1-\beta_1)}{\mu}}+C_3. \end{cases}$$

Notice that the right-hand side grows at the rate of the square root of $C_{5a}$ and $C_{5b}$, so there must exist some feasible constants $C_{5a}$ and $C_{5b}$. Summarizing, under mild restrictions $K \geq \max\{2\delta^2, e\delta, 4\}$,

the statement $\mathcal{L}(\boldsymbol{\theta}_k) - \mathcal{L}^* \le \gamma^k C_{5a} D_0 + C_{5b}\eta \log \frac{K}{\delta}$ holds with probability $1 - \delta$, completing the induction. $\qquad\square$

**Remark F.2.** *The assumption in the statement, $\frac{\beta_1}{\gamma} \le 0.95$, is very mild since $\beta_1 \le 0.9$ (Assumption 4.2) and $1 - \gamma$ equals a constant multiple of $\eta$, so with small $\eta$ this condition is very easy to satisfy. Moreover, similar to Remark 4.1, the assumed threshold $0.95$ can be replaced by any constant below $1$, and the order of the convergence rate will remain unaffected.*

### F.1 Proof of Convergence-Related Conclusions in Appendix D

***Proof of Theorem D.2.*** This is a direct corollary following from Lemma F.5. The loss function $\mathcal{L}$ is global $\mu$-PL in this case. Setting $k = K$, letting $\gamma^K = \mathcal{O}(\eta)$ gives $K = \mathcal{O}\left(\frac{1}{\eta}\log\frac{1}{\eta}\right)$, completing the proof. $\qquad\square$

Now we move on to prove the first part of Theorem D.1. The main difficulty comes from the fact that $\mathcal{L}$ is only guaranteed to satisfy $\mu$-PL condition within some neighborhood $\Gamma^{\epsilon_3}$; The iteration, once getting out of that neighborhood, cannot be characterized. Hence we need to bound the probability of that event.

The trick here is to construct a proxy loss function $\tilde{\mathcal{L}}$ that, agrees with $\mathcal{L}$ near $\Gamma$ but has a "wall of quadratic functions" upon $\mathcal{L}$ further away. $\tilde{\mathcal{L}}$ thus satisfies $(\mu, \bar{L})$-PL which allows us to use Lemma F.5. If the losses at all steps are small, this ensures that the iteration never leaves $\Gamma^{\epsilon_1}$, where $\mathcal{L}$ and $\tilde{\mathcal{L}}$ are identical. We formalize this idea in the sequel.

**Lemma F.6** (Tubular Neighborhood Theorem; Theorem 6.24 in Lee [2012]). *Let $\Gamma \subset \mathbb{R}^d$ satisfy Assumption 3.4 (in particular, $\Gamma$ is a $\mathcal{C}^\infty$ compact embedded submanifold). Then there exists $\tau_\Gamma > 0$ such that, writing $N\Gamma$ for the normal bundle,*

$$V := \{(p, \boldsymbol{u}) \in N\Gamma : \|\boldsymbol{u}\|_2 < \tau_\Gamma\}, \qquad E : V \to \mathbb{R}^d, \quad E(p, \boldsymbol{u}) = p + \boldsymbol{u},$$

*the map $E$ is a diffeomorphism onto the open tube $U := \Gamma^{\tau_\Gamma}$. Consequently, the nearest–point projection $P : U \to \Gamma$ is well-defined and $\mathcal{C}^\infty$, and every $\boldsymbol{\theta} \in U$ can be written uniquely as*

$$\boldsymbol{\theta} = P(\boldsymbol{\theta}) + \nu(\boldsymbol{\theta}), \quad \nu(\boldsymbol{\theta}) \in N_{P(\boldsymbol{\theta})}\Gamma.$$

**Corollary F.2** (Smooth distance and unit normal on the tube). *In the setting of Lemma F.6, let $U := \Gamma^{\tau_\Gamma}$ and write $E^{-1}(\boldsymbol{\theta}) = (P(\boldsymbol{\theta}), \nu(\boldsymbol{\theta}))$ on $U$. Define*

$$r(\boldsymbol{\theta}) := \mathrm{dist}(\boldsymbol{\theta}, \Gamma) = \|\nu(\boldsymbol{\theta})\|_2, \qquad \boldsymbol{n}(\boldsymbol{\theta}) := \frac{\nu(\boldsymbol{\theta})}{\|\nu(\boldsymbol{\theta})\|_2} \quad (\boldsymbol{\theta} \in U \setminus \Gamma).$$

*Then $r$ and $\boldsymbol{n}$ are $\mathcal{C}^\infty$ on $U \setminus \Gamma$, and*

$$\nabla r(\boldsymbol{\theta}) = \boldsymbol{n}(\boldsymbol{\theta}), \qquad \forall \boldsymbol{\theta} \in U \setminus \Gamma. \tag{5}$$

*Proof.* By Lemma F.6, $P$ and $\nu$ are $\mathcal{C}^\infty$ on $U$, hence so are $r$ and $\boldsymbol{n}$ on $U \setminus \Gamma$. Set $g(\boldsymbol{\theta}) := r(\boldsymbol{\theta})^2 = \|\nu(\boldsymbol{\theta})\|_2^2$. By the chain rule,

$$\nabla g(\boldsymbol{\theta}) = 2\left(\nabla\nu(\boldsymbol{\theta})\right)^\top \nu(\boldsymbol{\theta}).$$

From the identity $\boldsymbol{\theta} = P(\boldsymbol{\theta}) + \nu(\boldsymbol{\theta})$ we have

$$\nabla\nu(\boldsymbol{\theta}) = I - \nabla P(\boldsymbol{\theta}).$$

Since $\nabla P(\boldsymbol{\theta})$ maps into $T_{P(\boldsymbol{\theta})}\Gamma$ and $\nu(\boldsymbol{\theta}) \in N_{P(\boldsymbol{\theta})}\Gamma$, it follows that

$$\left(\nabla P(\boldsymbol{\theta})\right)^\top \nu(\boldsymbol{\theta}) = \boldsymbol{0},$$

hence

$$\nabla g(\boldsymbol{\theta}) = 2\left(I - \nabla P(\boldsymbol{\theta})\right)^\top \nu(\boldsymbol{\theta}) = 2\,\nu(\boldsymbol{\theta}).$$

Therefore, for $\boldsymbol{\theta} \in U \setminus \Gamma$ (so $r(\boldsymbol{\theta}) > 0$),

$$\nabla r(\boldsymbol{\theta}) = \frac{1}{2\,r(\boldsymbol{\theta})}\nabla g(\boldsymbol{\theta}) = \frac{\nu(\boldsymbol{\theta})}{\|\nu(\boldsymbol{\theta})\|_2} = \boldsymbol{n}(\boldsymbol{\theta}),$$

which is Equation (5). $\qquad\square$

**Lemma F.7** (Nonobtuse angle between $\nabla\mathcal{L}$ and the outward normal). *Let $\Gamma$ satisfy Assumption 3.4 and let $\mathcal{L}$ satisfy Assumption 3.1 and Assumption 3.2. Then there exists a constant $\tau \in (0, \tau_\Gamma]$ such that, for all $\boldsymbol{\theta} \in \Gamma^\tau$ with nearest-point projection $P(\boldsymbol{\theta})$, distance $r(\boldsymbol{\theta}) := \|\boldsymbol{\theta} - P(\boldsymbol{\theta})\|_2$, and outward unit normal $\boldsymbol{n}(\boldsymbol{\theta}) := (\boldsymbol{\theta} - P(\boldsymbol{\theta}))/r(\boldsymbol{\theta})$, we have*

$$\langle \nabla\mathcal{L}(\boldsymbol{\theta}), \boldsymbol{n}(\boldsymbol{\theta}) \rangle \geq 0. \tag{6}$$

*In other words, $\angle\big(\nabla\mathcal{L}(\boldsymbol{\theta}), \boldsymbol{n}(\boldsymbol{\theta})\big) \leq \pi/2$ on $\Gamma^\tau \setminus \Gamma$.*

**Remark F.3.** *This lemma also implies that $t \mapsto \mathcal{L}\big(P(\boldsymbol{\theta}) + t\boldsymbol{n}(\boldsymbol{\theta})\big)$ is non-decreasing on $(0, \tau]$, since $\frac{d}{dt}\mathcal{L}(\boldsymbol{\phi} + t\boldsymbol{n}) = \langle \nabla\mathcal{L}(\boldsymbol{\phi} + t\boldsymbol{n}), \boldsymbol{n} \rangle \geq 0$ on $(0, \tau]$. To put it vividly, $\Gamma^\tau$ is a valley, with $\Gamma$ being the floor at the center of it.*

*Proof.* By Assumption 3.4, each $\boldsymbol{\zeta} \in \Gamma$ is a local minimizer of $\mathcal{L}$, hence $\nabla\mathcal{L}(\boldsymbol{\zeta}) = \boldsymbol{0}$, and $\nabla^2\mathcal{L}(\boldsymbol{\zeta})$ is positive definite on $N_{\boldsymbol{\zeta}}\Gamma$, with a uniform lower-bound $m > 0$ of its eigenvalues on $N_{\boldsymbol{\zeta}}\Gamma$ (from the compactness of $\Gamma$). Let $\tau_\Gamma$ be as in Lemma F.6. For $\boldsymbol{\theta} \in \Gamma^{\tau_\Gamma}$ write $\boldsymbol{\phi} := P(\boldsymbol{\theta})$, $r := \|\boldsymbol{\theta} - \boldsymbol{\phi}\|_2$, and $\boldsymbol{n} := (\boldsymbol{\theta} - \boldsymbol{\phi})/r \in N_{\boldsymbol{\phi}}\Gamma$. Since $\mathcal{L}$ is $\mathcal{C}^5$-smooth, we can perform a third order Taylor expansion:

$$\nabla\mathcal{L}(\boldsymbol{\theta}) = \nabla\mathcal{L}(\boldsymbol{\phi}) + \nabla^2\mathcal{L}(\boldsymbol{\phi})(\boldsymbol{\theta} - \boldsymbol{\phi}) + \boldsymbol{\theta}^{(3)}, \qquad \|\boldsymbol{\theta}^{(3)}\|_2 \leq C^{(3)}r^2,$$

where $C^{(3)}$ is a constant independent of $\boldsymbol{\theta}$. Taking the inner product with $\boldsymbol{n}$ and using $\nabla\mathcal{L}(\boldsymbol{\phi}) = \boldsymbol{0}$ and $\boldsymbol{\theta} - \boldsymbol{\phi} = r\boldsymbol{n}$, we have

$$\langle \nabla\mathcal{L}(\boldsymbol{\theta}), \boldsymbol{n} \rangle = r\boldsymbol{n}^\top\nabla^2\mathcal{L}(\boldsymbol{\phi})\boldsymbol{n} + \langle \boldsymbol{\theta}^{(3)}, \boldsymbol{n} \rangle \geq mr - C^{(3)}r^2.$$

Choose $\tau := \min\{\tau_\Gamma, m/C^{(3)}\}$, then for all $r \leq \tau$, we obtain Equation (6). $\qquad\square$

***Proof of the first part of Theorem D.1.*** First we construct a tubular neighborhood around $\Gamma$ and introduce some notations. By Assumption 3.4, $\Gamma$ is the unique set of minimizers of $\mathcal{L}$ in some neighborhood $U$ of $\Gamma$. By Lemma F.7, there is a $\tau \in (0, \tau_\Gamma]$ for which the nonobtuse condition $\langle \nabla\mathcal{L}(\boldsymbol{\theta}), \boldsymbol{n}(\boldsymbol{\theta}) \rangle \geq 0$ holds on $\Gamma^\tau \setminus \Gamma$. By Lemma E.3, there exists $\epsilon_3 > 0$ such that $\mathcal{L}$ is $\mu$-PL on $\Gamma^{\epsilon_3}$. Shrinking $\epsilon_3$ if necessary, assume $\epsilon_3 \leq \tau$ and $\Gamma^{\epsilon_3} \subseteq U$.

Throughout this proof we work inside the tube $\Gamma^{\tau_\Gamma}$ given by Lemma F.6. In particular, for every $\boldsymbol{\theta} \in \Gamma^{\tau_\Gamma}$ we have the nearest–point projection $P(\boldsymbol{\theta}) \in \Gamma$, the normal offset $\nu(\boldsymbol{\theta}) \in N_{P(\boldsymbol{\theta})}\Gamma$, the distance and unit normal

$$r(\boldsymbol{\theta}) := \mathrm{dist}(\boldsymbol{\theta}, \Gamma) = \|\nu(\boldsymbol{\theta})\|_2, \qquad \boldsymbol{n}(\boldsymbol{\theta}) := \frac{\nu(\boldsymbol{\theta})}{\|\nu(\boldsymbol{\theta})\|_2},$$

and (on $U \setminus \Gamma$) the identity $\nabla r(\boldsymbol{\theta}) = \boldsymbol{n}(\boldsymbol{\theta})$ from Corollary F.2.

Next, recall the constant $\epsilon_1$ constructed in Lemma E.3 such that $0 < \epsilon_1 < \epsilon_3$. Define the "gap level"

$$\mathcal{L}_m := \min\big\{\inf\big\{\mathcal{L}(\boldsymbol{\theta}) : r(\boldsymbol{\theta}) \in [\epsilon_1, \epsilon_3]\big\}, \bar{L}\big\}.$$

The set $\{\boldsymbol{\theta} : r(\boldsymbol{\theta}) \in [\epsilon_1, \epsilon_3]\}$ is compact and disjoint from $\Gamma$, hence $\mathcal{L}_m > \mathcal{L}^*$.

Then we define the proxy objective $\tilde{\mathcal{L}} : \mathbb{R}^d \to \mathbb{R}$ by

$$\tilde{\mathcal{L}}(\boldsymbol{\theta}) := \begin{cases} \mathcal{L}(\boldsymbol{\theta}), & \mathrm{dist}(\boldsymbol{\theta}, \Gamma) \leq \epsilon_1, \\ \mathcal{L}(\boldsymbol{\theta}) + \dfrac{C}{2}\big(\mathrm{dist}(\boldsymbol{\theta}, \Gamma) - \epsilon_1\big)^2, & \mathrm{dist}(\boldsymbol{\theta}, \Gamma) > \epsilon_1, \end{cases}$$

where $C$ is a large constant satisfying $C \geq \mu$. Note that for $\boldsymbol{\theta} \in \mathbb{R}^d$,

$$\tilde{\mathcal{L}}(\boldsymbol{\theta}) < \mathcal{L}_m \implies \mathrm{dist}(\boldsymbol{\theta}) < \epsilon_1,$$

so on the sublevel set $\{\tilde{\mathcal{L}} < \mathcal{L}_m\}$ we have $\tilde{\mathcal{L}} = \mathcal{L}$.

Now define

$$\bar{L} := \frac{C}{2}(\epsilon_3 - \epsilon_1)^2 + \mathcal{L}^*,$$

and we prove the core property of the proxy loss function: $\tilde{\mathcal{L}}$ is $(\mu, \bar{L})$-PL. First we consider the case $\boldsymbol{\theta} \in \Gamma^{\epsilon_3}$. On $\Gamma^{\epsilon_3}$, the distance function $r(\boldsymbol{\theta}) = \mathrm{dist}(\boldsymbol{\theta}, \Gamma)$ is defined. Using $\nabla r(\boldsymbol{\theta}) = \boldsymbol{n}(\boldsymbol{\theta})$ from Corollary F.2 and the nonobtuse condition from Lemma F.7, for $r(\boldsymbol{\theta}) > \epsilon_1$,

$$\nabla \tilde{\mathcal{L}}(\boldsymbol{\theta}) = \nabla \mathcal{L}(\boldsymbol{\theta}) + C\big(r(\boldsymbol{\theta}) - \epsilon_1\big)\, \boldsymbol{n}(\boldsymbol{\theta}),$$

$$\|\nabla \tilde{\mathcal{L}}(\boldsymbol{\theta})\|_2^2 = \|\nabla \mathcal{L}(\boldsymbol{\theta})\|_2^2 + 2C\big(r(\boldsymbol{\theta}) - \epsilon_1\big) \langle \nabla \mathcal{L}(\boldsymbol{\theta}), \boldsymbol{n}(\boldsymbol{\theta}) \rangle + C^2 \big(r(\boldsymbol{\theta}) - \epsilon_1\big)^2$$

$$\geq \|\nabla \mathcal{L}(\boldsymbol{\theta})\|_2^2 + C^2 \big(r(\boldsymbol{\theta}) - \epsilon_1\big)^2.$$

Since $\mathcal{L}$ is $\mu$-PL on $\Gamma^{\epsilon_3}$,

$$\|\nabla \tilde{\mathcal{L}}(\boldsymbol{\theta})\|_2^2 \geq 2\mu\big(\mathcal{L}(\boldsymbol{\theta}) - \mathcal{L}^*\big) + 2C \cdot \frac{C}{2}\big(r(\boldsymbol{\theta}) - \epsilon_1\big)^2 \geq 2\min\{\mu, C\}\big(\tilde{\mathcal{L}}(\boldsymbol{\theta}) - \mathcal{L}^*\big).$$

For $r(\boldsymbol{\theta}) \leq \epsilon_1$, $\tilde{\mathcal{L}} = \mathcal{L}$ and the $\mu$-PL inequality holds trivially. Our choice of $C$ yields

$$\|\nabla \tilde{\mathcal{L}}(\boldsymbol{\theta})\|_2^2 \geq 2\mu\big(\tilde{\mathcal{L}}(\boldsymbol{\theta}) - \mathcal{L}^*\big) \qquad \text{for all } \boldsymbol{\theta} \in \Gamma^{\epsilon_3},$$

i.e., $\tilde{\mathcal{L}}$ is $\mu$-PL on $\Gamma^{\epsilon_3}$.

Combining with the fact that $\tilde{\mathcal{L}}(\boldsymbol{\theta}) > \bar{L}$ if $\boldsymbol{\theta} \notin \Gamma^{\epsilon_3}$, we conclude that $\tilde{\mathcal{L}}$ is $(\mu, \bar{L})$-PL.

Finally we come back to prove the main conclusion. Let $\bar{C}$ be the constant constructed in Corollary F.1. Note that for $\boldsymbol{\theta} \in \mathbb{R}^d \setminus \Gamma^{\epsilon_1}$,

$$\|\nabla \tilde{\mathcal{L}}(\boldsymbol{\theta})\|_2 \leq \bar{C} \Rightarrow \|\nabla \mathcal{L}(\boldsymbol{\theta}) + C\big(r(\boldsymbol{\theta}) - \epsilon_1\big)\, \boldsymbol{n}(\boldsymbol{\theta})\|_2 \leq \bar{C}$$

$$\Rightarrow \|C\big(r(\boldsymbol{\theta}) - \epsilon_1\big)\, \boldsymbol{n}(\boldsymbol{\theta})\|_2 \leq \bar{C} \quad \text{(from Lemma F.7)}$$

$$\Rightarrow r(\boldsymbol{\theta}) \leq \frac{\bar{C}}{C} + \epsilon_1.$$

Increasing $C$ if necessary, we can let $\bar{C}/C < (\epsilon_3 - \epsilon_1)/2$, which implies

$$\epsilon_m := \frac{\bar{C}}{C} + \epsilon_1 < \frac{\epsilon_1 + \epsilon_3}{2}.$$

Take some $\mathcal{L}_0 > \mathcal{L}^*$ such that $\mathcal{L}_0 - \mathcal{L}^* < (\mathcal{L}_m - \mathcal{L}^*)/2C_{5a}$, where $C_{5a}$ is the constant in Lemma F.5. Take a constant $\epsilon_0 < \epsilon_1$ such that all loss values inside $\Gamma_0^{\epsilon}$ are bounded by $\mathcal{L}_0$. There exists a sufficiently small learning rate $\eta$ such that (let $K = \lfloor (T+1)\eta^{-2} \rfloor$):

1. $C_{5a}\gamma^k \left(\mathcal{L}_0 - \mathcal{L}^*\right) + C_{5b}\eta \log \frac{K}{\delta} \leq 0.99\mathcal{L}_m - \mathcal{L}^*, \forall k \leq K, \delta \in (\eta^{200}, 1)$.

2. $\eta \cdot R \cdot R_2 < \epsilon_3 - \epsilon_m$.

The second property ensures that any single step of update cannot jump from the interior of $\Gamma^{\epsilon_m}$ to the exterior of $\Gamma^{\epsilon_3}$. However when $\boldsymbol{\theta}_{k-1} \in \Gamma^{\epsilon_3} \setminus \Gamma^{\epsilon_m}$, it follows that $\|\nabla \tilde{\mathcal{L}}(\boldsymbol{\theta})\|_2 > \bar{C}$, so $\mathcal{L}(\boldsymbol{\theta}_k) < \mathcal{L}(\boldsymbol{\theta}_{k-1})$ from Corollary F.1. By induction, we conclude that for any $\boldsymbol{\theta}_0 \in \Gamma^{\epsilon_0}$, if we launch an AGM from $\boldsymbol{\theta}_0$ and train using $\tilde{\mathcal{L}}$ and $\eta$, all loss values $\tilde{\mathcal{L}}(\boldsymbol{\theta}_k), k \in [0, K-1]$ do not exceed $\bar{L}$, which means the $\mu$-PL condition of $\tilde{\mathcal{L}}$ is satisfied at all $\boldsymbol{\theta}_k$ where $k \in [0, K-1]$. This meets the requirement of Lemma F.5.

By Lemma F.5 and the first property of $\eta$, we further conclude that all loss values $\tilde{\mathcal{L}}(\boldsymbol{\theta}_k)$ do not exceed $0.99\mathcal{L}_m$. Finally, by noting that

$$\tilde{\mathcal{L}}(\boldsymbol{\theta}) < \mathcal{L}_m \Rightarrow \boldsymbol{\theta} \in \Gamma^{\epsilon_1} \text{ and } \tilde{\mathcal{L}}(\boldsymbol{\theta}) = \mathcal{L}(\boldsymbol{\theta}),$$

we conclude from Lemma F.5 that: For any $\boldsymbol{\theta}_0 \in \Gamma^{\epsilon_0}$, if we launch an AGM from $\boldsymbol{\theta}_0$ and train using $\mathcal{L}$ and $\eta$, for any $k \leq K, \delta \in (\eta^{200}, 1)$, it holds almost surely that $\boldsymbol{\theta}_k \in \Gamma^{\epsilon_1}$, and it holds with probability at least $1 - \delta$ that

$$\mathcal{L}(\boldsymbol{\theta}_k) - \mathcal{L}^* \leq C_{5a} \cdot \gamma^k \left(\mathcal{L}(\boldsymbol{\theta}_0) - \mathcal{L}^*\right) + C_{5b} \cdot \eta \log \frac{K}{\delta},$$

and it takes $K_0 = \mathcal{O}(\frac{1}{\eta} \log \frac{1}{\eta})$ time to reach $\mathcal{L}(K_0) - \mathcal{L}^* = \mathcal{O}\left(\eta \log \frac{1}{\eta\delta}\right)$, completing the proof. $\quad \square$

# G  Proof of the SDE Approximation of AGMs

In this section, we present a detailed derivation of our slow SDE approximation of the AGM framework as formally stated in Theorem D.1. In Appendix G.1, we give some lemmas about the properties of the adaptive projection $\Phi_{\boldsymbol{S}}$, introduced and explained in Section 4. Then in Appendix G.2, we show that the iterations after convergence would continue staying near the manifold. Following that, we further calculate the moment change of the projected parameter $\Phi_{\boldsymbol{S}_t}(\boldsymbol{\theta}_t)$ during a "giant step" in Appendix G.3, which allows us to conduct an weak approximation in Appendix G.4. After going through the above paradigm, we finally prove an equivalent Theorem G.2 of the SDE approximation part of Theorem D.1.

Our slow SDE starts at a point after the time of convergence, so for simplicity, we will "shift the timeline" in this section.

**Remark G.1** (Time Shift). *To simplify the notations, we redefine $\boldsymbol{\theta}_0$ and $\boldsymbol{v}_0$ as follows. Starting from Appendix G.1, $\boldsymbol{\theta}_0$ and $\boldsymbol{v}_0$ will no longer represent the parameters that are initialized at the actual beginning of training. Instead, they represent the $\boldsymbol{\theta}_{K_0}$ and $\boldsymbol{v}_{K_0}$ yielded by the first part of Theorem D.1, $(\boldsymbol{\theta}_1, \boldsymbol{v}_1)$ denoting $(\boldsymbol{\theta}_{K_0+1}, \boldsymbol{v}_{K_0+1})$, and so on. Our SDE approximation then describes the dynamics of AGMs after reaching the state $(\boldsymbol{\theta}_0, \boldsymbol{v}_0)$.*

**Remark G.2.** *Recall for any time step $k$ that $\boldsymbol{\phi}_k := \Phi_{\boldsymbol{S}(\boldsymbol{v}_k)}(\boldsymbol{\theta}_k)$. With the "time shift" described in Remark G.1, the time steps before $K_0$ will become negative. However in some parts of the following calculation, to deal with the first-order momentum we still need up to $\mathcal{O}(\log \frac{1}{\eta})$ past timesteps. Without loss of generality, we assume that at time $K_0$ the iteration has already converged for time $\mathcal{O}(\log \frac{1}{\eta})$, i.e., $\forall K_0 - \mathcal{O}(\log \frac{1}{\eta}) \leq k \leq K_0$,*

$$\mathcal{L}(\boldsymbol{\theta}_k) - \mathcal{L}^* = \mathcal{O}\left(\eta \log \frac{1}{\eta}\right),$$

$$\|\boldsymbol{\theta}_k - \boldsymbol{\phi}_k\|_2 = \mathcal{O}\left(\sqrt{\eta \log \frac{1}{\eta}}\right),$$

$$\|\nabla \mathcal{L}(\boldsymbol{\theta}_k)\|_2 = \mathcal{O}\left(\sqrt{\eta \log \frac{1}{\eta}}\right).$$

*If not, we simply increase $K_0$ by $\mathcal{O}(\log \frac{1}{\eta})$ and the argument in the proof of the first part of Theorem D.1 still holds. After the time shift, the range of $k$ above will become $-\mathcal{O}(\log \frac{1}{\eta}) \leq k \leq 0$. Therefore, negative timesteps may appear in the derivation below and they are not typos; We just need their three properties above to control the order of some terms.*

## G.1  Lemmas for Adaptive Manifold Projection

Before we characterize the projections, we introduce some properties of the preconditioned projection function in this part.

**Lemma G.1** (Adaption of Lemma C.2 in Li et al. [2021b]). *For any $\boldsymbol{x} \in \mathbb{R}^d$, and any p.d matrix $\boldsymbol{S} \in \mathbb{R}^{d \times d}$, it holds that $\partial \Phi_{\boldsymbol{S}}(\boldsymbol{x}) \boldsymbol{S} \nabla \mathcal{L}(\boldsymbol{x}) = 0$, and*

$$\partial^2 \Phi_{\boldsymbol{S}}(\boldsymbol{x})[\boldsymbol{S} \nabla \mathcal{L}(\boldsymbol{x}), \boldsymbol{S} \nabla \mathcal{L}(\boldsymbol{x})] = -\partial \Phi_{\boldsymbol{S}}(\boldsymbol{x}) \boldsymbol{S} \nabla^2 \mathcal{L}(\boldsymbol{x}) \boldsymbol{S} \nabla \mathcal{L}(\boldsymbol{x}).$$

*Proof.* We consider a trajectory starting from $\boldsymbol{x}(0) = \boldsymbol{x}$, with an ODE $\frac{\mathrm{d}\boldsymbol{x}(t)}{\mathrm{d}t} = -\boldsymbol{S} \nabla \mathcal{L}(\boldsymbol{x}(t))$, thus by the definition of $\Phi_{\boldsymbol{S}}$, we have $\Phi_{\boldsymbol{S}}(\boldsymbol{x}) = \Phi_{\boldsymbol{S}}(\boldsymbol{x}(t))$, then we have

$$\frac{\mathrm{d}\Phi_{\boldsymbol{S}}(\boldsymbol{x}(t))}{\mathrm{d}t} = -\partial \Phi_{\boldsymbol{S}}(\boldsymbol{x}) \boldsymbol{S} \nabla \mathcal{L}(\boldsymbol{x}) = 0.$$

Further, we take the second derivative of $\Phi_{\boldsymbol{S}}(\boldsymbol{x}(t))$ with repsect to $t$

$$\frac{\mathrm{d}^2 \Phi_{\boldsymbol{S}}(\boldsymbol{x}(t))}{\mathrm{d}t^2} = \partial^2 \Phi_{\boldsymbol{S}}(\boldsymbol{x})[\boldsymbol{S} \nabla \mathcal{L}(\boldsymbol{x}), \boldsymbol{S} \nabla \mathcal{L}(\boldsymbol{x})] + \partial \Phi_{\boldsymbol{S}}(\boldsymbol{x}) \boldsymbol{S} \nabla^2 \mathcal{L}(\boldsymbol{x}) \boldsymbol{S} \nabla \mathcal{L}(\boldsymbol{x}) = 0.$$

Taking $t = 0$ completes the proof. $\qquad\square$

**Lemma G.2.** *For any $\boldsymbol{x} \in \Gamma$, and a p.d matrix $\boldsymbol{S}$, the following two identities hold:*

- *For all $\boldsymbol{v} \in T_{\boldsymbol{x}}(\Gamma)$, $\partial \Phi_{\boldsymbol{S}}(x)\boldsymbol{v} = \boldsymbol{v}$.*

- *For all $\boldsymbol{u} \in T_{\boldsymbol{x}}^{\perp}(\Gamma)$, $\partial \Phi_{\boldsymbol{S}}(\boldsymbol{x})\boldsymbol{S}\boldsymbol{u} = \boldsymbol{0}$.*

*Proof.* We first prove the first identity. For any $\boldsymbol{v} \in T_{\boldsymbol{x}}(\Gamma)$, let $\{\boldsymbol{v}(t), t \geq 0\}$ be parameterized smooth curve on $\Gamma$ such that $\boldsymbol{v}(0) = \boldsymbol{x}$ and $\frac{\mathrm{d}\boldsymbol{v}(t)}{\mathrm{d}t}\big|_{t=0} = \boldsymbol{v}$. Since $\boldsymbol{v}(t) \in \Gamma$, thus $\nabla \mathcal{L}(\boldsymbol{v}(t)) = 0$, which gives $\Phi_{\boldsymbol{S}}(\boldsymbol{v}(t)) = \boldsymbol{v}(t)$. Thus we have

$$\frac{\mathrm{d}\boldsymbol{v}(t)}{\mathrm{d}t}\bigg|_{t=0} = \frac{\mathrm{d}\Phi_{\boldsymbol{S}}(\boldsymbol{v}(t))}{\mathrm{d}t}\bigg|_{t=0} = \partial\Phi_{\boldsymbol{S}}(\boldsymbol{v}(t))\frac{\mathrm{d}\boldsymbol{v}(t)}{\mathrm{d}t}\bigg|_{t=0}.$$

Plugging $\frac{\mathrm{d}\boldsymbol{v}(t)}{\mathrm{d}t}\big|_{t=0} = \boldsymbol{v}$ gives $\Phi_{\boldsymbol{S}}(\boldsymbol{x})\boldsymbol{v} = \boldsymbol{v}$.

For $\boldsymbol{u} \in T_{\boldsymbol{x}}^{\perp}(\Gamma)$ and $t \geq 0$, we consider the Taylor expansion of $\nabla\mathcal{L}(\boldsymbol{x} + t\nabla^2\mathcal{L}(\boldsymbol{x})^{\dagger}\boldsymbol{u})$ at $t = 0$:

$$\nabla\mathcal{L}\left(\boldsymbol{x} + t\nabla^2\mathcal{L}(\boldsymbol{x})^{\dagger}\boldsymbol{u}\right) = \nabla^2\mathcal{L}(\boldsymbol{x}) \cdot t\nabla^2\mathcal{L}(\boldsymbol{x})^{\dagger}\boldsymbol{u} + o(t) = t\boldsymbol{u} + o(t),$$

where the second equation uses the fact $\nabla^2\mathcal{L}(x)$ is full-rank on $T_{\boldsymbol{x}}^{\perp}(\Gamma)$. Thus, by the continuity of $\partial\Phi_{\boldsymbol{S}}$ proved in Lemma E.3, it follows that

$$\lim_{t \to 0} \frac{\partial\Phi_{\boldsymbol{S}}\left(\boldsymbol{x} + t\nabla^2\mathcal{L}(\boldsymbol{x})^{\dagger}\boldsymbol{u}\right)\boldsymbol{S}\nabla\mathcal{L}\left(\boldsymbol{x} + t\nabla^2\mathcal{L}(\boldsymbol{x})^{\dagger}\boldsymbol{u}\right)}{t} = \partial\Phi_{\boldsymbol{S}}(\boldsymbol{x})\boldsymbol{S}\boldsymbol{u}.$$

By Lemma G.1, $\partial\Phi_{\boldsymbol{S}}\left(\boldsymbol{x} + t\nabla^2\mathcal{L}(\boldsymbol{x})^{\dagger}\boldsymbol{u}\right)\boldsymbol{S}\nabla\mathcal{L}\left(\boldsymbol{x} + t\nabla^2\mathcal{L}(\boldsymbol{x})^{\dagger}\boldsymbol{u}\right) = 0$ for all $t \geq 0$, implying that $\partial\Phi_{\boldsymbol{S}}(\boldsymbol{x})\boldsymbol{S}\boldsymbol{u} = 0$ for all $\boldsymbol{u} \in T_{\boldsymbol{x}}^{\perp}(\Gamma)$. $\qquad\square$

**Lemma G.3.** *For any $\boldsymbol{x} \in \Gamma$, and a p.d matrix $\boldsymbol{S}$, it holds that $\partial\Phi_{\boldsymbol{S}}(\boldsymbol{x})\boldsymbol{S}\nabla^2\mathcal{L}(\boldsymbol{x}) = \boldsymbol{0}$.*

*Proof.* From Lemma C.1 in Li et al. [2021b], we have for $\boldsymbol{u} \in T_{\boldsymbol{x}}(\Gamma)$, $\nabla^2\mathcal{L}(\boldsymbol{x})\boldsymbol{u} = 0$, and for $\boldsymbol{u} \in T_{\boldsymbol{x}}^{\perp}(\Gamma)$, Lemma G.2 gives that

$$\partial\Phi_{\boldsymbol{S}}(\boldsymbol{x})\boldsymbol{S}\boldsymbol{u} = \boldsymbol{0}.$$

The above identity completes the proof. $\qquad\square$

**Lemma G.4.** *For any $\boldsymbol{x} \in \Gamma$, $\boldsymbol{u}, \boldsymbol{v} \in \mathbb{R}^d$, p.d matrix $\boldsymbol{S}$, and $v \in T_x(\Gamma)$, it holds that*

$$\partial^2\Phi_{\boldsymbol{S}}(\boldsymbol{x})[\boldsymbol{u}\boldsymbol{v}^{\top}] = -\partial\Phi_{\boldsymbol{S}}(\boldsymbol{x})\boldsymbol{S}\partial^2(\nabla\mathcal{L})(\boldsymbol{x})[\nabla^2\mathcal{L}(\boldsymbol{x})^{\dagger}\boldsymbol{S}^{-1}\boldsymbol{u}\boldsymbol{v}^{\top}] - \boldsymbol{S}^{-1}\nabla^2\mathcal{L}(\boldsymbol{x})^{\dagger}\partial^2(\nabla\mathcal{L})(\boldsymbol{x})[\boldsymbol{S}\partial\Phi(\boldsymbol{x})\boldsymbol{u}\boldsymbol{v}^{\top}].$$

*Proof.* We define $\boldsymbol{P} := \boldsymbol{S}^{1/2}$. And we do a reparameterization as $\boldsymbol{x}' := \boldsymbol{P}^{-1}\boldsymbol{x}$, $\mathcal{L}'(\boldsymbol{x}) := \mathcal{L}(\boldsymbol{P}\boldsymbol{x})$, then we have

$$\partial\Phi'(\boldsymbol{x}') = \boldsymbol{P}\partial\Phi_{\boldsymbol{S}}(\boldsymbol{P}\boldsymbol{x})\boldsymbol{P}$$
$$\nabla^2 L'(\boldsymbol{x}') = \boldsymbol{P}\nabla^2 L(\boldsymbol{P}\boldsymbol{x})\boldsymbol{P}$$
$$\partial^2(\nabla L')(\boldsymbol{x}')[\boldsymbol{M}] = \boldsymbol{P}\partial^2(\nabla L)(\boldsymbol{P}\boldsymbol{x})[\boldsymbol{P}\boldsymbol{M}\boldsymbol{P}]$$
$$\partial^2\Phi'(\boldsymbol{x}')[\boldsymbol{M}] = \boldsymbol{P}\partial^2\Phi(\boldsymbol{x})[\boldsymbol{P}\boldsymbol{M}\boldsymbol{P}].$$

Notice that in the space of $\boldsymbol{x}'$, the adaptive projection mapping $\Phi_{\boldsymbol{S}}$ turns into a fixed gradient flow projection. And this allows us to directly apply Lemma C.4 in Li et al. [2021b], which gives

$$\partial^2\Phi'(x')[\boldsymbol{v}, \boldsymbol{u}] = -\partial\Phi'(x')\partial^2(\nabla\mathcal{L}')(x')[v, \nabla^2\mathcal{L}'(x')^{\dagger}\boldsymbol{u}] - \nabla^2\mathcal{L}'(x')^{\dagger}\partial^2(\nabla\mathcal{L}')(x')[v, \partial\Phi'(x')\boldsymbol{u}].$$

A slight modification using the above transformations gives

$$\partial^2\Phi_{\boldsymbol{S}}(\boldsymbol{x})[\boldsymbol{P}\boldsymbol{v}, \boldsymbol{P}\boldsymbol{u}] = -\partial\Phi_{\boldsymbol{S}}(\boldsymbol{x})\boldsymbol{S}\partial^2(\nabla\mathcal{L})(\boldsymbol{x})[\boldsymbol{P}\boldsymbol{v}, \nabla^2\mathcal{L}(\boldsymbol{x})^{\dagger}\boldsymbol{S}^{-1}\boldsymbol{P}\boldsymbol{u}]$$
$$- \boldsymbol{S}^{-1}\nabla^2\mathcal{L}(\boldsymbol{x})^{\dagger}\partial^2(\nabla\mathcal{L})(\boldsymbol{x})[\boldsymbol{P}\boldsymbol{v}, \boldsymbol{S}\partial\Phi(\boldsymbol{x})\boldsymbol{P}\boldsymbol{u}].$$

We now redefine $\boldsymbol{u} = \boldsymbol{P}\boldsymbol{u}$, $\boldsymbol{v} = \boldsymbol{P}\boldsymbol{v}$, and we organize the above equation

$$\partial^2\Phi_{\boldsymbol{S}}(\boldsymbol{x})[\boldsymbol{u}\boldsymbol{v}^{\top}] = -\partial\Phi_{\boldsymbol{S}}(\boldsymbol{x})\boldsymbol{S}\partial^2(\nabla\mathcal{L})(\boldsymbol{x})[\nabla^2\mathcal{L}(\boldsymbol{x})^{\dagger}\boldsymbol{S}^{-1}\boldsymbol{u}\boldsymbol{v}^{\top}]$$
$$- \boldsymbol{S}^{-1}\nabla^2\mathcal{L}(\boldsymbol{x})^{\dagger}\partial^2(\nabla\mathcal{L})(\boldsymbol{x})[\boldsymbol{S}\partial\Phi(\boldsymbol{x})\boldsymbol{u}\boldsymbol{v}^{\top}].$$

We completes the proof. $\qquad\square$

## G.2 Iteration Stays Near Manifold

Now we begin the final preparations before deriving the slow SDE near the manifold. Note that in the end of convergence analysis, the total steps equal $K = \lfloor (T+1)\eta^{-2} \rfloor$ and the converging step $K_0 = \mathcal{O}(\frac{1}{\eta}\log\frac{1}{\eta})$. So after time shifting, the high probability convergence of $\lfloor (T+1)\eta^{-2} \rfloor - \mathcal{O}(\frac{1}{\eta}\log\frac{1}{\eta}) > \lfloor T\eta^{-2} \rfloor$ steps are ensured in Lemma F.5. Now denote $K := \lfloor T\eta^{-2} \rfloor$ be the total number of steps in our analysis. Let $\beta$ be some constant in $(0, 0.5)$, whose exact value will be specified later. First, we bound the movement of projected steps by showing that $\phi$ shifts no more than $\tilde{\mathcal{O}}(\eta^{0.5-0.5\beta})$ within $\Delta K := \lfloor \eta^{-1-\beta} \rfloor$ steps, demonstrating the "slowness" of the dynamics of AGMs after the projection.

**Lemma G.5.** *For any $\delta = \mathcal{O}(\mathrm{poly}(\eta))$, with probability $1-\delta$, for any $k \in [0, K-\Delta K]$, $\Delta k \in [\Delta K]$,*

$$\|\phi_{k+\Delta k} - \phi_k\|_2 \leq C_6 \eta^{0.5-0.5\beta}\sqrt{\log\frac{1}{\eta\delta}}$$

*for some constant $C_6$.*

*Proof.* Recall from the proof of the first part of Theorem D.1 that $\boldsymbol{\theta}_k$ stays inside $\Gamma^{\epsilon_1}$ For any $k \in [0, K]$ almost surely, and from Lemma E.3 that $\Phi_{S(\boldsymbol{v})}(\boldsymbol{\theta})$ is $\mathcal{C}^4$ on $\mathcal{X}^{\epsilon_1} := \Gamma^{\epsilon_1} \times \mathbb{R}^D_{[0,R_1]}$. Since $\mathcal{X}^{\epsilon_1}$ is compact, $\Phi_{S(\boldsymbol{v})}(\boldsymbol{\theta})$ is then bounded and Lipschitz on $\mathcal{X}^{\epsilon_1}$. Similarly, $\partial\Phi_{S(\boldsymbol{v})}(\boldsymbol{\theta})$ is bounded and Lipschitz on $\mathcal{X}^{\epsilon_1}$. For any $k \in [0, K)$, let $\bar{k} = k - 2\log_{\beta_1}\eta$, we have:

$$\phi_{k+1} - \phi_k = \Phi_{S(\boldsymbol{v}_{k+1})}(\boldsymbol{\theta}_{k+1}) - \Phi_{S(\boldsymbol{v}_k)}(\boldsymbol{\theta}_k)$$

$$= \Phi_{S(\boldsymbol{v}_{\bar{k}})}(\boldsymbol{\theta}_{k+1}) - \Phi_{S(\boldsymbol{v}_{\bar{k}})}(\boldsymbol{\theta}_k) + \mathcal{O}\left(\eta^2\log\frac{1}{\eta}\right)$$

$$= \partial\Phi_{S(\boldsymbol{v}_{\bar{k}})}(\boldsymbol{\theta}_k)(\boldsymbol{\theta}_{k+1} - \boldsymbol{\theta}_k) + \mathcal{O}\left(\eta^2\log\frac{1}{\eta}\right)$$

$$= \partial\Phi_{S(\boldsymbol{v}_{\bar{k}})}(\boldsymbol{\theta}_k)(\eta S(\boldsymbol{v}_{k+1})\boldsymbol{m}_{k+1}) + \mathcal{O}\left(\eta^2\log\frac{1}{\eta}\right)$$

$$= \partial\Phi_{S(\boldsymbol{v}_{\bar{k}})}(\boldsymbol{\theta}_{\bar{k}})(\eta S(\boldsymbol{v}_{\bar{k}})\boldsymbol{m}_{k+1}) + \mathcal{O}\left(\eta^2\log\frac{1}{\eta}\right),$$

where the second equality comes from the fact that one step of update on $\boldsymbol{v}$ is of $\mathcal{O}(\eta^2)$ and the Lipschitzness of $S$ and $\Phi$, the third equality comes from $\|\boldsymbol{\theta}_{k+1} - \boldsymbol{\theta}_k\|_2 = \mathcal{O}(\eta)$, and the last equality follows from the boundedness and Lipschitzness of $\partial\Phi$. We can decompose $\boldsymbol{m}_k$ as:

$$\boldsymbol{m}_{k+1} = (1-\beta_1)\sum_{i=\bar{k}}^{k}\beta_1^{k-i}(\nabla\mathcal{L}(\boldsymbol{\theta}_i) + \boldsymbol{z}_i) + \mathcal{O}(\eta^2)$$

$$= (1-\beta_1)\sum_{i=\bar{k}}^{k}\beta_1^{k-i}\left(\nabla\mathcal{L}(\boldsymbol{\theta}_{\bar{k}}) + \mathcal{O}\left(\eta\log\frac{1}{\eta}\right)\right) + (1-\beta_1)\sum_{i=\bar{k}}^{k}\beta_1^{k-i}\boldsymbol{z}_i + \mathcal{O}(\eta^2).$$

A key observation is that $\partial\Phi_{S(\boldsymbol{v}_{\bar{k}})}(\boldsymbol{\theta}_{\bar{k}})S(\boldsymbol{v}_{\bar{k}})\nabla\mathcal{L}(\boldsymbol{\theta}_{\bar{k}}) = 0$ from Lemma G.3, which allows us to view $\phi_{k+1} - \phi_k$ as $\sum_{i=\bar{k}}^{k}\tilde{\boldsymbol{z}}_{k,i} + \mathcal{O}(\eta^2\log\frac{1}{\eta})$ where $\tilde{\boldsymbol{z}}_{k,i} = \partial\Phi_{S(\boldsymbol{v}_{\bar{k}})}(\boldsymbol{\theta}_{\bar{k}})(\eta(1-\beta_1)\beta_1^{k-i}S(\boldsymbol{v}_{\bar{k}})\boldsymbol{z}_i)$. Note that $\tilde{\boldsymbol{z}}_{k,i}$ is $\mathcal{F}_{i+1}$-measurable and its mean is $\boldsymbol{0}$, since $\tilde{\boldsymbol{z}}_{k,i}$ just applies a linear tensor transformation to $\boldsymbol{z}_i$. If we define a constant $C_{6a} := \sup\left\{\left\|\partial\Phi_{S(\boldsymbol{v})}(\boldsymbol{\theta})\right\|_2 \mid (\boldsymbol{v}, \boldsymbol{\theta}) \in \mathcal{X}^{\epsilon_1}\right\} \cdot (1-\beta_1) \cdot R_2$ that is independent of $k$ and $i$, then $\|\tilde{\boldsymbol{z}}_{k,i}\|_2$ is almost surely bounded by $\eta\beta_1^{k-i}C_{6a}\|\boldsymbol{z}_i\|_2$.

For any $k \in [0, K-\Delta K]$ and $\Delta k \in [\Delta K]$, we have

$$\phi_{k+\Delta k} - \phi_k = \sum_{j=k}^{k+\Delta k - 1}(\phi_{j+1} - \phi_j)$$

$$= \sum_{j=k}^{k+\Delta k - 1}\left(\sum_{i=j-2\log_{\beta_1}\eta}^{j}\tilde{\boldsymbol{z}}_{j,i} + O\left(\eta^2\log\frac{1}{\eta}\right)\right)$$

$$= \sum_{i=k-2\log_{\beta_1}\eta}^{k+\Delta k-1} \sum_{j=i}^{\min\left\{k+\Delta k-1, j+2\log_{\beta_1}\eta\right\}} \tilde{z}_{j,i} + \tilde{\mathcal{O}}(\eta^{1-\beta})$$

Denote $Z_i := \sum_{j=i}^{\min\left\{k+\Delta k-1, j+2\log_{\beta_1}\eta\right\}} \tilde{z}_{j,i}$, then each $Z_i$ is a linear transformation of $z_i$ so it is with zero mean, and also $\|Z_i\|_2 \le \eta \cdot \frac{C_{6a}R}{1-\beta_1}\|z_i\|_2 \le \eta \cdot \frac{C_{6a}R}{1-\beta_1}$ a.s. Azuma-Hoeffding's inequality then gives that for any $\delta = \mathcal{O}(\mathrm{poly}(\eta))$, with probability $1-\delta$,

$$\phi_{k+\Delta k} - \phi_k \le \sqrt{2\eta^2 \left(\frac{C_{6a}R}{1-\beta_1}\right)^2 \cdot \left(R_{\mathrm{grp}}H + 2\log_{\beta_1}\eta\right) \cdot \log\frac{2}{\delta}}$$

$$\le C_{6b}\sqrt{\eta^{1-\beta}\log\frac{2}{\delta}}$$

for some constant $C_{6b}$. Finally, plugging in $\delta' = \frac{\delta}{K\cdot\Delta K}$ and taking union bound over all $k \in [0, K-\Delta K]$ and $\Delta k \in [\Delta K]$ gives the theorem. $\qquad\square$

With the concentration bounds so far, we can show that the dynamics behaves "well" during the whole iteration, and we formalize this idea below.

**Definition G.1** ($\delta$-good). *For any $\delta = \mathcal{O}(\mathrm{poly}(\eta))$ and any step $\hat{K} \in [K]$, we define step $\hat{K}$ to be $\delta$-good if and only if the simultaneous establishment of the following statements:*

1. *For any $k \in [0, \hat{K}]$, $\phi_k \in \Gamma$ and $\|\theta_k - \phi_k\|_2 \le C_{8a}\sqrt{\eta\log\frac{1}{\eta\delta}}$.*

2. *For any $k \in [0, \hat{K} - \Delta K]$, $\Delta k \in [\Delta K]$, $\|\phi_{k+\Delta k} - \phi_k\|_2 \le C_{8b}\eta^{0.5-0.5\beta}\sqrt{\log\frac{1}{\eta\delta}}$.*

*Here $C_{8a}$ and $C_{8b} = C_6\sqrt{2}$ are two constants.*

**Lemma G.6.** *When $\eta$ is sufficiently small, with probability $1 - \eta^{100}$, the event $\eta^{100}$-good holds for any step $\hat{K}$ in $[K]$.*

*Proof.* Denote $\delta := \eta^{100}$. From Lemma F.5, with probability $1 - \delta/2$, all $k \in [0, K]$ satisfy $\mathcal{L}(\theta_k) - \mathcal{L}^* \le \mathcal{L}(\theta_0) - \mathcal{L}^* + C_{5b}\eta\log\frac{2K^2}{\delta}$. Note that $D_0 := \mathcal{L}(\theta_0) - \mathcal{L}^*$ is of $\mathcal{O}(\eta\log\frac{1}{\eta\delta})$ since time 0 now refer to the time after convergence. Combining Lemma E.1, this implies $\|\theta_k - \phi_k\|_2 \le \frac{2C_2}{\sqrt{2\mu C_1}} \cdot \sqrt{C_{5a}D_0 + C_{5b}\eta\log\frac{2K^2}{\delta}}$ for any $k \in [0, K]$. When $\eta$ is small enough such that $\|\theta_k - \phi_k\|_2 \le \frac{2C_2}{\sqrt{2\mu C_1}} \cdot \sqrt{C_{5a}D_0 + C_{5b}\eta\log\frac{2K^2}{\delta}} + \eta R/\epsilon < \epsilon_2$, any $\phi_k \in \Gamma$ with $k \ge 0$ will imply $\phi_{k+1} \in \Gamma$, since $\theta_{k+1}$ cannot escape $\Gamma^{\epsilon_2}$. Giving $\phi_0 \in \Gamma$ and using induction, we conclude that all $\phi_k \in \Gamma$ for $k \ge 0$.

When the above holds, the requirement of Lemma G.5 is met. Then with probability $1 - \delta/2$, for any $k \in [0, K-\Delta K]$, $\Delta k \in [\Delta K]$, we have $\|\phi_{k+\Delta k} - \phi_k\|_2 \le C_6\eta^{0.5-0.5\beta}\sqrt{\log\frac{2}{\eta\delta}}$.

Finally, we just take the union of Lemma F.5 and Lemma G.5. With $\log\frac{2K^2}{\delta} \le 8\log\frac{1}{\eta\delta}$ and $\log\frac{2}{\eta\delta} \le 2\log\frac{1}{\eta\delta}$ (which are mild restrictions since $\eta$ is small), we have the theorem. $\qquad\square$

We have proved that our iteration will behave well with high probability, but chances still exist that the iteration is driven out of working zones and becomes intractable. We define a well-behaved sequence that manually redirects the iteration when extreme cases happen.

**Definition G.2** (Well-behaved Sequence). *Denote the event of step $k$ being $\eta^{100}$-good as $\mathcal{E}_k$. Let $\phi_{\mathrm{null}}$ be a fixed point on $\Gamma$. Starting from $\hat{\theta}_0 = \theta_0$ and $\hat{v}_0 = v_0$, we define a sequence of $(\hat{\theta}_k, \hat{v}_k, \hat{m}_k)$ as follows:*

$$\hat{m}_{k+1} := \beta_1\hat{m}_k + (1-\beta_1)(\nabla\mathcal{L}(\hat{\theta}_k) + z_k)$$

$$\hat{v}_{k+1} := \beta_2\hat{v}_k + (1-\beta_2)V((\nabla\mathcal{L}(\hat{\theta}_k) + z_k)(\nabla\mathcal{L}(\hat{\theta}_k) + z_k)^\top)$$

$$\hat{\theta}_{k+1} := \mathbf{1}_{\mathcal{E}_k}\theta_{k+1} + \mathbf{1}_{\bar{\mathcal{E}}_k}\phi_{\mathrm{null}},$$

*where $\mathbf{1}$ is the indicator function: $\mathbf{1}_{\mathcal{E}} = 1$ if event $\mathcal{E}$ happens and $\mathbf{1}_{\mathcal{E}} = 0$ otherwise.*

Note that the update of $\hat{\boldsymbol{\theta}}_k$ can be written as

$$\hat{\boldsymbol{\theta}}_{k+1} := \hat{\boldsymbol{\theta}}_k - \eta S(\hat{\boldsymbol{v}}_{k+1})\hat{\boldsymbol{m}}_{k+1}$$
$$\underbrace{-\mathbf{1}_{\bar{\mathcal{E}}_k}(\hat{\boldsymbol{\theta}}_k - \eta S(\hat{\boldsymbol{v}}_{k+1})\hat{\boldsymbol{m}}_{k+1}) + \mathbf{1}_{\bar{\mathcal{E}}_k}\boldsymbol{\phi}_{\text{null}}}_{:=\boldsymbol{e}_k}$$

where $\boldsymbol{e}_k$ denotes the redirection under extreme cases. By definition, $\boldsymbol{e}_k$ equals zero in the vast majority of cases, and in other cases it's still bounded by a constant, so all moments of $\boldsymbol{e}_k$ are within $\mathcal{O}(\eta^{100})$ which is negligibly small.

### G.3 Moment Calculation of AGMs Near Manifold

**Additional Notations.** To utilize the analysis framework in Gu et al. [2023b], we first introduce some notations needed. Consistent with Gu et al. [2023b], we pretend that AGMs proceed with $H = \frac{1}{\eta}$ local steps, as a single worker (without multiple workers). We denote every $H$ steps as one round. Next, we define a "giant step", which encompasses $R_{\text{grp}} = \frac{1}{\eta^\beta}$ rounds, corresponding to $R_{\text{grp}} \cdot H$ steps. We consider a total timescope of $\frac{T}{\eta^2}$ steps, which corresponds to $\frac{T}{\eta^{1-\beta}}$ giant steps.

For any $0 \leq s < R_{\text{grp}}$ and $0 \leq t \leq H$, we use $\hat{\boldsymbol{\theta}}_t^{(s)}$ and $\hat{\boldsymbol{\theta}}_k$ (where $k = sH + t$) exchangeably to denote the parameter we get on the $t$-th local step of round $s$, which is also the $k$-th global step. Also note that for any $0 \leq s < R_{\text{grp}}$, $\hat{\boldsymbol{\theta}}_H^{(s)}$ and $\hat{\boldsymbol{\theta}}_0^{(s+1)}$ refer to the same thing. We define the notation $\hat{\boldsymbol{v}}_t^{(s)}$, $\hat{\boldsymbol{m}}_t^{(s)}$ and $\mathcal{E}_t^{(s)}$ in the same way as we did for $\boldsymbol{\theta}$. Furthermore, we define

$$\hat{\boldsymbol{g}}_t^{(s)} := \nabla \ell_t^{(s)}\left(\hat{\boldsymbol{\theta}}_t^{(s)}\right), \ \hat{\boldsymbol{S}}_k = S\left(\hat{\boldsymbol{v}}_k\right), \ \hat{\boldsymbol{S}}_t^{(s)} := S\left(\hat{\boldsymbol{v}}_t^{(s)}\right), \ \hat{\boldsymbol{S}}^{(s)} := \hat{\boldsymbol{S}}_0^{(s)}, \ \hat{\boldsymbol{\phi}}^{(s)} := \boldsymbol{\Phi}_{\hat{\boldsymbol{S}}^{(s)}}\left(\hat{\boldsymbol{\theta}}_0^{(s)}\right),$$

$$\hat{\boldsymbol{x}}_t^{(s)} := \hat{\boldsymbol{\theta}}_t^{(s)} - \hat{\boldsymbol{\phi}}^{(s)}, \ \Delta\hat{\boldsymbol{\phi}}^{(s)} := \hat{\boldsymbol{\phi}}^{(s)} - \hat{\boldsymbol{\phi}}^{(0)}, \ \boldsymbol{\Sigma}_0 := \boldsymbol{\Sigma}(\hat{\boldsymbol{\phi}}^{(0)}), \ \boldsymbol{P}_\parallel := \partial\boldsymbol{\Phi}_{\hat{\boldsymbol{S}}^{(0)}}(\hat{\boldsymbol{\phi}}^{(0)}), \ \boldsymbol{P}_\perp := \boldsymbol{I} - \boldsymbol{P}_\parallel,$$

$$\hat{\boldsymbol{q}}_t^{(s)} := \mathbb{E}\left[\hat{\boldsymbol{x}}_t^{(s)}\right], \ \hat{\boldsymbol{A}}_t^{(s)} := \mathbb{E}\left[\hat{\boldsymbol{x}}_t^{(s)}\hat{\boldsymbol{x}}_t^{(s)\top}\right], \ \hat{\boldsymbol{B}}_t^{(s)} := \mathbb{E}\left[\hat{\boldsymbol{x}}_t^{(s)}\Delta\hat{\boldsymbol{\phi}}^{(s)\top}\right].$$

**Corollary G.1.** *There exist constants $C_{9a}, C_{9b}, C_{9c}$ such that for all $0 \leq s < R_{\text{grp}}$, $0 \leq t \leq H$,*

$$\left\|\hat{\boldsymbol{x}}_t^{(s)}\right\|_2 \leq C_{9a}\sqrt{\eta \log \frac{1}{\eta}},$$

$$\left\|\hat{\boldsymbol{\theta}}_t^{(s)} - \hat{\boldsymbol{\theta}}_0^{(s)}\right\|_2 \leq C_{9b}\sqrt{\eta \log \frac{1}{\eta}},$$

$$\left\|\hat{\boldsymbol{\phi}}^{(s)} - \hat{\boldsymbol{\phi}}^{(0)}\right\|_2 \leq C_{9c}\eta^{0.5-0.5\beta}\sqrt{\log \frac{1}{\eta}}.$$

*Proof.* Substituting $\delta = \eta^{100}$. When $\mathcal{E}$ holds, this follows directly from the definition of $\delta$-goodness; Otherwise, all $\hat{\boldsymbol{\theta}}$ and $\hat{\boldsymbol{\phi}}$ are equal, and these quantities are equal to $\mathbf{0}$. $\qquad\qquad\square$

**Impact of Momentum.** Our conclusion regrading to the impact of Momentum on the implicit bias is similar to the conclusion in Wang et al. [2023]: It does not impact the implicit bias. Further, our analysis is based on moment methods and can give exact error bounds. First, we state some technical lemmas in order to show that introducing momentum will not cause the gradient to deviate too much from itself, i.e. $\mathbb{E}[\hat{\boldsymbol{m}}_t]$ is close to $\mathbb{E}[\hat{\boldsymbol{g}}_t]$. Once this guarantee is established, we can replace $\hat{\boldsymbol{m}}_t$ with $\hat{\boldsymbol{g}}_t$ in the moment calculation to simplify it. The general idea of the proof is to show that if $i$ is close to $t$, then $\mathbb{E}[\nabla\mathcal{L}(\hat{\boldsymbol{\theta}}_{i-1})]$ will become close to $\mathbb{E}[\nabla\mathcal{L}(\hat{\boldsymbol{\theta}}_{t-1})]$, and if $i$ is far from $t$, then the contribution of $\mathbb{E}[\nabla\mathcal{L}(\hat{\boldsymbol{\theta}}_{i-1})]$ would be negligible in $\mathbb{E}[\hat{\boldsymbol{m}}_t]$.

**Lemma G.7.** *For any $k \geq 0$, we have*

$$\left\|\mathbb{E}\left[\nabla\mathcal{L}\left(\hat{\boldsymbol{\theta}}_{k+1}\right) - \nabla\mathcal{L}\left(\hat{\boldsymbol{\theta}}_k\right)\right]\right\|_2 \leq C_{10}\eta^{1.5}$$

*for some constant $C_{10}$.*

*Proof.* We have

$$\nabla\mathcal{L}(\hat{\boldsymbol{\theta}}_{k+1}) - \nabla\mathcal{L}(\hat{\boldsymbol{\theta}}_k) = \nabla^2\mathcal{L}(\hat{\boldsymbol{\theta}}_k)(\hat{\boldsymbol{\theta}}_{k+1} - \hat{\boldsymbol{\theta}}_k) + \mathcal{O}\left(\left\|\hat{\boldsymbol{\theta}}_{k+1} - \hat{\boldsymbol{\theta}}_k\right\|_2^2\right)$$

$$= \nabla^2\mathcal{L}(\hat{\boldsymbol{\theta}}_k)(\hat{\boldsymbol{\theta}}_{k+1} - \hat{\boldsymbol{\theta}}_k) + \mathcal{O}(\eta^2) + \mathcal{O}(\|\boldsymbol{e}_k\|_2),$$

since $\left\|\hat{\boldsymbol{\theta}}_{k+1} - \hat{\boldsymbol{\theta}}_k\right\|_2 = \|\eta S(\hat{\boldsymbol{v}}_k)\hat{\boldsymbol{m}}_k - \boldsymbol{e}_k\|_2 = \mathcal{O}(\eta) + \mathcal{O}(\|\boldsymbol{e}_k\|_2)$. Let $\bar{k} = k - \log_{\beta_1}(\eta)$ be a threshold that is logarithmically close to $k$, then we have

$$\nabla^2\mathcal{L}(\hat{\boldsymbol{\theta}}_k)(\hat{\boldsymbol{\theta}}_{k+1} - \hat{\boldsymbol{\theta}}_k) = \left(\nabla^2\mathcal{L}(\hat{\boldsymbol{\theta}}_{\bar{k}}) + \mathcal{O}\left(\left\|\hat{\boldsymbol{\theta}}_k - \hat{\boldsymbol{\theta}}_{\bar{k}}\right\|_2\right)\right)\left(\hat{\boldsymbol{\theta}}_{k+1} - \hat{\boldsymbol{\theta}}_k\right)$$

$$= \nabla^2\mathcal{L}(\hat{\boldsymbol{\theta}}_{\bar{k}})\left(\hat{\boldsymbol{\theta}}_{k+1} - \hat{\boldsymbol{\theta}}_k\right) + \mathcal{O}\left(\eta \cdot \log_{\beta_1}(\eta) \cdot \eta\right) + \mathcal{O}(\|\boldsymbol{e}_k\|_2)$$

$$= \eta\nabla^2\mathcal{L}(\hat{\boldsymbol{\theta}}_{\bar{k}})S(\hat{\boldsymbol{v}}_{k+1})\hat{\boldsymbol{m}}_{k+1} + \mathcal{O}\left(\eta^2\log\frac{1}{\eta}\right) + \mathcal{O}(\|\boldsymbol{e}_k\|_2).$$

Recentering the Hessian term to $\hat{\boldsymbol{\theta}}_{\bar{k}}$ allows us to take conditional expectation $\mathbb{E}_{\bar{k}}$ on $S(\hat{\boldsymbol{v}}_{k+1})\hat{\boldsymbol{m}}_{k+1}$:

$$\mathbb{E}\left[\nabla^2\mathcal{L}(\hat{\boldsymbol{\theta}}_{\bar{k}})S(\hat{\boldsymbol{v}}_{k+1})\hat{\boldsymbol{m}}_{k+1}\right] = \mathbb{E}\left[\nabla^2\mathcal{L}(\hat{\boldsymbol{\theta}}_{\bar{k}})\mathbb{E}_{\bar{k}}\left[S(\hat{\boldsymbol{v}}_{k+1})\hat{\boldsymbol{m}}_{k+1}\right]\right].$$

After that, notice that

$$\|\mathbb{E}_{\bar{k}}\left[S(\hat{\boldsymbol{v}}_{k+1})\hat{\boldsymbol{m}}_{k+1}\right]\|_2 = \|\mathbb{E}_{\bar{k}}\left[S(\mathbb{E}_{\bar{k}}[\hat{\boldsymbol{v}}_{k+1}])\hat{\boldsymbol{m}}_{k+1}\right]\|_2 + \mathcal{O}(\|\hat{\boldsymbol{v}}_{k+1} - \mathbb{E}_{\bar{k}}[\hat{\boldsymbol{v}}_{k+1}]\|_2)$$

$$= \|S(\mathbb{E}_{\bar{k}}[\hat{\boldsymbol{v}}_{k+1}])\mathbb{E}_{\bar{k}}[\hat{\boldsymbol{m}}_{k+1}]\|_2 + \mathcal{O}(\|\hat{\boldsymbol{v}}_{k+1} - \mathbb{E}_{\bar{k}}[\hat{\boldsymbol{v}}_{k+1}]\|_2)$$

$$= \mathcal{O}(\underbrace{\|\mathbb{E}_{\bar{k}}[\hat{\boldsymbol{m}}_{k+1}]\|_2}_{=:D_1}) + \mathcal{O}(\underbrace{\|\hat{\boldsymbol{v}}_{k+1} - \mathbb{E}_{\bar{k}}[\hat{\boldsymbol{v}}_{k+1}]\|_2}_{=:D_2})$$

since $S$ and $\hat{\boldsymbol{m}}$ are both bounded by constant scale. We figure out the orders of these two terms respectively:

$$D_1 = \left\|\mathbb{E}_{\bar{k}}\left[\beta_1^{k-\bar{k}+1}\hat{\boldsymbol{m}}_{\bar{k}} + (1-\beta_1)\sum_{i=\bar{k}}^{k}\beta_1^{k-i}\hat{\boldsymbol{g}}_i\right]\right\|_2$$

$$= \mathcal{O}\left(\beta_1^{\log_{\beta_1}(\eta)}\right) + \left\|\mathbb{E}_{\bar{k}}\left[(1-\beta_1)\sum_{i=\bar{k}}^{k}\beta_1^{k-i}\hat{\boldsymbol{g}}_i\right]\right\|_2$$

$$= \mathcal{O}(\eta) + \left\|\mathbb{E}_{\bar{k}}\left[(1-\beta_1)\sum_{i=\bar{k}}^{k}\beta_1^{k-i}\nabla\mathcal{L}(\hat{\boldsymbol{\theta}}_i)\right]\right\|_2$$

$$= \mathcal{O}(\eta) + \mathcal{O}(\eta^{0.5}) = \mathcal{O}(\eta^{0.5})$$

since $\nabla\mathcal{L}$ is uniformly bounded by $\mathcal{O}(\eta^{0.5})$ after convergence (see Lemma F.5); And

$$D_2 = (1-\beta_2)\sum_{i=\bar{k}}^{k}\beta_2^{k-i}\left(V(\hat{\boldsymbol{g}}_i\hat{\boldsymbol{g}}_i^\top) - \mathbb{E}_{\bar{k}}\left[V(\hat{\boldsymbol{g}}_i\hat{\boldsymbol{g}}_i^\top)\right]\right)$$

$$= \mathcal{O}\left(b_2 \cdot (k-\bar{k})\right)$$

$$= \mathcal{O}\left(\eta^2\log\frac{1}{\eta}\right),$$

since $V$ is bounded by a constant scale. Now combining the above together, we have

$$\left\|\mathbb{E}[\nabla\mathcal{L}(\hat{\boldsymbol{\theta}}_{k+1}) - \nabla\mathcal{L}(\hat{\boldsymbol{\theta}}_k)]\right\|_2 = \eta\mathbb{E}\left[\nabla^2\mathcal{L}(\hat{\boldsymbol{\theta}}_{\bar{k}})\mathbb{E}_{\bar{k}}\left[S(\hat{\boldsymbol{v}}_{k+1})\hat{\boldsymbol{m}}_{k+1}\right]\right] + \mathcal{O}\left(\eta^2\log\frac{1}{\eta}\right)$$

$$+ \mathcal{O}(\mathbb{E}[\|\boldsymbol{e}_k\|_2])$$

$$= \eta \cdot \mathcal{O}(D_1 + D_2) + \mathcal{O}\left(\eta^2\log\frac{1}{\eta}\right) + \mathcal{O}(\eta^{100})$$

$$= \mathcal{O}(\eta^{1.5}),$$

which concludes the proof. $\square$

With Lemma G.7, we are ready to deduce the closeness between $\mathbb{E}[\hat{\boldsymbol{m}}_k]$ and $\mathbb{E}[\hat{\boldsymbol{g}}_k]$.

**Lemma G.8.** *For any $k \geq 0$, let $\bar{k} = k - 2\log_{\beta_1}(\eta)$, we have*

$$\|\mathbb{E}_{\bar{k}}[\hat{\boldsymbol{m}}_{k+1} - \hat{\boldsymbol{g}}_{k+1}]\|_2 \leq C_{11}\eta^{1.5}\log\frac{1}{\eta}, \quad a.s.$$

*Note that this also implies that $\|\mathbb{E}[\hat{\boldsymbol{m}}_{k+1} - \hat{\boldsymbol{g}}_{k+1}]\|_2 \leq C_{11}\eta^{1.5}\log\frac{1}{\eta}$.*

*Proof.* Expanding $\mathbb{E}_{\bar{k}}[\hat{\boldsymbol{m}}_{k+1}]$, we have

$$\mathbb{E}_{\bar{k}}[\hat{\boldsymbol{m}}_{k+1}] = \mathbb{E}_{\bar{k}}\left[(1 - \beta_1)\sum_{i=1}^{k}\beta_1^{k-i}\hat{\boldsymbol{g}}_i\right]$$

$$= (1 - \beta_1)\sum_{i=1}^{\bar{k}-1}\beta_1^{k-i}\hat{\boldsymbol{g}}_i + (1 - \beta_1)\sum_{i=\bar{k}}^{k}\beta_1^{k-i}\mathbb{E}_{\bar{k}}[\hat{\boldsymbol{g}}_i]$$

$$= \underbrace{(1 - \beta_1)\sum_{i=1}^{\bar{k}-1}\beta_1^{k-i}\hat{\boldsymbol{g}}_i}_{:=E_1} + \underbrace{(1 - \beta_1)\sum_{i=\bar{k}}^{k}\beta_1^{k-i}\nabla\mathcal{L}(\hat{\boldsymbol{\theta}}_i)}_{:=E_2}$$

Note that $E_1$ is neglegible:

$$\|E_1\|_2 = \left\|(1 - \beta_1)\sum_{i=1}^{\bar{k}-1}\beta_1^{k-i}\hat{\boldsymbol{g}}_i\right\|_2$$

$$= (1 - \beta_1)\sum_{i=1}^{\bar{k}-1}\beta_1^{k-i}\cdot\mathcal{O}(1)$$

$$\leq (1 - \beta_1)\sum_{i=2\log_{\beta_1}(\eta)}^{\infty}\beta_1^i\cdot\mathcal{O}(1)$$

$$= \mathcal{O}\left(\beta_1^{2\log_{\beta_1}(\eta)}\right) = \mathcal{O}\left(\eta^2\right),$$

and that $E_2$ is close to $\nabla\mathcal{L}(\hat{\boldsymbol{\theta}}_k)$:

$$\left\|E_2 - \mathbb{E}[\nabla\mathcal{L}(\hat{\boldsymbol{\theta}}_k)]\right\|_2 = \left\|(1 - \beta_1)\sum_{i=\bar{k}}^{k}\beta_1^{k-i}\mathbb{E}[\nabla\mathcal{L}(\hat{\boldsymbol{\theta}}_i)] - \mathbb{E}[\nabla\mathcal{L}(\hat{\boldsymbol{\theta}}_k)]\right\|_2$$

$$= \left\|(1 - \beta_1)\sum_{i=\bar{k}}^{k}\beta_1^{k-i}\mathbb{E}\left[\nabla\mathcal{L}(\hat{\boldsymbol{\theta}}_i) - \nabla\mathcal{L}(\hat{\boldsymbol{\theta}}_k)\right]\right\|_2 + \mathcal{O}(\eta^2)$$

$$\leq (1 - \beta_1)\cdot\left(k - \bar{k}\right)\cdot C_{10}\eta^{1.5} + \mathcal{O}(\eta^2). \quad \text{(by Lemma G.7)}$$

Combining the results of $E_1$ and $E_2$ gives

$$\|\mathbb{E}_{\bar{k}}[\hat{\boldsymbol{m}}_k - \hat{\boldsymbol{g}}_k]\|_2 \leq \|E_1\|_2 + \left\|E_2 - \mathbb{E}[\nabla\mathcal{L}(\hat{\boldsymbol{\theta}}_k)]\right\|_2$$

$$\leq (1 - \beta_1)\cdot 2\log_{\beta_1}(\eta)\cdot C_{10}\eta^{1.5} + \mathcal{O}(\eta^2)$$

$$\leq C_{11}\eta^{1.5}\log\frac{1}{\eta}$$

for some constant $C_{11}$, which completes the proof. $\quad\square$

### G.3.1 Moment Calculation Within a Giant Step

In this part, we aim to give the change of first and second moments of $\phi$ and $\hat{v}$, which is the basis of deriving the SDE for AGMs.

Now there are only a few preparations left before we get into the direct part of the moment calculation. For all $0 \leq s < R_{\mathrm{grp}}$, $0 \leq t \leq H$. Note that $\|\hat{v}_{k+1} - \hat{v}_k\|_2 = (1 - \beta_2) \left\| V \left( \hat{g}_k \hat{g}_k^\top \right) - \hat{v}_k \right\|_2 = \mathcal{O}(1 - \beta_2) = \mathcal{O}(\eta^2)$, so combining with the Lipschitzness of $S$ gives

$$\left\| \hat{S}_{k_2} - \hat{S}_{k_1} \right\|_2 = \mathcal{O}\left( (k_2 - k_1) \eta^2 \right)$$

for any $k_2 > k_1$ and $k_2 - k_1 = o\left( \eta^{-2} \right)$. Next, we begin our moment calculation analysis, starting from the update in a single step.

**Lemma G.9.** *For all $0 \leq k \leq R_{\mathrm{grp}} H$, it holds that*

$$\mathbb{E}\left[ \hat{\boldsymbol{\theta}}_{k+1} \right] = \mathbb{E}\left[ \hat{\boldsymbol{\theta}}_k - \eta \hat{S}_0 \hat{g}_k \right] + \mathcal{O}\left( \eta^{2.5 - \beta} \right).$$

*Proof.* We write the update rule of AGM under a single step as

$$\hat{\boldsymbol{\theta}}_{k+1} = \hat{\boldsymbol{\theta}}_k - \eta \hat{S}_k \hat{m}_{k+1} - e_k$$

$$= \hat{\boldsymbol{\theta}}_k - \eta \left[ \hat{S}_k \hat{g}_k + \hat{S}_k \left( \hat{m}_{k+1} - \hat{g}_k \right) \right] - e_k$$

$$= \hat{\boldsymbol{\theta}}_k - \eta \left[ \hat{S}_0 \hat{g}_k + \underbrace{\left( \hat{S}_k - \hat{S}_0 \right) \hat{g}_k}_{\Delta \hat{\boldsymbol{\theta}}_1} + \underbrace{\hat{S}_k \left( \hat{m}_{k+1} - \hat{g}_k \right)}_{\Delta \hat{\boldsymbol{\theta}}_2} \right] - e_k,$$

where we recall that $e_k := -\mathbf{1}_{\bar{\mathcal{E}}_k} (\hat{\boldsymbol{\theta}}_k - \eta S(\hat{v}_{k+1}) \hat{m}_{k+1}) + \mathbf{1}_{\bar{\mathcal{E}}_k} \phi_{\mathrm{null}}$. We can prove that $\Delta \hat{\boldsymbol{\theta}}_1$ and $\Delta \hat{\boldsymbol{\theta}}_2$ are small enough to be negligible in expectation for our following calculation.

Specifically, if $k = 0$ then $\Delta \hat{\boldsymbol{\theta}}_1 = \mathbf{0}$; and if $k > 0$, we can decompose $\mathbb{E}\left[ \Delta \hat{\boldsymbol{\theta}}_1 \right]$ as:

$$\mathbb{E}\left[ \Delta \hat{\boldsymbol{\theta}}_1 \right] = \mathbb{E}\left[ \left( \hat{S}_{k-1} - \hat{S}_0 \right) \hat{g}_k + \left( \hat{S}_k - \hat{S}_{k-1} \right) \hat{g}_k \right]$$

$$= \mathbb{E}\left[ \left( \hat{S}_{k-1} - \hat{S}_0 \right) \nabla \mathcal{L}\left( \hat{\boldsymbol{\theta}}_k \right) \right] + \mathbb{E}\left[ \left( \hat{S}_k - \hat{S}_{k-1} \right) \hat{g}_k \right]$$

$$= \mathcal{O}((k-1)\eta^2 \cdot \eta^{0.5}) + \mathcal{O}(\eta^2)$$

$$= \mathcal{O}(H \cdot R_{\mathrm{grp}} \cdot \eta^{2.5} + \eta^2)$$

$$= \mathcal{O}(\eta^{1.5 - \beta}).$$

Here, the second equality holds since the gradient noise term as step $k$ $z_k$ is conditioned on time $k$, when $\hat{S}_{k-1}$ has already been determined, thus we can take the conditional expectation.

For $\Delta \hat{\boldsymbol{\theta}}_2$, let $\bar{k} = k - 2 \log_{\beta_1}(\eta)$, we have

$$\mathbb{E}\left[ \Delta \hat{\boldsymbol{\theta}}_2 \right] = \mathbb{E}\left[ \hat{S}_{\bar{k}-1} \left( \hat{m}_{k+1} - \hat{g}_k \right) + \mathcal{O}\left( \eta^2 \log \frac{1}{\eta} \right) \right]$$

$$= \mathbb{E}\left[ \hat{S}_{\bar{k}-1} \mathbb{E}_{\bar{k}} \left[ \left( \hat{m}_{k+1} - \hat{g}_k \right) \right] \right] + \mathcal{O}\left( \eta^2 \log \frac{1}{\eta} \right)$$

$$= \mathcal{O}\left( \eta^{1.5} \log \frac{1}{\eta} \right) + \mathcal{O}\left( \eta^2 \log \frac{1}{\eta} \right)$$

$$= \mathcal{O}\left( \eta^{1.5} \log \frac{1}{\eta} \right),$$

where the second-to-last equality follows from Lemma G.8. Finally, we have

$$\mathbb{E}\left[ \hat{\boldsymbol{\theta}}_{k+1} \right] = \mathbb{E}\left[ \hat{\boldsymbol{\theta}}_k - \eta \hat{S}_0 \hat{g}_k \right] + \mathcal{O}\left( \eta^{2.5 - \beta} \right) + \mathcal{O}\left( \eta^{2.5} \log \frac{1}{\eta} \right) + \mathcal{O}(\eta^{100})$$

$$= \mathbb{E}\left[ \hat{\boldsymbol{\theta}}_k - \eta \hat{S}_0 \hat{g}_k \right] + \mathcal{O}\left( \eta^{2.5 - \beta} \right),$$

which concludes the proof. $\qquad\square$

After getting the update rule of $\hat{\boldsymbol{\theta}}_k$, we then derive the moment change during the single round with $\boldsymbol{H}$ steps. To this end, we recall our modification of manifold projection from a "Gradient Flow" manner to a "Preconditioned Flow" manner in Definition E.2.

**Definition G.3** (Preconditioned Flow Projection). *Fix a point $\theta_{null} \notin \Gamma$. Given a Positive Semi-Definite matrix $\boldsymbol{M}$. For $x \in \mathbb{R}^d$, consider the preconditioned flow $\frac{\mathrm{d}x(t)}{\mathrm{d}t} = -\boldsymbol{M}\nabla\mathcal{L}(x(t))$ with $x(0) = x$. We denote the preconditioned flow projection of $x$ as $\Phi_{\boldsymbol{M}}(x)$, i.e. $\Phi_{\boldsymbol{M}}(x) := \lim_{t\to+\infty} x(t)$ if the limit exists and belongs to $\Gamma$, and $\Phi_{\boldsymbol{M}}(x) = \theta_{null}$ otherwise.*

We decompose the preconditioner matrix in the very begining of the giant step as $\hat{\boldsymbol{S}}_0 = \hat{\boldsymbol{S}}(\hat{\boldsymbol{v}}_0) = \boldsymbol{P}\boldsymbol{P}$, where $\boldsymbol{P} = \hat{\boldsymbol{S}}_0^{1/2}$. We then provide the first moment calculation of $\hat{\boldsymbol{\phi}}$ in the following lemma. Before that, we first introduce the operator $\mathcal{V}_{\boldsymbol{H}}$.

**Definition G.4.** *Given a Positive Semi-Definite matrix $\boldsymbol{H} \in \mathbb{R}^{d\times d}$, whose $j$-th eigenvalue and the corresponding orthonormal eigenvector are denoted by $\lambda_j$ and $\boldsymbol{v}_j$. We then define the operator $\mathcal{V}_{\boldsymbol{H}}(\cdot) : \mathbb{R}^{d\times d} \to \mathbb{R}^{d\times d}$ as*

$$\mathcal{V}_{\boldsymbol{H}}(\cdot) = \sum_{i,j:\lambda_i\neq0\vee\lambda_j\neq0} \frac{1}{\lambda_i + \lambda_j} \left\langle \cdot, \boldsymbol{v}_i\boldsymbol{v}_j^\top \right\rangle \boldsymbol{v}_i\boldsymbol{v}_j^\top.$$

Intuitively, the above operator projects the one matrix into the basis of $\boldsymbol{H}$ and sums up the corresponding components with weights $\frac{1}{\lambda_i+\lambda_j}$. Then we present our moment calculation lemma.

**Lemma G.10.** *The expectation of the change of the manifold projection every round is*

$$\mathbb{E}\left[\hat{\boldsymbol{\phi}}^{(s+1)} - \hat{\boldsymbol{\phi}}^{(s)}\right] = -\frac{H\eta^2}{2}\hat{\boldsymbol{S}}_0\partial\Phi_{\hat{\boldsymbol{S}}_0}(\hat{\boldsymbol{\phi}}^{(0)})\hat{\boldsymbol{S}}_0\partial^2\nabla\mathcal{L}(\hat{\boldsymbol{\phi}}_{(0)})\left[\boldsymbol{P}\mathcal{V}_{\nabla^2\mathcal{L}'(\hat{\boldsymbol{\phi}}'_{(0)})}(\boldsymbol{\Sigma}_{0,\boldsymbol{P}})\boldsymbol{P}\right] + \tilde{\mathcal{O}}(\eta^{1.5-\beta})$$

*for $R_0 < s < R_{\mathrm{grp}}$, and*

$$\mathbb{E}\left[\hat{\boldsymbol{\phi}}^{(s+1)} - \hat{\boldsymbol{\phi}}^{(s)}\right] = \tilde{\mathcal{O}}(\eta)$$

*for $s \leq R_0$, where $R_0 := \max\left\{\left\lceil\frac{10}{\lambda_{\max}\alpha}\log\frac{1}{\eta}\right\rceil, \left\lceil 2\log_{1/\beta}\frac{1}{\eta}\right\rceil\right\}$ and $\boldsymbol{\Sigma}_{0,\boldsymbol{P}} := \boldsymbol{P}\boldsymbol{\Sigma}_0\boldsymbol{P}$.*

*Proof.* First, we consider the scenario when $R_0 < s < R_{\mathrm{grp}}$. Let $L'(\boldsymbol{x}) := L(\boldsymbol{P}\boldsymbol{x})$, then

$$\nabla L'(\boldsymbol{x}) = \boldsymbol{P}\nabla L(\boldsymbol{P}\boldsymbol{x})$$
$$\nabla^2 L'(\boldsymbol{x}) = \boldsymbol{P}\nabla^2 L(\boldsymbol{P}\boldsymbol{x})\boldsymbol{P}$$
$$\boldsymbol{\Sigma}'(\boldsymbol{x}) = \boldsymbol{P}\boldsymbol{\Sigma}(\boldsymbol{P}\boldsymbol{x})\boldsymbol{P}$$
$$\partial^2(\nabla L')(\boldsymbol{x})[\boldsymbol{M}] = \boldsymbol{P}\partial^2(\nabla L)(\boldsymbol{P}\boldsymbol{x})[\boldsymbol{P}\boldsymbol{M}\boldsymbol{P}].$$

For a one-step GD update, we consider an auxiliary process $\{\hat{\boldsymbol{\theta}}'_t\}$

$$\hat{\boldsymbol{\theta}}'_{t+1} = \hat{\boldsymbol{\theta}}'_t - \eta\nabla\mathcal{L}'(\hat{\boldsymbol{\theta}}'_t) + \mathcal{O}\left(\eta^{2.5-\beta}\right)$$
$$= \hat{\boldsymbol{\theta}}'_t - \eta\boldsymbol{P}\nabla\mathcal{L}(\boldsymbol{P}\hat{\boldsymbol{\theta}}'_t) + \mathcal{O}\left(\eta^{2.5-\beta}\right).$$

Similarly, we define $\hat{\boldsymbol{A}}_t^{'(s)} := \mathbb{E}[\hat{\boldsymbol{x}}_t^{'(s)}\hat{\boldsymbol{x}}_t^{'(s)\top}]$, $\hat{\boldsymbol{q}}_t^{'(s)} := \mathbb{E}[\hat{\boldsymbol{x}}_t^{'(s)}]$, and $\hat{\boldsymbol{B}}_t^{'(s)} := \mathbb{E}[\hat{\boldsymbol{x}}_t^{'(s)}\Delta\hat{\phi}^{'(s)\top}]$, and $\Phi(\boldsymbol{x})$ is the gradient flow projection of point $\boldsymbol{x}$. We further define $\hat{\boldsymbol{\phi}}'(s) := \Phi(\hat{\boldsymbol{\theta}}^{'(s)})$.

Now we are interested in the update of $\boldsymbol{P}\hat{\boldsymbol{\theta}}'$, which is

$$\boldsymbol{P}\hat{\boldsymbol{\theta}}'_{t+1} = \boldsymbol{P}\hat{\boldsymbol{\theta}}'_t - \eta\hat{\boldsymbol{S}}_0\nabla\mathcal{L}(\boldsymbol{P}\hat{\boldsymbol{\theta}}'_t) + \mathcal{O}\left(\eta^{2.5-\beta}\right). \tag{7}$$

One can obviously see the update rule of $\boldsymbol{P}\hat{\boldsymbol{\theta}}'$ resembles the update rule of $\hat{\boldsymbol{\theta}}$ in Lemma G.9. Now we set $\hat{\boldsymbol{\theta}}' = \boldsymbol{P}^{-1}\hat{\boldsymbol{\theta}}$, then Equation (7) is satisfied, and combining Equation (7) and Lemma G.9 gives

$$\boldsymbol{q}_{t+1}^{'(s)} = \boldsymbol{q}_{t+1}^{'(s)} - \eta\nabla\mathcal{L}'(\hat{\boldsymbol{\theta}}_t^{'(s)}) + \mathcal{O}\left(\eta^{2.5-\beta}\right).$$

Notice that the above equation resembles the single update for SGD, which allows us to apply Lemma I.36 from Gu et al. [2023b] for the update of $\hat{\boldsymbol{\theta}}'$, with loss function $\mathcal{L}'(\hat{\boldsymbol{\theta}})$, number of workers $k=1$ and manifold projection $\Phi'(\hat{\boldsymbol{\theta}})$, which gives

$$
\begin{aligned}
\boldsymbol{P}\mathbb{E}\left[\hat{\boldsymbol{\phi}}^{'(s+1)} - \hat{\boldsymbol{\phi}}^{'(s)}\right] &= \mathbb{E}\left[\hat{\boldsymbol{\phi}}^{(s+1)} - \hat{\boldsymbol{\phi}}^{(s)}\right] \\
&= -\frac{H\eta^2}{2}\boldsymbol{P}\boldsymbol{P}\partial\Phi_{\hat{\boldsymbol{S}}_0}(\hat{\boldsymbol{\phi}}^{(0)})\boldsymbol{P}\boldsymbol{P}\partial^2\nabla\mathcal{L}(\hat{\boldsymbol{\phi}}_{(0)})[\boldsymbol{P}\mathcal{V}_{\nabla^2\mathcal{L}'(\hat{\boldsymbol{\phi}}'_{(0)})}(\boldsymbol{P}\boldsymbol{\Sigma}_0\boldsymbol{P})\boldsymbol{P}] \\
&\quad + \tilde{O}(\eta^{1.5-\beta}),
\end{aligned}
$$

where the first equation uses the fact that $\boldsymbol{P}\hat{\boldsymbol{\phi}}(\hat{\boldsymbol{\theta}}') = \Phi_{\boldsymbol{S}}(\hat{\boldsymbol{\theta}})$, and it can be verified with the definitions of $\hat{\boldsymbol{\phi}}'$, $\Phi_{\boldsymbol{S}}$, and $\hat{\boldsymbol{\theta}}'$.

The proof when $s \le R_0$ is a direct conclusion of Lemma I.36 in Gu et al. [2023b] since the $R_0 \propto \log\frac{1}{\eta}$ in our case. $\qquad\square$

Notice the above equation for the moment of $\hat{\boldsymbol{\phi}}$ contains $\phi'$. The next corollary eliminates $\phi'$ from the formula.

**Corollary G.2.** *The expectation of the change of manifold projection every round is:*

$$
\mathbb{E}\left[\phi^{(s+1)} - \phi^{(s)}\right] = \begin{cases} \frac{H\eta^2}{2}\hat{\boldsymbol{S}}_0\partial^2\Phi_{\hat{\boldsymbol{S}}_0}(\phi^{(0)})[\hat{\boldsymbol{S}}_0\boldsymbol{\Sigma}_0\hat{\boldsymbol{S}}_0] + \tilde{\mathcal{O}}(\eta^{1.5-\beta}), & R_0 < s < R_{\mathrm{grp}} \\ \tilde{\mathcal{O}}(\eta), & s \le R_0 \end{cases}
$$

*Proof.* Notice that for the preconditioned projection, we also have the corresponding transformation

$$
\begin{aligned}
\partial\Phi'(\boldsymbol{x}') &= \boldsymbol{P}\partial\Phi_{\hat{\boldsymbol{S}}}(\boldsymbol{P}\boldsymbol{x}')\boldsymbol{P} \\
\partial^2\Phi'(\boldsymbol{x}')[\boldsymbol{M}] &= \boldsymbol{P}\partial^2\Phi(\boldsymbol{x}')[\boldsymbol{P}\boldsymbol{M}\boldsymbol{P}].
\end{aligned}
$$

The above two equations and Lemma I.36 in Gu et al. [2023b] complete the proof. $\qquad\square$

**Lemma G.11.** *The second moment of the change of manifold projection every round is*

$$
\mathbb{E}\left[(\hat{\boldsymbol{\phi}}^{(s+1)} - \hat{\boldsymbol{\phi}}^{(s)})(\hat{\boldsymbol{\phi}}^{(s+1)} - \hat{\boldsymbol{\phi}}^{(s)})^\top\right] = \begin{cases} H\eta^2\hat{\boldsymbol{S}}_0\boldsymbol{P}_{\|,\hat{\boldsymbol{S}}}\hat{\boldsymbol{S}}_0\boldsymbol{\Sigma}_0\hat{\boldsymbol{S}}_0\boldsymbol{P}_{\|,\hat{\boldsymbol{S}}}\hat{\boldsymbol{S}}_0 + \tilde{O}(\eta^{1.5-\beta}), & R_0 < s < R_{\mathrm{grp}} \\ \tilde{O}(\eta), & s \le R_0 \end{cases}
$$

*where $R_0 := \max\left\{\left\lceil\frac{10}{\lambda_{\max}\alpha}\log\frac{1}{\eta}\right\rceil, \left\lceil 2\log_{1/\beta}\frac{1}{\eta}\right\rceil\right\}$ and $\boldsymbol{P}_{\|,\hat{\boldsymbol{S}}} := \partial\Phi_{\hat{\boldsymbol{S}}}(\hat{\boldsymbol{\phi}}^{(0)})$.*

*Proof.* According to Lemma I.37 in Gu et al. [2023b], we could write the second moment for $\hat{\boldsymbol{\theta}}'$ as

$$
\mathbb{E}\left[(\hat{\boldsymbol{\phi}}^{'(s+1)} - \hat{\boldsymbol{\phi}}^{'(s)})(\hat{\boldsymbol{\phi}}^{'(s+1)} - \hat{\boldsymbol{\phi}}^{'(s)})^\top\right] = \begin{cases} H\eta^2\boldsymbol{\Sigma}'_{0,\|} + \tilde{O}(\eta^{1.5-\beta}), & R_0 < s < R_{\mathrm{grp}} \\ \tilde{\mathcal{O}}(\eta), & s \le R_0. \end{cases}
$$

Notice that

$$
\begin{aligned}
\boldsymbol{\Sigma}'_{0,\|} &:= \partial\Phi(\hat{\boldsymbol{\phi}}^{'(0)})\boldsymbol{\Sigma}'_0\partial\Phi(\hat{\boldsymbol{\phi}}^{'(0)}) \\
&= \boldsymbol{P}\partial\Phi_{\hat{\boldsymbol{S}}}(\hat{\boldsymbol{\phi}}^{(0)})\boldsymbol{P}\boldsymbol{P}\boldsymbol{\Sigma}_0\boldsymbol{P}\boldsymbol{P}\partial\Phi_{\hat{\boldsymbol{S}}}(\hat{\boldsymbol{\phi}}^{(0)})\boldsymbol{P}.
\end{aligned}
$$

When $R_0 \le s < R_{\mathrm{grp}}$,

$$
\begin{aligned}
\mathbb{E}\left[(\hat{\boldsymbol{\phi}}^{(s+1)} - \hat{\boldsymbol{\phi}}^{(s)})(\hat{\boldsymbol{\phi}}^{(s+1)} - \hat{\boldsymbol{\phi}}^{(s)})^\top\right] &= \mathbb{E}\left[\boldsymbol{P}(\hat{\boldsymbol{\phi}}^{'(s+1)} - \hat{\boldsymbol{\phi}}^{'(s)})(\hat{\boldsymbol{\phi}}^{'(s+1)} - \hat{\boldsymbol{\phi}}^{'(s)})^\top\boldsymbol{P}\right] \\
&= \hat{\boldsymbol{S}}_0\boldsymbol{P}_{\|,\hat{\boldsymbol{S}}}\hat{\boldsymbol{S}}_0\boldsymbol{\Sigma}_0\hat{\boldsymbol{S}}_0\boldsymbol{P}_{\|,\hat{\boldsymbol{S}}}\hat{\boldsymbol{S}}_0.
\end{aligned}
$$

The proof when $s \le R_0$ is a direct conclusion of Lemma I.37 in Gu et al. [2023b] since the $R_0 \propto \log\frac{1}{\eta}$ in our case. $\qquad\square$

Then we give the moment change of $\hat{\boldsymbol{\phi}}$ within a single giant step.

**Theorem G.1.** *Given* $\|\hat{\boldsymbol{\theta}}^{(0)} - \hat{\boldsymbol{\phi}}^{(0)}\|_2 = \mathcal{O}(\sqrt{\eta \log \frac{1}{\eta}})$*, for* $0 < \beta < 0.5$*, the first and second moments of* $\Delta \hat{\boldsymbol{\phi}}^{(R_{\mathrm{grp}})} := \hat{\boldsymbol{\phi}}^{(R_{\mathrm{grp}})} - \hat{\boldsymbol{\phi}}^{(0)}$ *are as follows:*

$$\mathbb{E}[\Delta \hat{\boldsymbol{\phi}}^{(R_{\mathrm{grp}})}] = \frac{\eta^{1-\beta}}{2} \hat{\boldsymbol{S}}_0 \partial^2 \Phi_{\hat{\boldsymbol{S}}_0}(\hat{\boldsymbol{\phi}}^{(0)})[\hat{\boldsymbol{S}}_0 \boldsymbol{\Sigma}_0 \hat{\boldsymbol{S}}_0] + \tilde{\mathcal{O}}(\eta^{1.5-2\beta}) + \tilde{\mathcal{O}}(\eta),$$

$$\mathbb{E}[\Delta \hat{\boldsymbol{\phi}}^{(R_{\mathrm{grp}})\top}] = \eta^{1-\beta} \hat{\boldsymbol{S}}_0 \boldsymbol{\Sigma}_\parallel(\hat{\boldsymbol{\phi}}^{(0)}, \hat{\boldsymbol{S}}^{(0)}) \hat{\boldsymbol{S}}_0 + \tilde{\mathcal{O}}(\eta^{1.5-1.5\beta}) + \tilde{\mathcal{O}}(\eta),$$

*where* $\boldsymbol{\Sigma}_\parallel(\boldsymbol{\phi}^{(0)}, \hat{\boldsymbol{S}}^{(0)}) := \boldsymbol{P}_{\parallel,\hat{\boldsymbol{S}}} \hat{\boldsymbol{S}}_0 \boldsymbol{\Sigma}_0 \hat{\boldsymbol{S}}_0 \boldsymbol{P}_{\parallel,\hat{\boldsymbol{S}}}.$

*Proof.* First we prove the first moment change as

$$\mathbb{E}[\Delta \hat{\boldsymbol{\phi}}^{(R_{\mathrm{grp}})}] = \mathbb{E}[\sum_{s=0}^{R_{\mathrm{grp}}-1} \hat{\boldsymbol{\phi}}^{(s+1)} - \hat{\boldsymbol{\phi}}^{(s)}]$$

$$= \sum_{s=0}^{R_0} \mathbb{E}[\hat{\boldsymbol{\phi}}^{(s+1)} - \hat{\boldsymbol{\phi}}^{(s)}] + \sum_{s=R_0+1}^{R_{\mathrm{grp}}-1} \mathbb{E}[\hat{\boldsymbol{\phi}}^{(s+1)} - \hat{\boldsymbol{\phi}}^{(s)}]$$

$$= \frac{\eta^{1-\beta}}{2} \hat{\boldsymbol{S}}_0 \partial^2 \Phi_{\hat{\boldsymbol{S}}_0}(\hat{\boldsymbol{\phi}}^{(0)})[\hat{\boldsymbol{S}}_0 \boldsymbol{\Sigma}_0 \hat{\boldsymbol{S}}_0] + \tilde{\mathcal{O}}(\eta^{1.5-2\beta}) + \tilde{\mathcal{O}}(\eta).$$

The last equation is a direct conclusion of Corollary G.2.

And for the second moment, we have

$$\mathbb{E}\left[\left(\sum_{s=0}^{R_{\mathrm{grp}}-1} \hat{\boldsymbol{\phi}}^{(s+1)} - \hat{\boldsymbol{\phi}}^{(s)}\right)\left(\sum_{s=0}^{R_{\mathrm{grp}}-1} \hat{\boldsymbol{\phi}}^{(s+1)} - \hat{\boldsymbol{\phi}}^{(s)}\right)^\top\right]$$

$$= \sum_{s=0}^{R_{\mathrm{grp}}-1} \mathbb{E}[(\hat{\boldsymbol{\phi}}^{(s+1)} - \hat{\boldsymbol{\phi}}^{(s)})(\hat{\boldsymbol{\phi}}^{(s+1)} - \hat{\boldsymbol{\phi}}^{(s)})^\top]$$

$$+ \sum_{s \neq s'} \mathbb{E}[(\hat{\boldsymbol{\phi}}^{(s+1)} - \hat{\boldsymbol{\phi}}^{(s)})]\mathbb{E}[(\hat{\boldsymbol{\phi}}^{(s'+1)} - \hat{\boldsymbol{\phi}}^{(s')})^\top]$$

$$= \eta^{1-\beta} \hat{\boldsymbol{S}}_0 \boldsymbol{\Sigma}_\parallel(\hat{\boldsymbol{\phi}}^{(0)}, \hat{\boldsymbol{S}}^{(0)}) \hat{\boldsymbol{S}}_0 + \tilde{\mathcal{O}}(\eta^{1.5-1.5\beta}) + \tilde{\mathcal{O}}(\eta),$$

where the last equation uses $\mathbb{E}[(\hat{\boldsymbol{\phi}}^{(s+1)} - \hat{\boldsymbol{\phi}}^{(s)})]\mathbb{E}[(\hat{\boldsymbol{\phi}}^{(s'+1)} - \hat{\boldsymbol{\phi}}^{(s')})^\top] = \tilde{\mathcal{O}}(\eta^2)$. $\qquad\square$

Next, we proceed with the updates of $\boldsymbol{v}$.

**Lemma G.12.** *Given* $c := \frac{1-\beta_2}{\eta^2}$*, and we have*

$$\mathbb{E}\left[\hat{\boldsymbol{v}}_0^{(R_{\mathrm{grp}})} - \hat{\boldsymbol{v}}_0^{(0)}\right] = c\eta^{1-\beta}\left(V\left(\boldsymbol{\Sigma}_0^{(0)}\right) - \hat{\boldsymbol{v}}_0^{(0)}\right) + \mathcal{O}\left(\eta^{1.5-1.5\beta}\right).$$

*Proof.* By the update rule of $\boldsymbol{v}$, we have

$$\hat{\boldsymbol{v}}_0^{(s+1)} - \hat{\boldsymbol{v}}_0^{(s)} = \hat{\boldsymbol{v}}_H^{(s)} - \hat{\boldsymbol{v}}_0^{(s)}$$

$$= \beta_2^H \hat{\boldsymbol{v}}_0^{(s)} + (1 - \beta_2)\sum_{i=1}^{H} \beta_2^{H-i} V\left(\hat{\boldsymbol{g}}_i^{(s)} \hat{\boldsymbol{g}}_i^{(s)\top}\right) - \hat{\boldsymbol{v}}_0^{(s)}$$

$$= \left(\beta_2^H - 1\right)\hat{\boldsymbol{v}}_0^{(0)} + (1 - \beta_2)\sum_{i=1}^{H} \beta_2^{H-i} V\left(\hat{\boldsymbol{g}}_i^{(s)} \hat{\boldsymbol{g}}_i^{(s)\top}\right).$$

Note that

$$\mathbb{E}\left[\hat{\boldsymbol{g}}_i^{(s)} \hat{\boldsymbol{g}}_i^{(s)\top}\right] = \mathbb{E}\left[\boldsymbol{\Sigma}(\hat{\boldsymbol{\theta}}_i^{(s)})\right]$$

$$= \mathbb{E}\left[\boldsymbol{\Sigma}(\hat{\boldsymbol{\phi}}_0^{(0)} + \boldsymbol{x}_i^{(s)})\right]$$

$$= \mathbb{E}\left[\boldsymbol{\Sigma}(\boldsymbol{\phi}_0^{(0)}) + \mathcal{O}\left(\eta^{0.5-0.5\beta}\right)\right]$$

$$= \boldsymbol{\Sigma}_0^{(0)} + \mathcal{O}\left(\eta^{0.5-0.5\beta}\right).$$

Combining with the linearity of $V$, we conclude that

$$\mathbb{E}\left[\hat{\boldsymbol{v}}_0^{(s+1)} - \hat{\boldsymbol{v}}_0^{(s)}\right] = \left(\beta_2^H - 1\right)\hat{\boldsymbol{v}}_0^{(0)} + \left(1 - \beta_2^H\right)\boldsymbol{V}\left(\boldsymbol{\Sigma}_0^{(0)}\right) + \mathcal{O}\left(\eta^{1.5-0.5\beta}\right)$$

$$\mathbb{E}\left[\hat{\boldsymbol{v}}_0^{(s+1)}\right] = \beta_2^H\hat{\boldsymbol{v}}_0^{(s)} + \left(1 - \beta_2^H\right)\boldsymbol{V}\left(\boldsymbol{\Sigma}_0^{(0)}\right) + \mathcal{O}\left(\eta^{1.5-0.5\beta}\right).$$

To transfer from $\hat{\boldsymbol{v}}_0^{(0)}$ to arbitrary $\hat{\boldsymbol{v}}_0^{(s)}$, we simply expand to get the result:

$$\mathbb{E}\left[\hat{\boldsymbol{v}}_0^{(s)}\right] = \beta_2^{sH}\hat{\boldsymbol{v}}_0^{(0)} + \left[\left(1 - \beta_2^H\right)V\left(\boldsymbol{\Sigma}_0^{(0)}\right) + \mathcal{O}\left(\eta^{1.5-0.5\beta}\right)\right]\left(1 + \beta_2^H + \beta_2^{2H} + \cdots + \beta_2^{(s-1)H}\right)$$

$$= \beta_2^{sH}\hat{\boldsymbol{v}}_0^{(0)} + \left[\left(1 - \beta_2^H\right)V\left(\boldsymbol{\Sigma}_0^{(0)}\right)\right]\left(\frac{1 - \beta_2^{sH}}{1 - \beta_2^H}\right) + \mathcal{O}\left(\eta^{1.5-0.5\beta}\right)\cdot\mathcal{O}\left(\eta^{-\beta}\right)$$

$$= \beta_2^{sH}\hat{\boldsymbol{v}}_0^{(0)} + \left(1 - \beta_2^{sH}\right)V\left(\boldsymbol{\Sigma}_0^{(0)}\right) + \mathcal{O}\left(\eta^{1.5-1.5\beta}\right).$$

Thus we have

$$\mathbb{E}\left[\hat{\boldsymbol{v}}_0^{(R_{\mathrm{grp}})} - \hat{\boldsymbol{v}}_0^{(0)}\right] = c\eta^{1-\beta}\left(V\left(\boldsymbol{\Sigma}_0^{(0)}\right) - \hat{\boldsymbol{v}}_0^{(0)}\right) + \mathcal{O}\left(\eta^{1.5-1.5\beta}\right).$$

where the last equation uses the fact that $1 - \beta_2^{R_{\mathrm{grp}}H} = 1 - (1 - c\eta^{1-\beta}) + O(\eta^{2-2\beta}) = c\eta + O(\eta^2)$. $\square$

Also, for the second moment change of $\hat{\boldsymbol{v}}$, we get the following lemma

**Lemma G.13.** *The second moment change of $\hat{\boldsymbol{v}}$ over a giant step is*

$$\mathbb{E}\left[\left(\hat{\boldsymbol{v}}_0^{(R_{\mathrm{grp}})} - \hat{\boldsymbol{v}}_0^{(0)}\right)\left(\hat{\boldsymbol{v}}_0^{(R_{\mathrm{grp}})} - \hat{\boldsymbol{v}}_0^{(0)}\right)^{\top}\right] = \mathcal{O}(\eta^{2-\beta}).$$

*Proof.*

$$\mathbb{E}\left[\left(\hat{\boldsymbol{v}}_0^{(s+1)} - \hat{\boldsymbol{v}}_0^{(s)}\right)\left(\hat{\boldsymbol{v}}_0^{(s+1)} - \hat{\boldsymbol{v}}_0^{(s)}\right)^{\top}\right] = \mathbb{E}\left[\left((\beta_2^H - 1) + (1 - \beta_2)\sum_{i=1}^{H}\beta_2^{H-i}V\left(\hat{\boldsymbol{g}}_i^{(s)}\hat{\boldsymbol{g}}_i^{(s)\top}\right)\right)\right.$$

$$\left.\left((\beta_2^H - 1) + (1 - \beta_2)\sum_{i=1}^{H}\beta_2^{H-i}V\left(\hat{\boldsymbol{g}}_i^{(s)}\hat{\boldsymbol{g}}_i^{(s)\top}\right)\right)^{\top}\right]$$

$$= \mathcal{O}\left((1 - \beta_2^H)^2\right) = \mathcal{O}\left(\eta^2\right).$$

$$\mathbb{E}\left[\left(\hat{\boldsymbol{v}}_0^{(R_{\mathrm{grp}})} - \hat{\boldsymbol{v}}_0^{(0)}\right)\left(\hat{\boldsymbol{v}}_0^{(R_{\mathrm{grp}})} - \hat{\boldsymbol{v}}_0^{(0)}\right)^{\top}\right] = \mathbb{E}\left[\left(\sum_{s=0}^{R_{\mathrm{grp}}-1}\left(\hat{\boldsymbol{v}}_0^{(s+1)} - \hat{\boldsymbol{v}}_0^{(s)}\right)\right)\left(\sum_{s=0}^{R_{\mathrm{grp}}-1}\left(\hat{\boldsymbol{v}}_0^{(s+1)} - \hat{\boldsymbol{v}}_0^{(s)}\right)^{\top}\right)\right]$$

$$= \sum_{s=0}^{R_{\mathrm{grp}}-1}\mathbb{E}\left[\left(\hat{\boldsymbol{v}}_0^{(s+1)} - \hat{\boldsymbol{v}}_0^{(s)}\right)\left(\hat{\boldsymbol{v}}_0^{(s+1)} - \hat{\boldsymbol{v}}_0^{(s)}\right)^{\top}\right]$$

$$+ \sum_{s\neq s'}\mathbb{E}\left[\left(\hat{\boldsymbol{v}}_0^{(s+1)} - \hat{\boldsymbol{v}}_0^{(s)}\right)\right]\mathbb{E}\left[\left(\hat{\boldsymbol{v}}_0^{(s'+1)} - \hat{\boldsymbol{v}}_0^{(s')}\right)^{\top}\right]$$

$$= \mathcal{O}(\eta^{2-\beta}).$$

The last equation uses

$$\mathbb{E}\left[\left(\hat{\boldsymbol{v}}_0^{(s+1)} - \hat{\boldsymbol{v}}_0^{(s)}\right)\left(\hat{\boldsymbol{v}}_0^{(s+1)} - \hat{\boldsymbol{v}}_0^{(s)}\right)^{\top}\right] = \mathcal{O}(\eta^2),$$

and

$$\mathbb{E}\left[\left(\hat{\boldsymbol{v}}_0^{(s+1)} - \hat{\boldsymbol{v}}_0^{(s)}\right)\right]\mathbb{E}\left[\left(\hat{\boldsymbol{v}}_0^{(s'+1)} - \hat{\boldsymbol{v}}_0^{(s')}\right)^{\top}\right] = \mathcal{O}(3 - 3\beta).$$

The above equation completes the proof. $\square$

## G.4 Weak Approximation

After we get the first and second moment changes within a giant step, we now utilize the moment calculation to prove the SDE approximation part of Theorem D.1. First, we recall our slow SDE for AGMs

$$
\begin{cases}
\mathrm{d}\boldsymbol{\zeta}(t) = P_{\boldsymbol{\zeta},\boldsymbol{S}(t)}\left(\boldsymbol{\Sigma}_{\parallel}^{1/2}(\boldsymbol{\zeta}(t);\boldsymbol{S}(t))\mathrm{d}\boldsymbol{W}_t - \tfrac{1}{2}\boldsymbol{S}(t)\nabla^3\mathcal{L}(\boldsymbol{\zeta})\left[\boldsymbol{\Sigma}_{\diamond}(\boldsymbol{\zeta}(t);\boldsymbol{S}(t))\right]\mathrm{d}t\right), \\
\mathrm{d}\boldsymbol{v}(t) = c\left(V(\boldsymbol{\Sigma}(\boldsymbol{\zeta})) - \boldsymbol{v}\right)\mathrm{d}t.
\end{cases}
$$

We then open the projection mapping $P_{\boldsymbol{\zeta},\boldsymbol{S}(t)}$ as

$$
\begin{cases}
\mathrm{d}\boldsymbol{\zeta} = \boldsymbol{S}(\boldsymbol{v})\partial\Phi_{\boldsymbol{S}(\boldsymbol{v})}(\boldsymbol{\zeta})\boldsymbol{S}(\boldsymbol{v})\boldsymbol{\Sigma}^{1/2}(\boldsymbol{\zeta})\mathrm{d}\boldsymbol{W}_t + \tfrac{1}{2}\boldsymbol{S}(\boldsymbol{v})\partial^2\Phi_{\boldsymbol{S}(\boldsymbol{v})}(\boldsymbol{\zeta})\left[\boldsymbol{S}(\boldsymbol{v})\boldsymbol{\Sigma}(\boldsymbol{\zeta})\boldsymbol{S}(\boldsymbol{v})\right]\mathrm{d}t, \\
\mathrm{d}\boldsymbol{v}(t) = c\left(V(\boldsymbol{\Sigma}(\boldsymbol{\zeta})) - \boldsymbol{v}\right)\mathrm{d}t.
\end{cases}
\tag{8}
$$

Now it suffices to prove the SDE in Equation (8) tracks the trajectory in AGMs within $\mathcal{O}(\frac{1}{\eta^2})$ steps in a weak approximation sense.

First, we have to show that the solution of Equation (8) in close in the minimizer manifold

**Lemma G.14.** *Let $\boldsymbol{X}(t) := (\boldsymbol{\zeta}(t)^\top, \boldsymbol{v}(t)^\top)^\top$ be the solution of Equation (8) with $\boldsymbol{\zeta}(0) \in \Gamma$, and $\boldsymbol{v}(0) \in \mathbb{R}^d$, then we have that $\boldsymbol{\zeta}(t) \in \Gamma$ for all $t \geq 0$.*

*Proof.* According to Filipović [2000], Du and Duan [2006], for a closed manifold $\mathcal{M}$ to be viable for the SDE $\mathrm{d}\boldsymbol{X}(t) = \boldsymbol{A}(\boldsymbol{X}(t))\mathrm{d}\boldsymbol{W}_t + \boldsymbol{b}(\boldsymbol{X}(t))\mathrm{d}t$, where $\boldsymbol{A}(\cdot) : \mathbb{R}^{d+D} \to \mathbb{R}^{(d+D)\times(d+D)}$ and $\boldsymbol{b}(\cdot) : \mathbb{R}^{d+D} \to \mathbb{R}^{d+D}$ are locally Lipchitz, it suffices to show that the following Nagumo type consistency condition holds:

$$
\mu(\boldsymbol{x}) := \boldsymbol{b}(\boldsymbol{x}) - \frac{1}{2}\sum_j D[A_j(\boldsymbol{x})]A_j(\boldsymbol{x}) \in T_{\boldsymbol{x}}(\mathcal{M}), \quad A_j(\boldsymbol{x}) \in T_{\boldsymbol{x}}(\mathcal{M}),
$$

where $D[\cdot]$ is the Jacobian operator and $A_j(\boldsymbol{x})$ denotes the $j$-th column of $A(\boldsymbol{x})$.

Following the argument in Gu et al. [2023b], here we also only need to show that $\boldsymbol{P}_{\perp,\boldsymbol{S}(\boldsymbol{v})}(\boldsymbol{x})\mu(\boldsymbol{x}) = 0$, where $\boldsymbol{P}_{\perp,\boldsymbol{S}(\boldsymbol{v})}(\boldsymbol{x}) := \boldsymbol{I}_d - \partial\Phi_{\boldsymbol{S}(\boldsymbol{v})}(\boldsymbol{x})$.

$$
\begin{aligned}
\boldsymbol{P}_{\perp,\boldsymbol{S}}(\boldsymbol{x})\sum_j D[A_j(\boldsymbol{x})]A_j(\boldsymbol{x}) &= \boldsymbol{P}_{\perp,\boldsymbol{S}}(\boldsymbol{x})\sum_j D\left[\partial\Phi_{\boldsymbol{S}}(\boldsymbol{x})\boldsymbol{S}\boldsymbol{\Sigma}_j^{1/2}\right]\partial\Phi_{\boldsymbol{S}}(\boldsymbol{x})\boldsymbol{S}\boldsymbol{\Sigma}_j^{1/2} \\
&= \boldsymbol{P}_{\perp,\boldsymbol{S}}(\boldsymbol{x})\boldsymbol{S}\sum_j \partial^2\Phi_{\boldsymbol{S}}(\boldsymbol{x})[\boldsymbol{S}\boldsymbol{\Sigma}_j^{1/2}, \boldsymbol{S}\partial\Phi_{\boldsymbol{S}}(\boldsymbol{x})\boldsymbol{S}\boldsymbol{\Sigma}_j^{1/2}] \\
&= -\boldsymbol{P}_{\perp,\boldsymbol{S}}(\boldsymbol{x})\boldsymbol{S}\boldsymbol{S}^{-1}\nabla^2\mathcal{L}(\boldsymbol{x})^\dagger\partial^2(\nabla\mathcal{L})(\boldsymbol{x})\left[\boldsymbol{S}\boldsymbol{\Sigma}_{\parallel}(\boldsymbol{x},\boldsymbol{S})\right].
\end{aligned}
$$

Notice that, since it is clear from the context, here we write $\boldsymbol{S} = \boldsymbol{S}$ for short. The last equation uses Lemma G.4. Agian, applying Lemma G.4 gives

$$
\boldsymbol{P}_{\perp,\boldsymbol{S}}(\boldsymbol{x})\boldsymbol{b}(\boldsymbol{x}) = -\frac{1}{2}\boldsymbol{P}_{\perp,\boldsymbol{S}}(\boldsymbol{x})\boldsymbol{S}\boldsymbol{S}^{-1}\nabla^2\mathcal{L}(\boldsymbol{x})^\dagger\partial^2(\nabla\mathcal{L})(\boldsymbol{x})\left[\boldsymbol{S}\boldsymbol{\Sigma}_{\parallel}(\boldsymbol{x},\boldsymbol{S})\right].
$$

The above equation completes the proof. $\qquad\square$

To establish Theorem D.1, we give an equivalent theorem, which captures the closeness of $\boldsymbol{X}(t)$ and $\bar{\boldsymbol{X}}_t$ in a long horizon. Also, for the proof of Theorem 4.1, it suffices to prove the following lemma, whose proof will be shown in Appendix G.5.

**Theorem G.2.** *If $\|\boldsymbol{\theta}^{(0)} - \phi^{(0)}\|_2 = \mathcal{O}(\sqrt{\eta\log\frac{1}{\eta}})$ and $\boldsymbol{\zeta}(0) = \phi^{(0)}$, $\boldsymbol{v}(0) = \boldsymbol{v}^{(0)}$, then for a giant step $R_{\mathrm{grp}} = \lfloor\frac{1}{\eta^{0.25}}\rfloor$, for every test function $g \in \mathcal{C}^3$,*

$$
\max_{0\leq n\leq\lfloor\frac{T}{\eta^{0.75}}\rfloor}\left|\mathbb{E}\left[g\left(\bar{\boldsymbol{X}}^{(nR_{\mathrm{grp}})}\right)\right] - \mathbb{E}\left[g\left(\boldsymbol{X}(n\eta^{0.75})\right)\right]\right| = C_g\eta^{0.25}(\log\frac{1}{\eta})^b,
$$

*where $C_g$ is a constant independent of $\eta$ but depends on $g(\cdot)$ and $b > 0$ is a universal constant independent of $g(\cdot)$ and $\eta$.*

### G.4.1 Preliminary and Additional Notations

We first introduce some notations and preliminary background. We consider the following stochastic gradient algorithms (SGAs)

$$\boldsymbol{x}_{n+1} = \boldsymbol{x}_n + \eta_e \boldsymbol{h}(\boldsymbol{x}_n, \boldsymbol{\xi}_n),$$

where $\boldsymbol{x}_n \in \mathbb{R}^{d+D}$ is the parameter vector, $\eta_e$ is the effective learning rate, $\boldsymbol{h}(\cdot, \cdot) : \mathbb{R}^{d+D} \times \mathbb{R}^{d+D} \to \mathbb{R}^{d+D}$ depend on the current parameter vector $\boldsymbol{x}_n$ and the noise vector $\boldsymbol{\xi}_n$ sampled from some distribution $\Xi(\boldsymbol{x}_n)$.

We also consider the Stochastic Differential Equation (SDE) of the following form:

$$\mathrm{d}\boldsymbol{X}_t = \boldsymbol{b}(\boldsymbol{X}_t, t)\mathrm{d}t + \sigma(\boldsymbol{X}_t, t)\mathrm{d}\boldsymbol{W}_t,$$

where $\boldsymbol{b} : \mathbb{R}^{d+D} \times \mathbb{R}^+ \to \mathbb{R}^{d+D}$ is the drift vector function and $\sigma : \mathbb{R}^{d+D} \times \mathbb{R}^+ \to \mathbb{R}^{(d+D) \times (d+D)}$ is the diffusion matrix function.

According to the moment calculations in Corollary G.2,Lemma G.11, Lemma G.12, and Lemma G.13, we set $\eta_e = \eta^{1-\beta}$, and

$$\boldsymbol{b}(\boldsymbol{X}_t, t) = \left( \left( \frac{1}{2} \partial^2 \Phi_{\boldsymbol{S}(\boldsymbol{v})}(\boldsymbol{\zeta}) \left[ \boldsymbol{\Sigma}(\boldsymbol{\zeta}, \boldsymbol{S}(\boldsymbol{v})) \right] \right)^\top, c \left( V(\boldsymbol{\Sigma}(\boldsymbol{\zeta})) - \boldsymbol{v} \right)^\top \right)^\top,$$

$$\sigma(\boldsymbol{X}_t, t) = \begin{pmatrix} \partial \Phi_{\boldsymbol{S}(\boldsymbol{v})}(\boldsymbol{\zeta}) \boldsymbol{\Sigma}^{1/2}(\boldsymbol{\zeta}, \boldsymbol{S}(\boldsymbol{v})), & \boldsymbol{0} \\ \boldsymbol{0}, & \boldsymbol{0} \end{pmatrix}.$$

Next, we are going to define the one giant step change of the parameter, both for SGAs and SDE.

$$\hat{\bar{\boldsymbol{X}}}^{(lR_{\mathrm{grp}})} := \left( \Phi_{\hat{\boldsymbol{S}}^{(lR_{\mathrm{grp}})}} \left( \hat{\boldsymbol{\theta}} \right)^\top, \boldsymbol{v}^{l\hat{R}_{\mathrm{grp}}}^\top \right)^\top \in \mathbb{R}^{d+D}, \quad \Delta^{(n)} := \hat{\bar{\boldsymbol{X}}}^{((n+1)R_{\mathrm{grp}})} - \hat{\bar{\boldsymbol{X}}}^{(nR_{\mathrm{grp}})},$$

$$\tilde{\Delta}^{(n)} := \boldsymbol{X}_{(n+1)\eta_e} - \hat{\bar{\boldsymbol{X}}}^{(nR_{\mathrm{grp}})}, \quad \boldsymbol{b}^{(n)} := \boldsymbol{b}(\hat{\bar{\boldsymbol{X}}}^{(nR_{\mathrm{grp}})}), \quad \sigma^{(n)} := \sigma(\hat{\bar{\boldsymbol{X}}}^{(nR_{\mathrm{grp}})}).$$

We now give a lemma to give the approximation of the first, second, and higher-order moment change of the SDE.

**Lemma G.15.** *There exists a positive constant $c_0$ independent of $\eta_e$ and $g$ such that for all $\boldsymbol{\zeta} \in \Gamma$, it holds for all $1 \le i \le d$ that*

$$\left| \mathbb{E}[\tilde{\Delta}_i(\boldsymbol{\zeta}, n)] - \eta_e b_i(\boldsymbol{\zeta}) \right| \le c_0 \eta_e^2,$$

$$\left| \mathbb{E}[\tilde{\Delta}_i(\boldsymbol{\zeta}, n)\tilde{\Delta}_j(\boldsymbol{\zeta}, n)] - \eta_e \sum_{l=1}^d \sigma_{i,l}(\boldsymbol{\zeta})\sigma_{l,j}(\boldsymbol{\zeta}) \right| \le c_0 \eta_e^2,$$

$$\mathbb{E}\left[ \left| \prod_{s=1}^6 \tilde{\Delta}_{i_s}(\boldsymbol{\zeta}, n) \right| \right] \le c_0 \eta_e^3.$$

*Proof.* (i) By Lemma G.14, the first half solution $\boldsymbol{\zeta}(t)$ in $\boldsymbol{X}(t)$ of Equation (8) stays in the manifold almost surely when $\boldsymbol{\zeta}(0) \in \Gamma$. (ii) We assume that $\mathcal{L} \in \mathcal{C}^5$, so $\boldsymbol{b}, \sigma \in \mathcal{C}^4$. (iii) We know that $\Gamma$ is compact by Assumption 3.4. Then we can directly apply Lemma B.3 in Malladi et al. [2022] and Lemma 26 in Li et al. [2019]. □

**Lemma G.16** (Adaption of Lemma I.41 in Gu et al. [2023b])**.** *Given drift term and diffusion term $\boldsymbol{b}, \sigma \in G^\alpha$ and Lipschitz. Let $s \in [0, T]$ and $g \in G^\alpha$. Then for $t \in [s, T]$, we can define:*

$$u(\boldsymbol{x}, s, t) := \mathbb{E}_{\boldsymbol{X}_t \sim \mathcal{P}_X(\boldsymbol{x}, s, t)}[g(\boldsymbol{X}_t)].$$

*where $\mathcal{P}_X(\boldsymbol{x}, s, t)$ denotes the distribution of $\boldsymbol{X}_t$ with the initial condition $\boldsymbol{X}(s) = \boldsymbol{x}$. Then $u(\cdot, s, t) \in G^\alpha$ uniformly in $s, t$.*

### G.4.2 Proof of the Approximation for Slow SDE of AGMs

For the giant step constant $\beta \in (0, 0.5)$, we define several quantities $a_1 = \frac{1.5-2\beta}{1-\beta} \in (1, 1.5)$, $a_2 = \frac{1}{1-\beta} \in (1, 2)$, $a_3 = \frac{1.5-1.5\beta}{1-\beta} = 1.5$, and $a_4 = \frac{2-2\beta}{1-\beta} = 2$. In this part, we will show that only $a_1$ and $a_2$ would impact the error bound in our approximation theorem.

The following lemma captures the difference between the SDEs' and the AGMs' first and second moment changes, as a key step to control the approximation error, utilizing the moment calculation results from the last section.

**Lemma G.17.** *If* $\|\boldsymbol{\theta}^{(0)} - \phi^{(0)}\|_2 = \mathcal{O}(\sqrt{\eta \log \frac{1}{\eta}})$, *then it holds for all* $0 \le n \le \lfloor T/\eta_e \rfloor$ *and* $1 \le i \le d$ *that*

$$\left| \mathbb{E}[\Delta_i^{(n)} - \tilde{\Delta}_i^{(n)} \mid \mathcal{E}_0^{(nR_{\mathrm{grp}})}] \right| \le c_1 \left( \eta_e^{a_1} (\log \frac{1}{\eta_e})^b + \eta_e^{a_2} (\log \frac{1}{\eta_e})^b \right),$$

$$\left| \mathbb{E}[\Delta_i^{(n)} \Delta_j^{(n)} - \tilde{\Delta}_i^{(n)} \tilde{\Delta}_j^{(n)} \mid \mathcal{E}_0^{(nR_{\mathrm{grp}})}] \right| \le c_1 \left( \eta_e^{a_1} (\log \frac{1}{\eta_e})^b + \eta_e^{a_2} (\log \frac{1}{\eta_e})^b \right),$$

$$\mathbb{E}\left[ \left| \prod_{s=1}^6 \Delta_{i_s}^{(n)} \mid \mathcal{E}^{(nR_{\mathrm{grp}})} \right| \right] \le c_1^2 \eta_e^{2a_1} (\log \frac{1}{\eta_e})^{2b},$$

$$\mathbb{E}\left[ \left| \prod_{s=1}^6 \tilde{\Delta}_{i_s}^{(n)} \mid \mathcal{E}^{(nR_{\mathrm{grp}})} \right| \right] \le c_1^2 \eta_e^{2a_1} (\log \frac{1}{\eta_e})^{2b},$$

*where* $c_1$ *and* $b$ *are constants independent of* $\eta_e$ *and* $g$.

*Proof.* According to Appendix G.2, we have that

$$\mathbb{E}\left[ \left| \prod_{s=1}^6 \Delta_{i_s}^{(n)} \mid \mathcal{E}^{(nR_{\mathrm{grp}})} \right| \right] = \mathcal{O}(\eta^{3-3\beta}).$$

We can further use Corollary G.2, Lemma G.11, Lemma G.12, and Lemma G.13, which gives

$$\left| \mathbb{E}[\Delta_i^{(n)} - \eta_e b_i^{(n)}] \right| \le c_2 \left( \eta_e^{a_1} (\log \frac{1}{\eta_e})^b + \eta_e^{a_2} (\log \frac{1}{\eta_e})^b \right), \tag{9}$$

$$\left| \mathbb{E}[\Delta_i^{(n)} \Delta_j^{(n)} - \eta_e \sum_{l=1}^d \sigma_{i,l}^{(n)} \sigma_{l,j}^{(n)}] \right| \le c_2 \left( \eta_e^{a_1} (\log \frac{1}{\eta_e})^b + \eta_e^{a_2} (\log \frac{1}{\eta_e})^b \right) \tag{10}$$

$$\mathbb{E}\left[ \left| \prod_{s=1}^6 \Delta_{i_s}^{(n)} \right| \right] \le c_2^2 \eta_e^{2a_1} (\log \frac{1}{\eta_e})^{2b}. \tag{11}$$

Notice that the above equations uses $a_1 < a_3$ and $a_2 < a_4$ for all $\beta \in (0, 0.5)$. These three equations and Lemma G.15 give the Lemma. $\qquad\square$

**Lemma G.18.** *For a test function* $g \in \mathcal{C}^3$, *and we define* $u_{l,n}(\boldsymbol{x}) := u(\boldsymbol{x}, l\eta_e, n\eta_e) = \mathbb{E}_{\boldsymbol{X}_t \sim \mathcal{P}(\boldsymbol{x}, l\eta_e, n\eta_e)}[g(\boldsymbol{X}_t)]$. *If* $\|\boldsymbol{\theta}^{(0)} - \phi^{(0)}\|_2 = \mathcal{O}(\sqrt{\eta \log \frac{1}{\eta}})$, *then for all* $0 \le l \le n - 1$, *and* $1 \le n \le \lfloor T/\eta_e \rfloor$, *it holds that*

$$\left| \mathbb{E}[u_{l+1,n}(\bar{\boldsymbol{X}}^{(lR_{\mathrm{grp}})} + \Delta^{(l)}) - u_{l+1,n}(\bar{\boldsymbol{X}}^{(lR_{\mathrm{grp}})} + \tilde{\Delta}^{(l)}) \mid \bar{\boldsymbol{X}}^{(lR_{\mathrm{grp}})}] \right| \le C_{g,3} (\eta_e^{a_1} + \eta_e^{a_2}) \log(\frac{1}{\eta_e})^b,$$

*where* $C_{g,3}$ *is some positive constant independent of* $\eta_e$ *but can depend on* $g$.

*Proof.* Given $g \in \mathcal{C}^3$, by Lemma G.16, we have $u_{l,n}(\boldsymbol{x}) \in \mathcal{C}^3$ for all $l$ and $n$. Which is to say that there exists a function $Q(\cdot) \in G$, such that the partial derivative of $u_{l,n}(\boldsymbol{X})$ with respect to $l, n, \boldsymbol{x}$ up

to the third order is bounded by $Q(\boldsymbol{x})$. By the law of total expectation and triangle inequality,

$$
\left| \mathbb{E}[u_{l+1,n}(\hat{\bar{\boldsymbol{X}}}^{(lR_{\mathrm{grp}})} + \Delta^{(l)}) - u_{l+1,n}(\hat{\bar{\boldsymbol{X}}}^{(lR_{\mathrm{grp}})} + \tilde{\Delta}^{(l)}) \mid \hat{\bar{\boldsymbol{X}}}^{(lR_{\mathrm{grp}})}] \right|
$$

$$
\leq \underbrace{\left| \mathbb{E}[u_{l+1,n}(\hat{\bar{\boldsymbol{X}}}^{(lR_{\mathrm{grp}})} + \Delta^{(l)}) - u_{l+1,n}(\hat{\bar{\boldsymbol{X}}}^{(lR_{\mathrm{grp}})} + \tilde{\Delta}^{(l)}) \mid \hat{\bar{\boldsymbol{X}}}^{(lR_{\mathrm{grp}})}, \mathcal{E}_0^{(lR_{\mathrm{grp}})}] \right|}_{I_1}
$$

$$
+ \eta^{100} \underbrace{\mathbb{E}[\left| u_{l+1,n}(\hat{\bar{\boldsymbol{X}}}^{(lR_{\mathrm{grp}})} + \Delta^{(l)}) \right| \mid \hat{\bar{\boldsymbol{X}}}^{(lR_{\mathrm{grp}})}, \mathcal{E}_0^{(lR_{\mathrm{grp}})}]}_{I_2}
$$

$$
+ \eta^{100} \underbrace{\mathbb{E}[\left| u_{l+1,n}(\hat{\bar{\boldsymbol{X}}}^{(lR_{\mathrm{grp}})} + \tilde{\Delta}^{(l)}) \right| \mid \hat{\bar{\boldsymbol{X}}}^{(lR_{\mathrm{grp}})}, \mathcal{E}_0^{(lR_{\mathrm{grp}})}]}_{I_3}.
$$

For $I_2$ and $I_3$, due to the compactness of $\Gamma$ and $\boldsymbol{v} \preceq R_1$ from Assumption 3.3, $Q(\boldsymbol{x})$ can be bounded for some constant $C_{g,4}$ independent of $\eta_e$ but could depend on test function $g$. Hence, we have that $I_2 + I_3 \leq C_{g,4}\eta^{100}$.

Using the triangle inequality, we first decompose $I_1$ into several terms as

$$
I_1 \leq \underbrace{\sum_{i=1}^{d} \left| \mathbb{E}\left[ \frac{\partial u_{l,n}}{\partial X_i}(\hat{\bar{\boldsymbol{X}}}^{(R_{\mathrm{grp}})}) \left( \Delta_i^{(l)} - \tilde{\Delta}_i^{(l)} \right) \mid \hat{\bar{\boldsymbol{X}}}^{(lR_{\mathrm{grp}})}, \mathcal{E}_0^{(lR_{\mathrm{grp}})} \right] \right|}_{I_{1,1}}
$$

$$
+ \underbrace{\frac{1}{2} \sum_{1 \leq i,j \leq d} \left| \mathbb{E}\left[ \frac{\partial^2 u_{l,n}}{\partial X_i \partial X_j}(\hat{\bar{\boldsymbol{X}}}^{(R_{\mathrm{grp}})}) \left( \Delta_j^{(l)} \Delta_i^{(l)} - \tilde{\Delta}_i^{(l)} \tilde{\Delta}_j^{(l)} \right) \mid \hat{\bar{\boldsymbol{X}}}^{(lR_{\mathrm{grp}})}, \mathcal{E}_0^{(lR_{\mathrm{grp}})} \right] \right|}_{I_{1,2}}
$$

$$
+ |\mathcal{R}| + |\tilde{\mathcal{R}}|,
$$

where the third order remainders $\mathcal{R}$ and $\tilde{\mathcal{R}}$ are

$$
\mathcal{R} = \frac{1}{6} \sum_{1 \leq i,j,k \leq d} \left| \mathbb{E}\left[ \frac{\partial^3 u_{l,n}}{\partial X_i \partial X_j \partial X_k}(\hat{\bar{\boldsymbol{X}}}^{(R_{\mathrm{grp}})} + \alpha\Delta^{(l)}) \left( \Delta_j^{(l)} \Delta_i^{(l)} \Delta_k^{(l)} \right) \mid \hat{\bar{\boldsymbol{X}}}^{(lR_{\mathrm{grp}})}, \mathcal{E}_0^{(lR_{\mathrm{grp}})} \right] \right|
$$

$$
\tilde{\mathcal{R}} = \frac{1}{6} \sum_{1 \leq i,j,k \leq d} \left| \mathbb{E}\left[ \frac{\partial^3 u_{l,n}}{\partial X_i \partial X_j \partial X_k}(\hat{\bar{\boldsymbol{X}}}^{(R_{\mathrm{grp}})} + \tilde{\alpha}\tilde{\Delta}^{(l)}) \left( \tilde{\Delta}_j^{(l)} \tilde{\Delta}_i^{(l)} \tilde{\Delta}_k^{(l)} \right) \mid \hat{\bar{\boldsymbol{X}}}^{(lR_{\mathrm{grp}})}, \mathcal{E}_0^{(lR_{\mathrm{grp}})} \right] \right|,
$$

where $\alpha, \tilde{\alpha} \in (0, 1)$. Again, notice that the $\Gamma$ is compact and $vv \preceq R_1$, thus we can bound the derivatives of $u_{l,n}(\boldsymbol{x})$ for any $\boldsymbol{X}$ as

$$
\left| \frac{\partial u_{l+1,n}}{\partial \boldsymbol{X}_i}(\boldsymbol{X}) \right| \leq C_{g,4}, \ \left| \frac{\partial^2 u_{l+1,n}}{\partial \boldsymbol{X}_i \partial \boldsymbol{X}_j}(\boldsymbol{X}) \right| \leq C_{g,4}, \ \left| \frac{\partial^3 u_{l+1,n}}{\partial \boldsymbol{X}_i \partial \boldsymbol{X}_j \partial \boldsymbol{X}_k}(\boldsymbol{X}) \right| \leq C_{g,4}. \tag{12}
$$

For the term $I_{1,1}$ and $I_{1,2}$, by applying Lemma G.17, we have that

$$
I_{1,1} \leq d c_1 C_{g,4}(\eta_e^{a_1} + \eta_e^{a_2})(\log \frac{1}{\eta_e})^b, \ I_{1,2} \leq \frac{d^2}{2} c_1 C_{g,4}(\eta_e^{a_1} + \eta_e^{a_2})(\log \frac{1}{\eta_e})^b.
$$

Next, we bound the remainders $\mathcal{R}$ and $\tilde{\mathcal{R}}$. By Cauchy-Schwarz inequality,

$$
|\mathcal{R}| \leq \frac{1}{6} \sum_{1 \leq i,j,k \leq d} \sqrt{\mathbb{E}\left[ \left( \frac{\partial^3 u_{l,n}}{\partial X_i \partial X_j \partial X_k}(\hat{\bar{\boldsymbol{X}}}^{(R_{\mathrm{grp}})} + \alpha\Delta^{(l)}) \right)^2 \mid \hat{\bar{\boldsymbol{X}}}^{(lR_{\mathrm{grp}})}, \mathcal{E}_0^{(lR_{\mathrm{grp}})} \right]} \times
$$

$$
\sqrt{\mathbb{E}\left[ \left( \Delta_j^{(l)} \Delta_i^{(l)} \Delta_k^{(l)} \right)^2 \mid \hat{\bar{\boldsymbol{X}}}^{(lR_{\mathrm{grp}})}, \mathcal{E}_0^{(lR_{\mathrm{grp}})} \right]}
$$

$$
\leq \frac{d^3}{6} C_{g,4} c_1 \eta_e^{a_1} \log(\frac{1}{\eta_e})^b,
$$

where the last inequality uses Lemma G.17 and Equation (12).

Similarly, we can prove that there exists a positive constant $C_{g,5}$ such that

$$|\tilde{\mathcal{R}}| \leq \frac{d^3}{6} C_{g,5} c_1 \eta_e^{a_1} \log(\frac{1}{\eta_e})^b.$$

Combining the bounds for $I_1$, $I_2$, and $I_3$ gives the lemma. $\qquad\square$

### G.5 Proof of Theorem G.2

Finally, we are ready to prove Theorem G.2.

*Proof of Theorem G.2.* For $0 \leq l \leq n = \lfloor \frac{T}{\eta^{0.75}} \rfloor$, we denote the random variable by $\hat{x}_{l,n}$ such that follows a distribution $\mathcal{P}_{\boldsymbol{X}}(\hat{\bar{\boldsymbol{X}}}^{(lR_{\text{grp}})}, l\eta_e, n\eta_e)$. When we set $l = n$, $\mathcal{P}(\hat{x}_{n,n} = \hat{\bar{\boldsymbol{X}}}^{(nR_{\text{grp}})})$ and setting $l = 0$ gives $\hat{x}_{0,n} \sim \boldsymbol{X}(n\eta_e)$. Recall the previous definition that $u(\boldsymbol{x}, s, t) = \mathbb{E}_{\boldsymbol{X}_t \sim \mathcal{P}_{\boldsymbol{X}}(\boldsymbol{x},s,t)}[g(\boldsymbol{X}_t)]$, and we define that $\mathcal{T}_{l+1,n} := u_{l+1,n}(\hat{\bar{\boldsymbol{X}}}^{(lR_{\text{grp}})} + \Delta^{(l)}, (l+1)\eta_e, n\eta_e) - u_{l+1,n}(\hat{\bar{\boldsymbol{X}}}^{(lR_{\text{grp}})} + \tilde{\Delta}^{(l)}, (l+1)\eta_e, n\eta_e)$. Using the definition of $\boldsymbol{x}_{l,n}$, we can rewrite the distance between AGMs and SDE measured by a test function $g$ as

$$\left| \mathbb{E}\left[ g(\bar{\boldsymbol{X}}^{(nR_{\text{grp}})}) - g(\boldsymbol{X}(n\eta_e)) \right] \right|$$
$$\leq \left| \mathbb{E}\left[ g(\boldsymbol{x}_{n,n}) - g(\boldsymbol{x}_{0,n}) \mid \mathcal{E}_0^{(nR_{\text{grp}})} \right] \right| + \mathcal{O}(\eta^{100}).$$

The above equation uses the law of total expectation and the definition of $\delta$-good event $\mathcal{E}_0^{(nR_{\text{grp}})}$ in Definition G.1. Then the Triangle inequality gives

$$\left| \mathbb{E}\left[ g(\boldsymbol{x}_{n,n}) - g(\boldsymbol{x}_{0,n}) \mid \mathcal{E}_0^{(nR_{\text{grp}})} \right] \right| \leq \sum_{l=0}^{n-1} \left| \mathbb{E}\left[ g(\hat{\boldsymbol{x}}_{l+1,n}) - g(\hat{\boldsymbol{x}}_{l,n}) \mid \mathcal{E}_0^{(nR_{\text{grp}})} \right] \right| + \mathcal{O}(\eta^{100})$$

$$= \sum_{l=0}^{n-1} \left| \mathbb{E}\left[ \mathcal{T}_{l+1,n} \mid \mathcal{E}_0^{(nR_{\text{grp}})} \right] \right| + \mathcal{O}(\eta^{100})$$

$$= \sum_{l=0}^{n-1} \left| \mathbb{E}\left[ \mathbb{E}\left[ \mathcal{T}_{l+1,n} \mid \hat{\bar{\boldsymbol{X}}}^{(lR_{\text{grp}})}, \mathcal{E}_0^{(nR_{\text{grp}})} \right] \mid \mathcal{E}_0^{(nR_{\text{grp}})} \right] \right| + \mathcal{O}(\eta^{100})$$

$$\leq \sum_{l=0}^{n-1} \mathbb{E}\left[ \left| \mathbb{E}\left[ \mathcal{T}_{l+1,n} \mid \hat{\bar{\boldsymbol{X}}}^{(lR_{\text{grp}})}, \mathcal{E}_0^{(nR_{\text{grp}})} \right] \right| \mid \mathcal{E}_0^{(nR_{\text{grp}})} \right] + \mathcal{O}(\eta^{100})$$

$$\leq n C_{g,3}(\eta_e^{a_1} + \eta_e^{a_2}) \log(\frac{1}{\eta_e})^b$$

$$\leq T C_{g,3}(\eta_e^{a_1-1} + \eta_e^{a_2-1}) \log(\frac{1}{\eta_e})^b.$$

where the second last inequality uses Lemma G.18. Recall that $a_1 = \frac{1.5-2\beta}{1-\beta}$, $a_2 = \frac{1}{1-\beta}$, $\beta \in (0, 0.5)$. Let $\beta = 0.25$, and we complete the proof. $\qquad\square$

## H  Proof of Theorems in Appendix B

### H.1  Proof of Adam and AdamE's Implicit Biases with Label Noise

In this part, we give the proof of Theorem B.1, Lemma B.1 and Lemma B.2.

*Proof of Theorem B.1.* Recall the SDE formula in Equation (8) and Lemma G.10:

$$\begin{cases} d\boldsymbol{\zeta}(t) = \partial\Phi_{\boldsymbol{S}(\boldsymbol{v})}(\boldsymbol{\zeta})\boldsymbol{S}(\boldsymbol{v})\boldsymbol{\Sigma}^{1/2}(\boldsymbol{\zeta})d\boldsymbol{W}_t - \frac{1}{2}\boldsymbol{S}_t\partial\Phi_{\boldsymbol{S}(\boldsymbol{v})}(\boldsymbol{\zeta})\boldsymbol{S}_t\partial^2(\nabla\mathcal{L})(\boldsymbol{\zeta})[\boldsymbol{P}\mathcal{V}_{\nabla^2\mathcal{L}'(\phi'_{(0)})}(\boldsymbol{P}\boldsymbol{\Sigma}_0\boldsymbol{P})\boldsymbol{P}]dt, \\ d\boldsymbol{v}(t) = c\left(V(\boldsymbol{\Sigma}(\boldsymbol{\zeta})) - \boldsymbol{v}\right)dt. \end{cases}$$

Plugging in the following:

- The definition that $\boldsymbol{P} := \boldsymbol{S}_0^{1/2}$,

- Lemma G.3: For any $\boldsymbol{\zeta} \in \Gamma$ and $p.d$ matrix $\boldsymbol{S}$, $\partial\Phi_{\boldsymbol{S}}(\boldsymbol{\zeta})\boldsymbol{S}\nabla^2\mathcal{L}(\boldsymbol{\zeta}) = \boldsymbol{0}$,

- The label noise condition: $\boldsymbol{\Sigma}(\boldsymbol{\zeta}) := \alpha\nabla^2\mathcal{L}(\boldsymbol{\zeta})$ for any $\boldsymbol{\zeta} \in \Gamma$ and some constant $\alpha > 0$.

yields the final result:

$$\begin{cases} \mathrm{d}\boldsymbol{\zeta}(t) = -\frac{\alpha}{2}\boldsymbol{S}_t\partial\Phi_{\boldsymbol{S}_t}(\boldsymbol{\zeta})\boldsymbol{S}_t\partial^2(\nabla\mathcal{L})(\boldsymbol{\zeta})[\boldsymbol{S}_t]\mathrm{d}t, \\ \mathrm{d}\boldsymbol{v}(t) = c\left(V(\boldsymbol{\Sigma}(\boldsymbol{\zeta})) - \boldsymbol{v}\right)\mathrm{d}t. \end{cases} \tag{13}$$

The above equation completes the proof. $\qquad\square$

For Lemma B.1 and Lemma B.2, we first present the following formal statements and give the corresponding proofs for each of them.

**Lemma H.1** (Adam's Implicit Bias under Label Noise and $\epsilon = 0$). *With the label noise condition, every fixed point of Equation (3) for Adam with $\epsilon = 0$ satisfies $\nabla_\Gamma tr\left(\mathrm{Diag}(\boldsymbol{H})^{1/2}\right) = 0$, where $\nabla_\Gamma f$ stands for the gradient of a function $f$ projected to the tangent space of $\Gamma$.*

*Proof of Lemma H.1.* Consider the a fixed point $(\boldsymbol{\zeta}^*, \boldsymbol{v}^*)$ of the ODE (13). It must satisfy

$$S(\boldsymbol{v}^*)\partial\Phi_{S(\boldsymbol{v}^*)}(\boldsymbol{\zeta}^*)S(\boldsymbol{v}^*)\nabla^3\mathcal{L}(\boldsymbol{\zeta}^*)[S(\boldsymbol{v}^*)] = \boldsymbol{0}, \tag{14}$$

and

$$\boldsymbol{v}^* = V(\boldsymbol{\Sigma}(\boldsymbol{\zeta}^*)). \tag{15}$$

First, we simplify the notation by denoting the following:

$$\boldsymbol{S}^* := S(\boldsymbol{v}^*), \quad \boldsymbol{P}_{\parallel}^* := \partial\Phi_{S(\boldsymbol{v}^*)}\boldsymbol{S}^*, \quad \boldsymbol{H}^* = \nabla^2\mathcal{L}(\boldsymbol{\zeta}^*).$$

Then Equation (14) becomes

$$\boldsymbol{S}^*\boldsymbol{P}_{\parallel}^*\nabla^3\mathcal{L}(\boldsymbol{\zeta}^*)[\boldsymbol{S}^*] = \boldsymbol{0}.$$

Since $\boldsymbol{v}^* = V(\boldsymbol{\Sigma}(\boldsymbol{\zeta}^*))$ and $\boldsymbol{\Sigma}(\boldsymbol{\zeta}^*) = \alpha\boldsymbol{H}^*$, we have that $\boldsymbol{v}^* = \mathrm{diag}(\boldsymbol{\Sigma}(\boldsymbol{\zeta}^*)) = \alpha\mathrm{diag}(\boldsymbol{H}^*)$. Then $\boldsymbol{S}^* = \mathrm{Diag}(\frac{1}{(\alpha\mathrm{diag}(\boldsymbol{H}^*))^{1/2}})$, and we can rewrite $\nabla^3\mathcal{L}(\boldsymbol{\zeta}^*)[\boldsymbol{S}^*]$ as

$$\nabla^3\mathcal{L}(\boldsymbol{\zeta}^*)[\boldsymbol{S}^*] = \sum_{j=1}^{d}\frac{1}{(\alpha H_{jj}^*)^{1/2}}\nabla(H_{jj}^*) = \frac{2}{\sqrt{\alpha}}\sum_{j=1}^{d}\nabla((H_{jj}^*)^{1/2}) = \frac{2}{\sqrt{\alpha}}\nabla tr\left(\mathrm{Diag}(\boldsymbol{H}^*)^{1/2}\right).$$

Therefore, Equation (14) is equivalent to

$$\boldsymbol{S}^*\boldsymbol{P}_{\parallel}^*\nabla tr\left(\mathrm{Diag}(\boldsymbol{H}^*)^{1/2}\right) = \boldsymbol{0}.$$

W.L.O.G, we can decompose $\boldsymbol{P}_{\parallel}^*$ and $\boldsymbol{H}^*$ into block matrices as

$$\boldsymbol{P}_{\parallel}^* = \begin{pmatrix} \boldsymbol{0}, & \boldsymbol{0} \\ \boldsymbol{0}, & \boldsymbol{P}_{d-m}^* \end{pmatrix}, \boldsymbol{H}^* = \begin{pmatrix} \boldsymbol{0}, & \boldsymbol{0} \\ \boldsymbol{0}, & \boldsymbol{H}_{d-m}^* \end{pmatrix},$$

where $\boldsymbol{P}_{d-m}^*, \boldsymbol{H}_{d-m}^* \in \mathbb{R}^{(d-m)\times(d-m)}$ are full-rank matrices. Under this decomposition, the first $m$ diagonal elements in $(\mathrm{Diag}(\boldsymbol{H}))^{1/2}$ is 0, and the first $m$ diagonal elements in $\boldsymbol{S}^*$ is 0. Specifically,

$$\boldsymbol{S}^* = \begin{pmatrix} \boldsymbol{0} & \boldsymbol{0} \\ \boldsymbol{0} & \boldsymbol{S}_{d-m}^* \end{pmatrix}, \quad \mathrm{Diag}(\boldsymbol{H}^*) = \begin{pmatrix} \boldsymbol{0} & \boldsymbol{0} \\ \boldsymbol{0} & \mathrm{Diag}(\boldsymbol{H}_{d-m}^*) \end{pmatrix}.$$

Then the constraint in Equation (14) can be reduced into

$$-\nabla_\Gamma tr\left(\mathrm{Diag}(\boldsymbol{H}^*)^{1/2}\right) = \boldsymbol{0}, \tag{16}$$

proving the theorem. $\qquad\square$

In the practical use, we usually set $\epsilon$ to an extremely small constant; for example, PyTorch's official documentation sets the default value of $\epsilon$ to $10^{-8}$. Such a small constant will hardly alter the form of the implicit bias, but to make our analysis more rigorous, we also derived the case for Adam with $\epsilon > 0$.

**Lemma H.2** (Adam's Implicit Bias under Label Noise and $\epsilon > 0$). *With the label noise condition, every fixed point of Equation* (3) *for Adam with $\epsilon > 0$ satisfies that*

$$\nabla_\Gamma tr \left( \text{Diag}(\boldsymbol{H})^{1/2} - \frac{\epsilon}{\sqrt{\alpha}} \ln \left( \sqrt{\alpha} \text{Diag}(\boldsymbol{H})^{1/2} + \epsilon \right) \right) = \boldsymbol{0}.$$

*Proof.* Let $(\boldsymbol{\zeta}^*, \boldsymbol{v}^*)$ be a fixed point of the ODE (13) (equivalently, of Equation (3)). Then, as in the proof of Lemma H.1, it satisfies the stationarity constraints

$$S(\boldsymbol{v}^*)\partial\Phi_{S(\boldsymbol{v}^*)}(\boldsymbol{\zeta}^*)S(\boldsymbol{v}^*)\nabla^3\mathcal{L}(\boldsymbol{\zeta}^*)[S(\boldsymbol{v}^*)] = \boldsymbol{0}, \tag{17}$$

$$\boldsymbol{v}^* = V\big(\boldsymbol{\Sigma}(\boldsymbol{\zeta}^*)\big). \tag{18}$$

Introduce the shorthand

$$\boldsymbol{S}^* := S(\boldsymbol{v}^*), \qquad \boldsymbol{P}_\|^* := \partial\Phi_{S(\boldsymbol{v}^*)}\boldsymbol{S}^*, \qquad \boldsymbol{H}^* := \nabla^2\mathcal{L}(\boldsymbol{\zeta}^*).$$

Under the label noise condition we have $\boldsymbol{\Sigma}(\boldsymbol{\zeta}^*) = \alpha\boldsymbol{H}^*$, hence by (18)

$$\boldsymbol{v}^* = V\big(\boldsymbol{\Sigma}(\boldsymbol{\zeta}^*)\big) = \text{diag}\big(\boldsymbol{\Sigma}(\boldsymbol{\zeta}^*)\big) = \alpha\text{diag}(\boldsymbol{H}^*).$$

For Adam with $\epsilon > 0$, the diagonal preconditioner is

$$\boldsymbol{S}^* = \text{Diag}\left( \frac{1}{\sqrt{\boldsymbol{v}^*} + \epsilon} \right) = \text{Diag}\left( \frac{1}{\sqrt{\alpha\text{diag}(\boldsymbol{H}^*)} + \epsilon} \right).$$

We now compute $\nabla^3\mathcal{L}(\boldsymbol{\zeta}^*)[\boldsymbol{S}^*]$. Since $\boldsymbol{S}^*$ is diagonal, we only need to sum up the diagonal terms:

$$\nabla^3\mathcal{L}(\boldsymbol{\zeta}^*)[\boldsymbol{S}^*] = \sum_{j=1}^d \frac{1}{\sqrt{\alpha H_{jj}^*} + \epsilon} \nabla\big(H_{jj}^*\big). \tag{19}$$

Define the scalar function

$$\psi(x) := \sqrt{\alpha x} - \epsilon \ln \big(\sqrt{\alpha x} + \epsilon\big), \qquad x \geq 0.$$

A direct differentiation gives

$$\psi'(x) = \frac{\alpha}{2\big(\sqrt{\alpha x} + \epsilon\big)}.$$

Therefore,

$$\frac{1}{\sqrt{\alpha x} + \epsilon} = \frac{2}{\alpha}\psi'.$$

Plugging this identity into (19) and using the chain rule yields

$$\nabla^3\mathcal{L}(\boldsymbol{\zeta}^*)[\boldsymbol{S}^*] = \frac{2}{\alpha}\sum_{j=1}^d \psi'(H_{jj}^*)\nabla\big(H_{jj}^*\big) = \frac{2}{\alpha}\nabla\left[ \sum_{j=1}^d \psi(H_{jj}^*) \right] = \frac{2}{\alpha}\nabla tr\Big(\psi\big(\text{Diag}(\boldsymbol{H}^*)\big)\Big). \tag{20}$$

Noting that $\psi$ acts elementwise on the diagonal, we can also write

$$tr\Big(\psi\big(\text{Diag}(\boldsymbol{H}^*)\big)\Big) = tr\left( \sqrt{\alpha}\text{Diag}(\boldsymbol{H}^*)^{1/2} - \epsilon \ln \big(\sqrt{\alpha}\text{Diag}(\boldsymbol{H}^*)^{1/2} + \epsilon\big) \right).$$

Substitute (20) into (17):

$$\boldsymbol{S}^*\boldsymbol{P}_\|^* \frac{2}{\alpha}\nabla tr \left( \sqrt{\alpha}\text{Diag}(\boldsymbol{H}^*)^{1/2} - \epsilon \ln \big(\sqrt{\alpha}\text{Diag}(\boldsymbol{H}^*)^{1/2} + \epsilon\big) \right) = \boldsymbol{0}.$$

Since $\epsilon > 0$, the diagonal matrix $\boldsymbol{S}^*$ has strictly positive diagonal entries and is therefore invertible, and we can cancel it out. With arguments similar to those in Lemma H.1, only the bottom-right part of $\boldsymbol{H}^*$ (that corresponds to the dimensions within $\Gamma$) are nonzero, and we obtain the following result:

$$\nabla_\Gamma tr \left( \sqrt{\alpha}\text{Diag}(\boldsymbol{H}^*)^{1/2} - \epsilon \ln \big(\sqrt{\alpha}\text{Diag}(\boldsymbol{H}^*)^{1/2} + \epsilon\big) \right) = \boldsymbol{0},$$

a direct calculation gives

$$\nabla_\Gamma \text{tr} \left( \text{Diag}(\boldsymbol{H}^*)^{1/2} - \frac{\epsilon}{\sqrt{\alpha}} \ln \left( \frac{\sqrt{\alpha}}{\epsilon} \text{Diag}(\boldsymbol{H}^*)^{1/2} + \boldsymbol{I} \right) \right) = \boldsymbol{0},$$

which proves the claim. □

**Lemma H.3** (AdamE's Implicit Bias under Label Noise and $\epsilon = 0$). *With the label noise condition, every fixed point of Equation* (3) *for AdamE-$\lambda$ with $\lambda \in [0, 1)$ and $\epsilon = 0$ satisfies $\nabla_\Gamma tr \left( \text{Diag}(\boldsymbol{H})^{1-\lambda} \right) = 0$.*

*Proof of Lemma H.3.* The proof is similar to the proof of Lemma H.1, but when calculating $\boldsymbol{S}^*$, we have $\boldsymbol{S}^* = \text{Diag}(\frac{1}{(\alpha \text{diag}(\boldsymbol{H}^*))^\lambda})$ instead of $\boldsymbol{S}^* = \text{Diag}(\frac{1}{(\alpha \text{diag}(\boldsymbol{H}^*))^{1/2}})$, which gives

$$\nabla^3 \mathcal{L}(\boldsymbol{\zeta}^*)[\boldsymbol{S}^*] = \sum_{j=1}^d \frac{1}{(\alpha H_{jj}^*)^\lambda} \nabla(H_{jj}^*) = \frac{1}{(1-\lambda)\alpha^\lambda} \sum_{j=1}^d \nabla((H_{jj}^*)^{1-\lambda})$$

$$= \frac{1}{(1-\lambda)\alpha^\lambda} \nabla \text{tr} \left( \text{Diag}(\boldsymbol{H}^*)^{1-\lambda} \right).$$

This leads to the constraint $-\nabla_\Gamma \text{tr} \left( \text{Diag}(\boldsymbol{H}^*)^{1-\lambda} \right) = \boldsymbol{0}$ in the same way as Equation (16), which completes the proof. □

## H.2 Proof of Lemma B.3

*Proof.* We only prove the second argument in Lemma B.3 with any $e_0 \in (0, 1]$, since taking $e_0 = 0.5$ yields the first argument. First, we recall that the minimizer manifold $\Gamma$ is defined as

$$\Gamma := \left\{ \boldsymbol{\theta} | \langle \boldsymbol{z}_i, \boldsymbol{u}^{\odot 2} - \boldsymbol{v}^{\odot 2} \rangle = y_i, \forall i \in [n] \right\}.$$

So if any $\boldsymbol{\theta} = \binom{\boldsymbol{u}}{\boldsymbol{v}}$ belongs to $\Gamma$, and another $\tilde{\boldsymbol{\theta}} = \binom{\tilde{\boldsymbol{u}}}{\tilde{\boldsymbol{v}}}$ satisfies that $\tilde{u}_i^{\odot 2} - \tilde{v}_i^{\odot 2} = u_i^{\odot 2} - v_i^{\odot 2}$ for any $i \in [d]$, then $\tilde{\boldsymbol{\theta}}$ also belongs to $\Gamma$.

Next, we derive the explicit expression of the Hessian matrix when $\boldsymbol{\theta} \in \Gamma$:

$$\nabla^2 \mathcal{L}(\boldsymbol{\theta}) = \frac{2}{n} \sum_{i=1}^n 2 \binom{\boldsymbol{z}_i \odot \boldsymbol{u}}{-\boldsymbol{z}_i \odot \boldsymbol{v}} \binom{\boldsymbol{z}_i \odot \boldsymbol{u}}{-\boldsymbol{z}_i \odot \boldsymbol{v}}^\top + \left( \langle \boldsymbol{z}_i, \boldsymbol{u}^{\odot 2} - \boldsymbol{v}^{\odot 2} \rangle - y_i \right) \begin{pmatrix} \text{Diag}(\boldsymbol{z}) & 0 \\ 0 & -\text{Diag}(\boldsymbol{z}) \end{pmatrix}$$

$$= \frac{4}{n} \sum_{i=1}^n \binom{\boldsymbol{z}_i \odot \boldsymbol{u}}{-\boldsymbol{z}_i \odot \boldsymbol{v}} \binom{\boldsymbol{z}_i \odot \boldsymbol{u}}{-\boldsymbol{z}_i \odot \boldsymbol{v}}^\top.$$

Hence, we have that

$$\text{tr}(\text{Diag}(\boldsymbol{H})^{e_0}) \propto \sum_{i=1}^d (|u_i|^{2e_0} + |v_i|^{2e_0}),$$

and $\left\| \boldsymbol{u}^{\odot 2} - \boldsymbol{v}^{\odot 2} \right\|_{e_0}^{e_0} = \sum_{i=1}^d |u_i^2 - v_i^2|^{e_0}$. Let $e_0 \in (0, 1]$, and we recall that our goal is to prove that given the following condition

$$\boldsymbol{\theta} \in \arg \min_{\boldsymbol{\theta}' \in \Gamma} \text{tr}(\text{Diag}(\boldsymbol{H})^{e_0}) = \arg \min_{\boldsymbol{\theta}' \in \Gamma} \sum_{i=1}^d (|u_i|^{2e_0} + |v_i|^{2e_0}), \tag{21}$$

it holds that $\boldsymbol{\theta} \in \arg \min_{\boldsymbol{\theta}' \in \Gamma} \|\hat{\boldsymbol{w}}\|_{e_0}$.

First, we prove that $u_i = 0 \vee v_i = 0$ holds for any $i \in [d]$. Assume for the contrary that there exists some $i$ such that $u_i \neq 0$ and $v_i \neq 0$, then we construct another reference point $\tilde{\boldsymbol{\theta}} = \binom{\tilde{\boldsymbol{u}}}{\tilde{\boldsymbol{v}}}$ by letting $\tilde{\boldsymbol{u}}$ and $\boldsymbol{u}$ agree on all indices other than $i$, and that

$$\tilde{u}_i = \sqrt{u_i^2 - v_i^2}, \tilde{v}_i = 0, \quad \text{if } |u_i| \geq |v_i|, \tag{22}$$

$$\tilde{u}_i = 0, \tilde{v}_i = \sqrt{v_i^2 - u_i^2}, \quad \text{otherwise.} \tag{23}$$

With this construction, $\tilde{\boldsymbol{u}}^{\odot 2} - \tilde{\boldsymbol{v}}^{\odot 2} = \boldsymbol{u}^{\odot 2} - \boldsymbol{v}^{\odot 2}$, so $\tilde{\boldsymbol{\theta}} \in \Gamma$. One can observe that

$$|\tilde{u}_i|^{2e_0} + |\tilde{v}_i|^{2e_0} < |u_i|^{2e_0} + |v_i|^{2e_0},$$

which contradicts the condition in Equation (21).

Now we are ready to prove $\boldsymbol{\theta} \in \arg\min_{\boldsymbol{\theta}' \in \Gamma} \|\widehat{\boldsymbol{w}}\|_{e_0}$. Also, we prove this by contradiction. Now assume $\boldsymbol{\theta} \notin \arg\min_{\boldsymbol{\theta}' \in \Gamma} \|\boldsymbol{u}^{\odot 2} - \boldsymbol{v}^{\odot 2}\|_{e_0}$. There must exist some $\tilde{\boldsymbol{\theta}} \in \Gamma$ such that

$$\left\|\tilde{\boldsymbol{u}}^{\odot 2} - \tilde{\boldsymbol{v}}^{\odot 2}\right\|_{e_0} < \left\|\boldsymbol{u}^{\odot 2} - \boldsymbol{v}^{\odot 2}\right\|_{e_0}.$$

W.L.O.G., one can assume that for any $i \in [d]$, either $\tilde{u}_i = 0$ or $\tilde{v}_i = 0$, else we can construct another minimizer that preserves $\left\|\tilde{\boldsymbol{u}}^{\odot 2} - \tilde{\boldsymbol{v}}^{\odot 2}\right\|_{e_0}$ as Equation (22) and Equation (23). However, given the condition $u_i = 0 \vee v_i = 0$, we have that

$$\sum_{i=1}^d |u_i^2 - v_i^2|^{e_0} = \sum_{i=1}^d |u_i|^{2e_0} + |v_i|^{2e_0},$$

and $\sum_{i=1}^d |\tilde{u}_i^2 - \tilde{v}_i^2|^{e_0} = \sum_{i=1}^d |\tilde{u}_i|^{2e_0} + |\tilde{v}_i|^{2e_0}$, which indicates that $\sum_{i=1}^d |\tilde{u}_i|^{2e_0} + |\tilde{v}_i|^{2e_0} < \sum_{i=1}^d |u_i|^{2e_0} + |v_i|^{2e_0}$, a contradiction. $\quad\square$

# I   Regularizers under label noise for AGMs in Table 1

In this section, we provide additional discussions on the regularizer for the AGM optimizers in Table 1 besides Adam, AdamE (refer to Appendix H), and Shampoo (refer to Appendix J).

**SGD.**   Under label noise, the implicit bias of SGD has been extensively studied by previous works; As discussed in Section 2, approaches such as fixed point analysis [Blanc et al., 2020], slow SDE [Li et al., 2021b] and implicit gradient regularization [Barrett and Dherin, 2020] all agree on the result that SGD implicitly regularizes $\mathrm{tr}(\boldsymbol{H})$ on the minimizer manifold. Our work provides a new insight on the implicit bias of SGD by comparing with that of Adam. Specifically, SGD treats each direction equally which results in a rotation invariant $\mathrm{tr}(\boldsymbol{H})$ as the implicit regularizer, while Adam has the second-order momentum as a denominator, so Adam regularizes the entries with small gradients relatively faster, as is indicated in its implicit bias $\mathrm{tr}((\mathrm{Diag}\boldsymbol{H})^{1/2})$.

**RMSProp.**   The RMSProp optimizer Hinton et al. [2012] can be seen as a special case of Adam, where $\beta_1 = 0$. One can observe that $\beta_1$ does not appear in the slow SDE system, which implies that as long as $1 - \beta_1$ is of constant order, the choice of $\beta_1$ has nothing to do with the dynamics of Adam on the minimizer manifold. The intuition is that, after the iteration approaches the manifold, the gradient $\nabla\mathcal{L}(\boldsymbol{\theta}_k)$ moves very slowly as $k$ proceeds. Since the momentum only captures $\mathcal{O}(\log 1/\eta)$ past steps, the different between momentum and the gradient at that step becomes negligible. Therefore, RMSProp possesses an implicit bias identical to Adam: $\mathrm{tr}((\mathrm{Diag}\boldsymbol{H})^{1/2})$.

**Adam-mini and Adalayer.**   Adam-mini [Zhang et al., 2025] and Adalayer [Zhao et al., 2025] belong to the same kind of variant of Adam that partitions the parameters. In Adam-mini the partitions are blocks, and in Adalayer the partitions are layers. In the sequel, we provide a brief derivation of the implicit bias of "Partitioned Adam", which is applicable to any kind of optimizer whose functions $V$ and $S$ can be expressed in the form of

$$V(\boldsymbol{M})_i := \frac{1}{|B_{\pi(i)}|} \sum_{j \in B_{\pi(i)}} M_{jj}$$

$$S(\boldsymbol{v}) := \mathrm{Diag}\big(1/(\sqrt{\boldsymbol{v}} + \epsilon)\big)$$

where $\mathcal{B} = \{B_1, B_2, \cdots, B_N\}$ is a partition of $[d]$, and for each $i \in [d]$, $\pi(i)$ denotes the index of the set containing $i$, i.e. $i \in B_{\pi(i)}$. We derive the case for $\epsilon = 0$.

Recall from the proof of Lemma H.1 that the gradient of the implicit regularizer being minimized on manifold can be expressed as

$$\partial^2 (\nabla\mathcal{L})[\boldsymbol{S}] = \nabla[\langle \boldsymbol{S}, \boldsymbol{H}\rangle] - \nabla(\boldsymbol{S})[\boldsymbol{H}]. \tag{24}$$

In our case $S$ is diagonal, so we can calculate the contribution of each set in the partition, and add them up. Next we focus on a single set, and re-index it as $\{1, 2, \cdots, G\}$ without loss of generality. In this set $\mathrm{tr}\boldsymbol{H}/G$ is used as a shared second-order momentum, so we have

$$\boldsymbol{S} = \left[\frac{\mathrm{tr}\boldsymbol{H}}{G}\boldsymbol{I}_G\right]^{1/2}.$$

Combining with $\boldsymbol{P} = \boldsymbol{S}^{1/2}$ gives us

$$\langle \boldsymbol{S}, \boldsymbol{H}\rangle = \mathrm{tr}\left(\boldsymbol{PHP}\right) = \sqrt{G \cdot \mathrm{tr}\boldsymbol{H}}.$$

For the second term, we again denote $h_j := \boldsymbol{H}_{jj}$, and we further denote $t := \mathrm{tr}\boldsymbol{H}/G = \frac{1}{G}\sum_{j=1}^G h_j$.

$$\nabla\left(\boldsymbol{S}\right)[\boldsymbol{H}] = \sum_j \nabla\left(t^{-1/2}\right) \cdot h_j$$

$$= \sum_j \nabla\left(t\right) \cdot h_j \cdot -\frac{1}{2}t^{-3/2}$$

$$= G \cdot \nabla\left(t\right) \cdot -\frac{1}{2}t^{-1/2}$$

$$= -G\nabla\left(t^{1/2}\right) = -\nabla\sqrt{G \cdot \mathrm{tr}\boldsymbol{H}}.$$

Plugging into (24) gives the implicit bias contributed by this set as $\sqrt{G \cdot \mathrm{tr}\boldsymbol{H}}$. Finally, summing up all the sets, we conclude the overall implicit bias as

$$\sum_{i\in[N]} \sqrt{|B_i| \cdot \mathrm{tr}\boldsymbol{H}_{B_i}}.$$

Here, $\boldsymbol{H}_{B_i}$ means the submatrix of $\boldsymbol{H}$ if we restrict the rows and columns to $B_i$.

## J Shampoo Optimizer as an AGM

### J.1 A brief introduction to Shampoo

Shampoo, unlike most conventional stochastic first-order optimization methods, utilizes the fact that many model parameters in practice are tensor-like, and can lead to faster convergence in optimization. In this paper, we consider the case where the parameter is a matrix (2-dimensional tensor) with shape $d_1 \times d_2$. In this case, the Shampoo algorithm can be expressed as follows:

---
**Algorithm 1** Shampoo with matrix-like parameters

---
**Require:** horizon $K$, learning rate $\eta > 0$, stabilizing constant $\epsilon > 0$.
1: Initialize $\boldsymbol{\Theta}_0 \in \mathbb{R}^{d_1\times d_2}$; $\boldsymbol{S}_{1,0} \in \mathbb{R}^{d_1\times d_1}$; $\boldsymbol{S}_{2,0} \in \mathbb{R}^{d_2\times d_2}$
2: **for** $k = 1$ to $K$ **do**
3:     Receive loss function $\ell_k : \mathbb{R}^{d_1\times d_2} \to \mathbb{R}$
4:     $\boldsymbol{G}_k \leftarrow \nabla\ell_k(\boldsymbol{\Theta}_k) \in \mathbb{R}^{d_1\times d_2}$
5:     Update $\boldsymbol{S}_{1,k+1}$ with $\boldsymbol{S}_{1,k}$ and $\boldsymbol{G}_k\boldsymbol{G}_k^\top$
6:     Update $\boldsymbol{S}_{2,k+1}$ with $\boldsymbol{S}_{2,k}$ and $\boldsymbol{G}_k^\top\boldsymbol{G}_k$
7:     $\boldsymbol{\Theta}_{k+1} \leftarrow \boldsymbol{\Theta}_k - \eta\left(\boldsymbol{S}_{1,k+1} + \epsilon\boldsymbol{I}_{d_1}\right)^{-\lambda} \boldsymbol{G}_k \left(\boldsymbol{S}_{2,k+1} + \epsilon\boldsymbol{I}_{d_2}\right)^{-\lambda}$
8: **end for**

---

Here, $\lambda$ is a constant that equals $1/4$ when Shampoo was originally proposed by Gupta et al. [2018], while later works [Anil et al., 2020, Shi et al., 2023, Morwani et al., 2024] suggests switching to $\lambda = 1/2$. We incorporate the constant $\epsilon$ here, being consistent with Gupta et al. [2018] and the practical need to stabilize training. There are multiple choices for the update rule in Lines 5 and 6. Originally, Gupta et al. [2018] simply sum up the past terms to maintain $\boldsymbol{S}_1$ and $\boldsymbol{S}_2$. Later in practice [Morwani et al., 2024, Lin et al., 2025], this was replaced by an Exponential Moving Average (EMA):

$$\boldsymbol{S}_{1,k+1} = \beta_2\,\boldsymbol{S}_{1,k} + (1-\beta_2)\,\boldsymbol{G}_k\boldsymbol{G}_k^\top, \qquad \boldsymbol{S}_{2,k+1} = \beta_2\,\boldsymbol{S}_{2,k} + (1-\beta_2)\,\boldsymbol{G}_k^\top\boldsymbol{G}_k. \qquad (25)$$

We adopt the update rule (25) for Lines 5 and 6 in Algorithm 1, which aligns with practical implementations and naturally fits within our AGM framework; We use $\lambda = 1/2$ to stick with the latest result, but the following analysis actually holds for any constant $\lambda \geq 0$.

## J.2 Shampoo under the AGM Framework

In Shampoo the parameter takes a matrix form, while our AGM framework requires a vector, so we need to define some reshaping rules to view Shampoo seamlessly as an AGM.

First, we define the vectorization of matrices $\text{vec}(\cdot) : \mathbb{R}^{p \times q} \to \mathbb{R}^{pq}$ as

$$\boldsymbol{X} \mapsto \boldsymbol{x}, \quad \text{where} \quad x_{iq+j} = X_{i,j}, \ \forall \ 0 \leq i < p, 0 \leq j < q.$$

We introduce the *Kronecker product*, a matrix operation that generalizes the outer product to matrices. Given any $\boldsymbol{A} \in \mathbb{R}^{p \times q}$ and $\boldsymbol{B} \in \mathbb{R}^{r \times s}$, their Kronecker product $\boldsymbol{A} \otimes \boldsymbol{B} \in \mathbb{R}^{pr \times qs}$ is defined as the block matrix obtained by multiplying each entry of $\boldsymbol{A}$ by the entire matrix $\boldsymbol{B}$. It satisfies several useful properties:

$$\text{vec}(\boldsymbol{A}\boldsymbol{X}\boldsymbol{B}^\top) = (\boldsymbol{B} \otimes \boldsymbol{A})\text{vec}(\boldsymbol{X}), \quad \text{for any } \boldsymbol{X} \in \mathbb{R}^{q \times s}, \tag{26}$$

$$(\boldsymbol{A}^t \otimes \boldsymbol{B}^t) = (\boldsymbol{A} \otimes \boldsymbol{B})^t, \quad \text{for any } \boldsymbol{A} \in \mathbb{S}_{++}^p, \boldsymbol{B} \in \mathbb{S}_{++}^r, t \in R, \tag{27}$$

where $\mathbb{S}_{++}^p, \mathbb{S}_{++}^r$ denote the set of positive definite matrices in $\mathbb{R}^{p \times p}$ and $\mathbb{R}^{r \times r}$ respectively. For any step $k \in [0, K-1]$, we define $\boldsymbol{\theta}_k := \text{vec}(\boldsymbol{\Theta}_k)$ and $\boldsymbol{g}_k := \text{vec}(\boldsymbol{G}_k)$. Let $d = d_1 d_2$, then $\boldsymbol{\theta}_k, \boldsymbol{g}_k \in \mathbb{R}^d$. The Shampoo update (Line 7, Algorithm 1) can now be rewritten as

$$\boldsymbol{\theta}_{k+1} = \boldsymbol{\theta}_k - \eta \left( (\boldsymbol{S}_{2,k+1} + \epsilon \boldsymbol{I}_{d_2})^\top \otimes (\boldsymbol{S}_{1,k+1} + \epsilon \boldsymbol{I}_{d_1}) \right)^{-1/2} \boldsymbol{g}_k. \tag{28}$$

Next, for any $\boldsymbol{M} \in \mathbb{R}^{d \times d}$, we introduce a new indexing scheme that represents $\boldsymbol{M}$ as a 4-dimensional tensor by decomposing each row and column index into two sub-indices. Specifically, for all $0 \leq i, j < d_1$ and $0 \leq l, r < d_2$, we use

$$M_{i,l,r,j}$$

as an alternative representation of

$$M_{id_2+l, \ jd_2+r}.$$

Then we define two functions $V_L : \mathbb{R}^{d \times d} \to \mathbb{R}^{d_1 \times d_1}, V_R : \mathbb{R}^{d \times d} \to \mathbb{R}^{d_2 \times d_2}$ as:

$$[V_L(\boldsymbol{M})]_{i,j} = \sum_s M_{i,s,s,j},$$

$$[V_R(\boldsymbol{M})]_{i,j} = \sum_s M_{s,i,j,s}.$$

Note that for any step $k \in [1, K]$,

$$\left[V_L(\boldsymbol{g}_k \boldsymbol{g}_k^\top)\right]_{i,j} = \sum_s \left[\boldsymbol{g}_k \boldsymbol{g}_k^\top\right]_{i,s,s,j} = \sum_s [\boldsymbol{G}_k]_{i,s}[\boldsymbol{G}_k^\top]_{s,j} = \left[\boldsymbol{G}_k \boldsymbol{G}_k^\top\right]_{i,j}, \tag{29}$$

$$\left[V_R(\boldsymbol{g}_k \boldsymbol{g}_k^\top)\right]_{i,j} = \sum_s \left[\boldsymbol{g}_k \boldsymbol{g}_k^\top\right]_{s,i,j,s} = \sum_s [\boldsymbol{G}_k^\top]_{i,s}[\boldsymbol{G}_k]_{s,j} = \left[\boldsymbol{G}_k^\top \boldsymbol{G}_k\right]_{i,j}. \tag{30}$$

Now we are ready to rewrite the Shampoo optimizer in the AGM form.

**Definition J.1** (Shampoo, written in the AGM form).

$$\boldsymbol{V}_{k+1} := \beta_2 \boldsymbol{V}_k + (1 - \beta_2)V\left(\boldsymbol{g}_k \boldsymbol{g}_k^\top\right) \tag{31}$$

$$\boldsymbol{\theta}_{k+1} := \boldsymbol{\theta}_k - \eta S(\boldsymbol{V}_{k+1})\boldsymbol{g}_k. \tag{32}$$

Here, each $\boldsymbol{V}_k \in \mathbb{R}^{d_1 \times d_1} \times \mathbb{R}^{d_2 \times d_2}$ is a tuple of two matrices with shape $d_1 \times d_1$ and $d_2 \times d_2$ respectively; The functions $V$ and $S$ are defined as

$$V : \mathbb{R}^{d \times d} \to \mathbb{R}^{d_1 \times d_1} \times \mathbb{R}^{d_2 \times d_2}, \quad \boldsymbol{M} \mapsto (V_L(\boldsymbol{M}), V_R(\boldsymbol{M})),$$

$$S : \mathbb{R}^{d_1 \times d_1} \times \mathbb{R}^{d_2 \times d_2} \to \mathbb{R}^{d \times d}, \quad (V_1, V_2) \mapsto \left((V_2 + \epsilon \boldsymbol{I}_{d_2})^\top \otimes (V_1 + \epsilon \boldsymbol{I}_{d_1})\right)^{-1/2}.$$

With (29) and (30), it is straightforward to verify that (31) recovers (25) and that $\boldsymbol{V}_k = (\boldsymbol{S}_{1,k}, \boldsymbol{S}_{2,k})$; Hence (32) recovers (28) as well.

## J.3 Vectorized AGM form

It is worth noting that in Definition J.1, we slightly generalized the definition of $S$ and $V$ so that $\boldsymbol{V}_k$ is not a plain vector now, but a tuple of two matrices. This is only for a simplification of the expression of functions $S$ and $V$ which makes them easy to understand. In this subsection, we establish a form of Shampoo that is slightly more complicated, but rigorously fit in the shape required by AGM, and we will use this form in the next subsection to analyze why Shampoo's bias on the manifold cannot be written using an explicit regularizer.

For two specific shapes of matrices: $d_1 \times d_1$ and $d_2 \times d_2$, we define functions that inverse the vectorization effect:

$$\text{mat}_L : \mathbb{R}^{d_1^2} \to \mathbb{R}^{d_1 \times d_1}, \quad [\text{mat}_L(\boldsymbol{v})]_{i,j} = v_{id_1+j}, \forall 0 \le i, j < d_1,$$

$$\text{mat}_R : \mathbb{R}^{d_2^2} \to \mathbb{R}^{d_2 \times d_2}, \quad [\text{mat}_R(\boldsymbol{v})]_{i,j} = v_{id_2+j}, \forall 0 \le i, j < d_2.$$

For any vector $\boldsymbol{v} \in \mathbb{R}^{d_1^2 + d_2^2}$, we write $(\boldsymbol{v}_L, \boldsymbol{v}_R)$ and $\boldsymbol{v}$ interchangeably so that $\boldsymbol{v}_L$ represent the first $d_1^2$ entries and $\boldsymbol{v}_R$ represent the rest. Now we present a vectorized version of Definition J.1.

**Definition J.2** (Shampoo, written in the vectorized AGM form)**.**

$$\boldsymbol{v}_{k+1} := \beta_2 \boldsymbol{v}_k + (1 - \beta_2) V\left(\boldsymbol{g}_k \boldsymbol{g}_k^\top\right)$$

$$\boldsymbol{\theta}_{k+1} := \boldsymbol{\theta}_k - \eta S(\boldsymbol{v}_{k+1}) \boldsymbol{g}_k.$$

Here, each $\boldsymbol{v}_k \in \mathbb{R}^D$ where $D = d_1^2 + d_2^2$, and the functions $V$ and $S$ are defined as

$$V : \mathbb{R}^{d \times d} \to \mathbb{R}^D, \quad \boldsymbol{M} \mapsto \left(\text{vec}\left(V_L(\boldsymbol{M})\right), \text{vec}\left(V_R(\boldsymbol{M})\right)\right),$$

$$S : \mathbb{R}^D \to \mathbb{R}^{d \times d}, \quad \boldsymbol{v} \mapsto \left(\left(\text{mat}_R(\boldsymbol{v}_R) + \epsilon \boldsymbol{I}_{d_2}\right)^\top \otimes \left(\text{mat}_L(\boldsymbol{v}_L) + \epsilon \boldsymbol{I}_{d_1}\right)\right)^{-1/2}.$$

## J.4 Discussion on Shampoo's Implicit Bias under Label Noise

Recall that for all AGMs, under label noise, Equation (14) and Equation (15) hold, and we adopt the notations $\boldsymbol{S}^*, \boldsymbol{P}_\parallel^*, \boldsymbol{H}^*$ in Lemma H.1 to write them as

$$\boldsymbol{S}^* \partial \boldsymbol{P}_\parallel^* \nabla^3 \mathcal{L}(\boldsymbol{\zeta}^*)[\boldsymbol{S}^*] = 0, \quad \boldsymbol{v}^* = V(\boldsymbol{\Sigma}(\boldsymbol{\zeta}^*)).$$

Here we can expand $\boldsymbol{S}^*$ as

$$\boldsymbol{S}^* = S(\boldsymbol{v}^*)$$

$$= \left(\left(\text{mat}_R(\boldsymbol{v}_R^*) + \epsilon \boldsymbol{I}_{d_2}\right)^\top \otimes \left(\text{mat}_L(\boldsymbol{v}_L^*) + \epsilon \boldsymbol{I}_{d_1}\right)\right)^{-1/2}$$

$$= \left(\left(V_R(\boldsymbol{\Sigma}(\boldsymbol{\zeta}^*)) + \epsilon \boldsymbol{I}_{d_2}\right)^\top \otimes \left(V_L(\boldsymbol{\Sigma}(\boldsymbol{\zeta}^*)) + \epsilon \boldsymbol{I}_{d_1}\right)\right)^{-1/2},$$

and we further denote

$$\boldsymbol{\Sigma}_L^* := V_L(\boldsymbol{\Sigma}(\boldsymbol{\zeta}^*)) + \epsilon \boldsymbol{I}_{d_1}, \quad \boldsymbol{\Sigma}_R^* := V_R(\boldsymbol{\Sigma}(\boldsymbol{\zeta}^*)) + \epsilon \boldsymbol{I}_{d_2}.$$

We define a function $A : \mathbb{R}^d \to \mathbb{R}^d$ as

$$A(\boldsymbol{\zeta}) := \nabla^3 \mathcal{L}(\boldsymbol{\zeta}) \left[S(V(\boldsymbol{\Sigma}(\boldsymbol{\zeta})))\right].$$

Note that

$$A(\boldsymbol{\zeta}^*) = \nabla^3 \mathcal{L}(\boldsymbol{\zeta}^*) \left[\boldsymbol{S}^*\right]. \tag{33}$$

We provide theoretical evidence that Shampoo may not admit any explicit regularizer under label noise by showing that *the function $A$ is not a conservative vector field*, thus no potential function $\psi$ exists such that $\nabla \psi \equiv A$.

To see it clearer, if such a function $\psi$ exists, then with the same techniques as Lemma H.1 we can simplify (14) into

$$\nabla_\Gamma \psi(\boldsymbol{\zeta}^*) = 0,$$

so the regularizer is exactly $\psi(\boldsymbol{\zeta}^*)$, and vice versa.

Unfortunately, even in a simple case, where $\boldsymbol{H}^*$ is assumed to be diagonal (for example, the scenario of diagonal net), the potential function $\psi$ does not exist. In this case, we can assume that

$$\boldsymbol{H}^* = \mathrm{Diag}(\lambda_1, \lambda_2, \ldots, \lambda_d),$$

where $\lambda_i \geq 0$ for any $1 \leq i \leq d$. Consequently,

$$\boldsymbol{\Sigma}(\boldsymbol{\zeta}^*) = \alpha \boldsymbol{H}^* = \alpha \mathrm{Diag}(\lambda_1, \lambda_2, \ldots, \lambda_d).$$

Now we have

$$
\begin{aligned}
\boldsymbol{\Sigma}_L^* &= V_L(\boldsymbol{\Sigma}(\boldsymbol{\zeta}^*)) + \epsilon \boldsymbol{I}_{d_1} \\
&= \alpha \mathrm{Diag}\left( \sum_{i=0}^{d_2-1} \lambda_{1+id_1}, \sum_{i=0}^{d_2-1} \lambda_{2+id_1}, \ldots, \sum_{i=0}^{d_2-1} \lambda_{d_1+id_1} \right) + \epsilon \boldsymbol{I}_{d_1} \\
&= \mathrm{Diag}(r_1, r_2, \ldots, r_{d_1}) \in \mathbb{R}^{d_1 \times d_1}, \quad \text{where } r_j := \alpha \sum_{i=0}^{d_2-1} \lambda_{j+id_1} + \epsilon; \\
\boldsymbol{\Sigma}_R^* &= V_R(\boldsymbol{\Sigma}(\boldsymbol{\zeta}^*)) + \epsilon \boldsymbol{I}_{d_2} \\
&= \alpha \mathrm{Diag}\left( \sum_{i=0}^{d_1-1} \lambda_{1+id_2}, \sum_{i=0}^{d_1-1} \lambda_{2+id_2}, \ldots, \sum_{i=0}^{d_1-1} \lambda_{d_2+id_2} \right) + \epsilon \boldsymbol{I}_{d_2} \\
&= \mathrm{Diag}(l_1, l_2, \ldots, l_{d_2}) \in \mathbb{R}^{d_2 \times d_2}, \quad \text{where } l_j := \alpha \sum_{i=0}^{d_1-1} \lambda_{j+id_2} + \epsilon.
\end{aligned}
$$

Therefore, the preconditioner matrix $\boldsymbol{S}^*$ in Equation (33) can be written as

$$
\begin{aligned}
\boldsymbol{S}^* &= (\boldsymbol{\Sigma}_R^* \otimes \boldsymbol{\Sigma}_L^*)^{-1/2} \\
&= \mathrm{Diag}\left( l_1^{-1/2} \cdot (\boldsymbol{\Sigma}_R^*)^{-1/2}, l_2^{-1/2} \cdot (\boldsymbol{\Sigma}_R^*)^{-1/2}, \ldots, l_{d_2}^{-1/2} \cdot (\boldsymbol{\Sigma}_R^*)^{-1/2} \right).
\end{aligned}
$$

One can straightforwardly verify that the curl of $A$ at $\boldsymbol{\zeta}^*$:

$$\nabla \times A(\boldsymbol{\zeta}^*) = \nabla \times \nabla^3 \mathcal{L}(\boldsymbol{\zeta}^*)\,[\boldsymbol{S}^*]$$

is nonzero. By the Stokes-Cartan theorem (Theorem 16.11 in Lee [2012]), there does not exist a potential function $\psi$ such that $\nabla \psi \equiv A$. Therefore, we argue that in general, the regularization effect of Shampoo under label noise cannot be reduced to an explicit regularizer, for which the diagonal case is a counterexample.

