# OpenReview forum: "Adam Reduces a Unique Form of Sharpness: Theoretical Insights Near the Minimizer Manifold"
_NeurIPS.cc/2025/Conference — NeurIPS 2025 poster_

### Official Review · Reviewer_9RWz · 2025-06-26

**Clarity:** 3
**Significance:** 3
**Originality:** 4
**Rating:** 5
**Confidence:** 4

**Summary:**

To analyze the optimization trajectories of Adam after reaching the minimizer manifold, the authors propose the *slow sde* framework, which applies to a general class of adaptive optimizers, including Adam. Authors prove that such a continuous SDE solution can approximate the iterates of Adam within an error lower than $O(\eta^{1/4})$. In addition, they also provide a convergence guarantee for Adam to reach the minimizer manifold. Based on such an SDE approximation, the authors analyze the fixed points of the SDE and provide the implicit bias of Adam and AdamE-$\lambda$, respectively.

**Questions:**

Could the authors explain more about ''slow sde" for both the SGD and Adam? For me, the conventional SDE is clear. However, the "slow sde" is relatively hard to follow. It's difficult to understand why the trajectories of SGD can be characterized in this form. What do the diffusion term and the drift term indicate? Why is the current updating direction of SGD (from my understanding, this should depend on its local information) related to its convergence point (as reflected in the definition of $\Phi$)?

Besides, could you please explain how you deal with the second term in line 306 of the proof sketch? Since the definition of the calculation $\nabla (\cdot) [H]$ is defined for a Hessian matrix, it's a little hard for me to follow this calculation.

In addition, although you have mentioned that the first momentum would not affect the approximation result, I wonder whether you can also add it as a part of your "slow sde" system? This is not a major concern, just for better illustration.

**Ethical Concerns:**

["NO or VERY MINOR ethics concerns only"]

**Final Justification:**

Most of my issues have been resolved. I believe this presents good work!

**Limitations:**

No negative societal impact. Potential technical weaknesses have been discussed.

**Paper Formatting Concerns:**

No.

**Quality:**

4

**Strengths And Weaknesses:**

**Strengths:**
1. The analysis regarding the continuous approximation of optimizers plays a key role in helping to understand their behavior. Therefore, such a result is of great significance, especially given that there are few related works for Adam due to its complicated updating rules compared to SGD.
2. The theoretical findings and illustration of this paper are relatively complete and solid. Almost all the theoretical conclusions are established with convincing evidence and clear illustrations. For example, the authors provide both the convergence guarantee to reach the minimizer manifold, then the SDE approximation inner the neighborhood of such a minimizer manifold. I would, however, suggest that authors consider relocating subsection 4.3.1, the convergence of Adam, to the beginning of Section 4.

**Weakness:**
Some of the following points might be more suitable to state in the question part, but I still state them here since I'm not very familiar with the SDE and differential geometry. Due to the same reason, I have set my confidence level of this rating to 2. Authors could correct me if I have any misunderstandings regarding their results.

1. The requirement that $1-\beta_2$ is comparable with $\eta^2$, coupled with the $O(\eta^{1/4})$  approximation error established in Theorem 4.1, implies that the "slow ODE" presented in this paper might not adequately capture Adam's trajectories. For example, taking $\beta_2=0.9999$, a value already larger than common settings, still yields a convergence guarantee on the scale of $1/3$.  Consequently, this approximation might not provide a strong guarantee. Is such a requirement raised as you use the $\eta^2$ as the interpolation scale to establish the ode form of $v_t$? Similarly, I wonder whether there are some typos in the formula of Theorem 4.1 of the term $\eta^2$. I also wonder why there exists a factor $\sqrt\eta$ before the diffusion term of the conventional SDE in Line 129. From my perspective, the noise of the stochastic gradient might be uncorrelated with the learning rate.

2.  While the conclusion of the implicit bias regarding the trace of Hessian is convincing to me, some explanations of such an result would be much better. For example, what are the connections or comparisons between this result and the direct conclusion that Adam can maximize the $\ell_\infty$-margin in a recent work [1]. At least to me, the conclusion in [1] is easier to understand its implication. I know you have provided an example in subsection 5.2. However, its extremely specific settings ($z_i \in \\{1, -1\\}$) make me feel that the conclusion related to the implicit regularization of different forms of norm lacks sufficient generalization. In addition, I suggest that the authors could design some experiments to directly support their main theorems on the trace of the Hessian matrix.

[1] Zhang et al. The implicit bias of Adam on separable data. NeurIPS.

---

> ### Author Rebuttal · Authors · 2025-07-31
>
> # Response to Reviewer 9RWz
>
> Dear Reviewer `9RWz`,
>
> We sincerely appreciate your acknowledgment that our continuous approximation result “is of great significance,” and that our theoretical findings “are relatively complete and solid.” Your comments are invaluable, and we hope this response addresses your concerns.
>
> ---
>
> **W1 part1: The requirement that $1-\beta_2$ is comparable with $\eta^2$ makes the $O(\eta^{1/4})$ approximation error not very tight.**
>
> **A:** We thank the reviewer for raising this point about the approximation tightness.
> - Our main contribution is to correctly derive Adam's slow SDE for the first time, which already involves highly non-trivial technical challenges (Section 4.3). This SDE approximation is useful, as it enables the analysis of Adam's implicit regularizer (Section 5.1) and helps anticipate Adam's advantage or disadvantage relative to SGD on problems such as sparse linear regression and matrix factorization. While further tightening the bound would be valuable, we view this as an important direction for future work.
> - We also note that our current error bound is already the best known: even the state-of-the-art analysis of SGD [1] achieves the same order $O(\eta^{1/4})$. Other works on SGD can only establish an asymptotic convergence of SGD to its slow SDE in distribution, without explicit error bounds [2,3].
>
> **W1 part2: Whether there are some typos in the formula of Theorem 4.1 of the term $\eta^2$**
>
> **A.**  Thank you for your feedback. **We confirm that is not a typo actually.** This theorem shows that, in the small  $\eta$  regime, once Adam approaches the minimizer manifold, its long-horizon behavior within $\tilde{O}(\frac{1}{\eta^2})$ steps can be well approximated by the SDE defined in Equation (2). More specifically, in an SDE, a continuous time duration $T$ corresponds to $\frac{T}{\eta^2}$ steps of Adam in discrete time. A similar approximation theorem for the slow SDE of SGD can be found in [1].
>
> **W1 part3: I wonder why there exists a factor $\sqrt{\eta}$ before the diffusion term of the conventional SDE in Line 129. **
>
> **A:** This is a good question. The factor $\sqrt{\eta}$ in front of the diffusion term is standard in SDE analyses of SGD. See [4,1,2,5,6]; they all add this $\sqrt{\eta}$ factor in their conventional SDEs.
>
> In the SGD update, the gradient term is scaled by $\eta$. Thus, the corresponding diffusion term in the SDE, which represents the stochastic noise, scales by $\sqrt{\eta}$. This scaling arises from Euler's first-order discretization of the stochastic process. Putting it mathematically, for an SDE
>
> $$dX_t = a(X_t)dt + b(X_t)dW_t,$$
>
> its Euler discretization with a time step $\Delta t=\eta$ is
>
> $$X_{t+\eta} = X_{t} + \eta a(X_{t}) + b(X_{t}) (W_{t + \eta}-W_{t}),$$
>
> where $(W_{t + \eta}-W_{t}) \sim \mathcal{N}(0,\eta\mathbf{I})$. We want the diffusion term and drift term above to coincide with those of a single-step SGD update. For an SGD update, we write out its update rule at step $k$ as
>
> $$\theta_{k+1} = \theta_k - \eta \nabla \mathcal{L}(\theta_k) + \eta z_k,$$
>
> where $z_k := \ell_{k}(\theta_k) - \mathcal{L}(\theta_k)$ and we define the covariance matrix $\Sigma(\theta_k):=\text{Cov}(V_k,V_k\mid\theta_k)$. We have that $\text{Cov}(\eta V_k,\eta V_k\mid\theta_k)=\eta^2\Sigma(\theta_k)$. We identify $t$ with $k\eta$, therefore, if we set
>
> $$a(\theta_k) =  \nabla \mathcal{L}(\theta_k), \quad b(\theta_k) = (\eta\Sigma(\theta_k))^{1/2},$$
>
> then the diffusion and drift terms of SDE will match those of the SGD update. Hence, this motivates the approximating equation
> $$dX_t = \nabla \mathcal{L}(X_t)dt + \sqrt{\eta}\Sigma(X_t)^{1/2}dW_t.$$
>
> Thus, the factor$\sqrt{\eta}$ is necessary to ensure that the diffusion term is correctly scaled and reflects the noise variance, maintaining consistency with the overall SGD dynamics. We hope this clears up your concerns.
>
> **W2 part1: Can you explain the connection to 'The implicit bias of Adam on separable data'?**
>
> **A:** Thank you for this good question. We have cited and discussed this work in our paper (2nd paragraph of Section 2). The paper you cited is, in fact, **orthogonal to our research**. Their work studies overparameterized linear logistic regression, where the minimizers are at infinity and Adam converges only in direction. There is no minimizer manifold in their setting. By contrast, our analysis focuses on settings with finite minimizers (e.g., logistic regression with L2 regularization). Therefore, their results do not apply to our setting, nor do ours apply to theirs. Our setup is arguably more realistic since real-world models (1) are non-linear and (2) have finite minimizers.
>
> **W2 part2: I suggest that the authors could design some experiments to directly support their main theorems on the trace of the Hessian matrix.**
>
> **A:** We would like to point out that our paper already provides experiments on matrix factorization to validate the implicit bias of Adam (Appendix B). In particular, we plot the curves of $\text{tr}(\mathbf{H})$ and $\text{tr}(\text{Diag}(\mathbf{H})^{1/2})$ for Adam, and demonstrate that Adam is minimizing $\text{tr}(\text{Diag}(\mathbf{H})^{1/2})$ rather than $\text{tr}(\mathbf{H})$ (Figure 4, top-right). This causes a negative effect on generalization compared with SGD (Figure 4, bottom-right).
>
> ---
>
> **Q1: Explain more about ''Slow SDE'' for both SGD and Adam.**
>
> Thanks for your feedback. We will break down your question into 3 sub-questions and answer them one by one.
>
> **Q1.1: Could the authors explain more about ''slow sde" for both the SGD and Adam? For me, the conventional SDE is clear. However, the "slow sde" is relatively hard to follow.**
>
> **A:** In the following, we demonstrate the difference of slow SDE with conventional SDE from two aspects.
>
> - **The slow SDE traces the trajectory of SGD in a longer horizon compared with the conventional SDE**. As depicted in Appendix A, conventional SDEs can only track iteration for $O(\eta^{-1})$ time, while slow SDEs can approximate the process after the parameter gets close to the minimizer manifold (up to $O(\eta^{-2})$). This longer time horizon makes it much easier to see the long-term regularization effects of SGD directly from the SDE.
> - **Also, slow SDE traces the trajectory of the projection of the model parameter onto the manifold, rather than the parameter itself.** This allows us to better understand how the implicit regularization effect gradually influences the global minimizer that SGD centers around over time.
>
> **Q1.2: It's difficult to understand why the trajectories of SGD can be characterized in this form. What do the diffusion term and the drift term indicate?**
>
> **A:** We provide an interpretation of slow SDE beginning at line 168. Next, we first give a comprehensive explanation of the diffusion term, drift term, and slow SDE.
>
> - The **diffusion term** captures the stochastic noise from mini-batch gradients and other gradient noise (e.g., label noise). It injects random perturbations along directions into the tangent space of the manifold, driving the parameter to wander around the manifold instead of settling at a single point.
> - Meanwhile, **drift term** serves as a deterministic force that pushes the solution toward flatter, more stable regions on the manifold. When the label noise condition is satisfied, the indication of the drift term become clearer, where it can be interpreted as reducing a regularization term $(\text{tr}(\nabla^2 \mathcal{L})$ for SGD, and $\text{tr}(\text{diag}(\nabla^2 \mathcal{L})^{1/2})$ for Adam) inside the manifold.
>
> The mechanism behind slow SDE is quite complicated. In simple terms, in normal space, SGD behaves like an Ornstein-Uhlenbeck (OU) process locally, whose mixing time is $\Theta(\eta^{-1})$. After writing out Taylor's expansion of this process up to the third order, further analysis can show that the third term in this Taylor expansion creates a $\Theta(\eta^2)$ velocity in the tangent space, and there is no push-back force in the tangent space. Thus, the small velocity accumulates over time, making SGD continuously move on the manifold to a certain direction. And slow SDE characterizes this slow dynamics.
>
> **Q1.3: Why is the current updating direction of SGD (from my understanding, this should depend on its local information) related to its convergence point?**
>
> **A:** **Actually the convergence point defined by $\Phi$ is local information.** This is because the iteration stays very close to the manifold during the timescope of slow SDE (refer to Figure 3\(c\) in Appendix A for an illustration). It is important to note that the slow SDE serves as an approximation of the SGD only when the parameter is near the minimizer manifold.
>
> **Q2: Explain how you deal with the second term in line 306 of the proof sketch?**
>
> **A:** Here we just open this term using the definition of  $\partial F(\theta)[\mathbf{M}]$ in Section 3. Specifically, we have
>
> $$\nabla \left(\mathbf{S}\right)\left[\nabla^2\mathcal{L}\right] = \sum_{j,k}\nabla\left(\mathbf{S}_{jk}\right)\cdot\mathbf{H}_{jk},$$
>
> and then plug in the expression of $\mathbf{S} = \text{Diag}(H)^{-1/2} + O(\epsilon)$ into this equation gives
>
> $$\nabla \left(\mathbf{S}\right)\left[\nabla^2\mathcal{L}\right] = \sum_{j}\mathbf{H}_{jj}^{-1/2}\mathbf{H}_{jj} + O(\epsilon \text{tr}(\mathbf{H})),$$
>
> and take $\epsilon \to 0$ gives the final results. The more detailed calculation can be found in Appendix G
>
> **Q3: Can we include first momentum in the slow SDE?**
>
> **A:** It depends on how close $\beta_1$ is to 1.
> - As our paper assumed, If $b_1 = 1 - \beta_1$ is a constant (independent of $\eta$), the error stays $O(\eta^{1/4})$ and we cannot add $\beta_1$ into slow SDE. (Assumption C.6,line 646)
> - If $b_1 = o(1)$ (e.g. $O(\eta)$), then $\beta_1$ would change the slow SDE form, but this case is less practical, since $\beta_1$ is typically not greater than $0.9$ in real-world areas. So we left it for future work.

---

> ### Author Response · Authors · 2025-08-01
> **Response to Reviewer 9RWz (Detailed Answer for Q3 & clean copy of Q2 & missing reference)**
>
> Dear reviewer `9RWz`,
>
> Thank you very much for your detailed questions. As there were many points raised in the review, we prepared a fairly lengthy response. However, due to the word limit of the rebuttal, we couldn’t elaborate fully in the main reply, so we’ve included additional clarifications in the comments.
>
> ---
>
> First, this capacity limitation prevents us from providing the titles of the articles cited:
> - **W1 part 1** We also note that our current error bound is already the best known: even the state-of-the-art analysis of SGD [1] achieves the same order
> . Other works on SGD can only establish an asymptotic convergence of SGD to its slow SDE in distribution, without explicit error bounds [2,3]
> - **W1 part2** More specifically, in an SDE, a continuous time duration $T$ corresponds to $\frac{T}{\eta^2}$ steps of Adam in discrete time. A similar approximation theorem for the slow SDE of SGD can be found in [1].
> - **W1 part 3** The factor $\sqrt{\eta}$ in front of the diffusion term is standard in SDE analyses of SGD. See [4,1,2,5,6] ; they all add this $\sqrt{\eta}$ factor in their conventional SDEs.
>
> ---
>
> Additionally, due to markdown rendering errors that made the formula in Q2 hard to read, we’ve included a clean copy of an Extended version of the response to Q2 here for better readability.
>
> **Q2: Explain how you deal with the second term in line 306 of the proof sketch?**
>
> **A:** Thank you for your careful reading. Here we first open this term using the definition of  $\partial F(\theta)[\mathbf{M}]$ in Section 3. Specifically, we have:
>
> $$\nabla (\mathbf{S})[\nabla^2\mathcal{L}]  =\sum_{j,k}\nabla(\mathbf{S}\_{jk})H\_{jk},$$
>
> where $\mathbf{H}:= \nabla^2\mathcal{L}$  is the Hessian matrix, and then plug in the expression of $\mathbf{S} = \text{Diag}(\mathbf{H})^{-1/2} + O(\epsilon)$ into this equation gives
>
> $$\nabla \left(\mathbf{S}\right)\left[\nabla^2\mathcal{L}\right] = \sum_{j}\nabla(H\_{jj}^{-1/2})H\_{jj} + O(\epsilon \text{tr}(\mathbf{H})),$$
>
> where
> $$\sum_{j}\nabla (H\_{jj}^{-1/2}) H\_{jj} = \sum_{j} \nabla(H\_{jj}) \cdot (-\frac{1}{2}H\_{jj}^{-1/2}) = - \nabla \text{tr}\left(\left(\text{Diag}\left(\mathbf{H}\right)\right)^{1/2}\right),$$
>
> and take $\epsilon \to 0$ gives the final results. The more detailed calculation can be found in Appendix G (at the bottom of page 42) of the full version submitted in the "Supplementary Material" zip package.
>
> ---
>
> Next, we resubmit our detailed response to Q3.
>
> **Q3: Although you have mentioned that the first momentum would not affect the approximation result, I wonder whether you can also add it as a part of your "slow sde" system?**
>
> **A:** Thanks for your question. The answer depends on how close $\beta_1$ is to $1$.
> - First, we would like to emphasize that **since our paper considers the regime where $b_1 := 1 - \beta_1$ is of constant order (Assumption C.6 & Remark line 646, Appendix C), it is impossible to incorporate $\beta_1$ directly into the slow SDE system.** In this regime, as long as $b_1$ is constant with $\eta$, the approximation rate is always $O(\eta^{1/4})$.
> - **If one assumes that $b_1 = o(1)$ (e.g., $O(\eta)$), $\beta_1$ could affect the form of slow SDE**, which is left for future work. However, **this regime is less realistic than the one we focus on.** Assuming $b_1$ is constant aligns with common practice better, since $\beta_1$ is typically not greater than $0.9$ in real-world areas such as NLP [7,8] and CV [9,10,11].
>
> The reason why  $\beta_1$ does not appear as a part of the SDE system under our conditions is that, after the iteration approaches the manifold, the difference between the current gradient  $g_t$  and momentum $M_t$ becomes negligible in expectation. For more details, see Line 931, Appendix F.3. A recent paper [3] on SGDM (SGD with Momentum) with constant $\beta_1$ also establishes a form of slow SDE for SGDM that does not depend on $\beta_1$.
>
> ---
>
> [1] Why (and When) does Local SGD Generalize Better than SGD?
>
> [2] What Happens after SGD Reaches Zero Loss? --A Mathematical Framework.
>
> [3] The Marginal Value of Momentum for Small Learning Rate SGD.
>
> [4] Stochastic Modified Equations and Dynamics of Stochastic Gradient Algorithms I: Mathematical Foundations.
>
> [5] On the Validity of Modeling SGD with Stochastic Differential Equations (SDEs)
>
> [6] On the SDEs and Scaling Rules for Adaptive Gradient Algorithms
>
> [7] Bert: Pre-training of deep bidirectional transformers for language understanding
>
> [8] Improving language understanding by generative pre-training
>
> [9] Image-to-image translation with conditional adversarial networks
>
> [10] An image is worth 16x16 words: Transformers for image recognition at scale
>
> [11] U-net: Convolutional networks for biomedical image segmentation

---

> ### Author Response · Authors · 2025-08-06
> **Look forward to a feedback from the reviewer**
>
> **Dear Reviewer `9RWz`,**
>
> We sincerely thank you for your timely response and valuable feedback. As the rebuttal deadline approaches, we would appreciate your further feedback to ensure that our responses have fully addressed your concerns.
>
> To provide a comprehensive response to your detailed questions, we first addressed each of the weaknesses and questions you raised point by point in our rebuttal. We then supplemented this with an official comment to offer more thorough clarifications, particularly for your Questions 2 and 3, and to include the full titles of cited articles for enhanced clarity.
>
> We believe these two documents, taken together, now more directly address your remaining concerns. We greatly appreciate your professionalism and thoughtful comments and look forward to your feedback.
>
> Sincerely,
>
> The Authors

---

> > ### Comment · Reviewer_9RWz · 2025-08-06
> >
> > Regarding the approximation rate discussion, I should clarify that I did not intend to demand a resolution to this issue. I fully understand that theoretical concessions are sometimes necessary due to inherent challenges, and I appreciate the authors' efforts in achieving guarantees comparable to SGD. However, I would like to note that the dependencies between $\eta$ and $\beta_2$ limit the generalizability of these results to broader settings—a distinction from SGD. While this does not diminish the significance of the study, it indeed presents a weakness.
> >
> > For the Slow SDE clarification, I appreciate the authors' explanation. If I understand correctly, the Slow SDE captures Adam’s trajectories over a phase $T$, and due to the interpolation of $\eta^2$, it effectively represents the entire $T/\eta^2$ steps. Is this interpretation accurate? Additionally, I acknowledge the reply regarding the $\sqrt{\eta}$ factor. I also don't know why I did not figure out such an evident fact at that time :).
> >
> > Concerning [1], I may have been unclear in my initial comment. My intent was not to request a technical comparison but rather to seek intuitive insights similar to those provided in [1] - I don't mean that their conclusions are stronger. The example in Subsection 5.2 is insightful, though its highly specific construction somewhat limits its broader applicability. I wonder if I could obtain similar examples, but more general.
> >
> > Admittedly, I still lack a deep mathematical grasp of the Slow-SDE mechanism, but I accept that a thorough understanding may require delving into the technical details. The current response suffices for my review.
> >
> > Regarding the first-moment discussion, if it indeed influences your differential equation system, I recommend incorporating the analysis (or at least a brief discussion) in the camera-ready version for completeness.
> >
> > Lastly, due to my poor background in differential geometry (and I’m unsure if it’s essential for this work), I initially rated my confidence as 2. **However, after seeing other reviews, I believe my review is among the most detailed and relevant. Given that some critiques dismiss the work on grounds that could apply to the entire field, I feel a responsibility to evaluate its significance more precisely. Accordingly, I’ve raised my rating to 5 and my confidence to 4 to reflect my positive assessment of this contribution.**

---

> > > ### Author Response · Authors · 2025-08-07
> > > **A Note of Thanks to the Reviewer**
> > >
> > > We sincerely thank you for your exceptionally detailed and insightful review, and are particularly grateful for your strong support of our work. We also want to confirm that your interpretation of the Slow SDE: "the Slow SDE captures Adam’s trajectories over a phase $T$, and due to the interpolation of $\eta^2$, it effectively represents the entire $T/\eta^2$ steps." is indeed accurate.
> > >
> > > Thanks again for your valuable feedback and support!

---

### Official Review · Reviewer_cpiw · 2025-07-02

**Clarity:** 3
**Significance:** 3
**Originality:** 3
**Rating:** 5
**Confidence:** 3

**Summary:**

This paper studies the Adam optimizer and shows that it reduces a specific form of sharpness that leads to qualitatively different solutions from SGD. The paper shows that when using Adam to train over-parameterized models with label noise, Adam minimizes $tr(Diag(H)^{1/2})$ a measure of "sharpness" which differs from the $tr(H)$ that is minimized with SGD. Finally, they show that for the specific case of solving sparse linear regression with diagonal linear networks Adam achieves better sparsity and generalization than SGD.

**Questions:**

None.

**Ethical Concerns:**

["NO or VERY MINOR ethics concerns only"]

**Final Justification:**

My main concern was that there was no appendix and it was unclear whether their results hold for ReLU based networks. Both of these have been resolved in the rebuttal process.

**Limitations:**

Yes.

**Paper Formatting Concerns:**

The entire Supplementary Material including the Appendix is missing from the paper.

**Quality:**

3

**Strengths And Weaknesses:**

## Strengths:
- The paper is written clearly and the results bring new insights into how the implicit bias of Adam differs from SGD.
- The paper provides concrete examples that illustrate the qualitative difference between the kinds of solutions Adam learns vs. SGD (such as Fig. 1 and the sparse linear regression example).
- The paper introduces the AdamE-$\lambda$ optimizer to directly tune the implicit bias of Adam.
- The theoretical contributions apply broadly to many AGM and is not specific to Adam.

## Weaknesses:
- The entire Appendix is missing which makes it impossible to verify the correctness of the proofs.
- Adam is typically used to train ReLU neural networks which are non-smooth and non-convex. It is not clear whether the analysis here translates to this more realistic setting.

---

> ### Author Rebuttal · Authors · 2025-07-31
>
> ### Response to Reviewer cpiw
>
> Dear reviewer `cpiw`,
>
> We thank you for your feedback and for acknowledging that our paper “is clearly written and brings new insights” into Adam's implicit bias. We believe the reviewer's main concerns stem from not being able to correctly locate the appendix of our paper. We address the two key weaknesses below and are confident that these clarifications, along with the supplementary material, resolve the primary reasons for the borderline score.
>
> **Q1: The entire Appendix is missing?**
>
> **A:** We followed the NeurIPS traditions to put our appendix to the supplementary material. This can be downloaded through the 'zip' icon inside the *Supplementary Material* section in the OpenReview submission page, and the full paper `adam_sde_nips.pdf` can be found inside. We included the full proof, a formal restatement of theorems, an illustration of the difference between Slow SDE and conventional SDE, an additional matrix factorization experiment, and the code for the diagonal net experiment in the supplementary material.
>
> **Q2: Adam is typically used to train ReLU neural networks which are non-smooth and non-convex, how do your analysis transfer to realistic setting.**
>
> **A:** Actually, **our analysis is aligned with many realistic settings**:
> - First, **frontier LLMs have largely switched from ReLU activations to smooth activation functions**, such as SwiGLU, Swish, softplus, and GeLU, to improve training stability. Specifically, many mainstream LLMs such as Llama, Qwen, and DeepSeek series all use SwiGLU as the activation function. Therefore, it is reasonable to assume smoothness in our analysis.
> - Also, we would like to clarify that **our paper does not require either global or local convexity of the loss function**. Our main results (see full statements of our main results in Appendix C) focus on the dynamics near the minimizer manifold, where the local landscape may not be convex (but it can be proved to satisfy $\mu$-PL property, a much weaker condition).

---

> ### Author Response · Authors · 2025-08-06
> **Look forward to a feedback from the reviewer**
>
> **Dear Reviewer `cpiw`,**
>
> We sincerely thank you for your feedback and acknowledgment of our paper. As the rebuttal deadline approaches, we would greatly appreciate any additional input to help ensure that our responses fully address your concerns.
>
> We believe some of your main concerns may have arisen from difficulty locating the appendix, where the full proof of our main results is provided. We would like to kindly ask whether reviewing the appendix has helped clarify these points.
>
> Your insights are invaluable in helping us improve the clarity of our contributions and overall presentation. Please don’t hesitate to let us know if you have any further questions or suggestions.
>
> Sincerely,
>
> The Authors

---

> > ### Comment · Reviewer_cpiw · 2025-08-07
> > **Response to Authors**
> >
> > Thank you, I now see that the Appendix with all the proofs is there. My issues have been resolved and I will raise my score appropriately.

---

### Official Review · Reviewer_GRmZ · 2025-07-02

**Clarity:** 2
**Significance:** 2
**Originality:** 3
**Rating:** 4
**Confidence:** 1

**Summary:**

This paper investigates the behavior of the Adam optimizer under the assumption that the set of minimizers forms a manifold. By analyzing the corresponding slow stochastic differential equation (SDE), the authors demonstrate that Adam  promotes sparsity in the solution.

**Questions:**

See the weaknesses above.

Moreover, Diag(H)^{1/2} is Diag(H^{1/2}) or (Diag(H))^{1/2}?

**Ethical Concerns:**

["NO or VERY MINOR ethics concerns only"]

**Final Justification:**

The issues are solved. I have raised the score.

**Quality:**

3

**Strengths And Weaknesses:**

I'm not familiar with the field. Here are some strengths and weaknesses:

Strengths:

This paper generalizes the slow SDE to more general adaptive optimizers, like Adam. The result also tells us the difference between SGD and Adam minimizers they find.

Weaknesses:

1.The assumption implies that the trajectory x(t) of the optimizer converges to a single point, which can be quite strong. In practice, especially in non-convex settings or under noisy dynamics (like SDEs), only subsequence convergence may be guaranteed.

2. Why Assume the Minimizer Lies on a Manifold? This is often based on empirical observation as said in the paper. However, from a theoretical standpoint, assuming the set of minimizers is a manifold requires strong structural properties.


minor comments:
Line 159: better to use the terminology: in the normal space.

line 202: preconditioner flow: lacks blanket.

Definition 4.3: Typo Correct: given → Given.

---

> ### Author Rebuttal · Authors · 2025-07-31
>
> ### Response to Reviewer GRmZ
>
> Dear reviewer `GRmZ`,
>
> We appreciate your thoughtful and constructive feedback. We have carefully considered all comments and provide our point-by-point responses below. We believe these responses will address your concerns and improve your understanding of the manuscript.
>
> **Q1: The assumption implies that the trajectory x(t) of the optimizer converges to a single point, which can be quite strong. In practice, especially in non-convex settings or under noisy dynamics (like SDEs), only subsequence convergence may be guaranteed.**
>
> **A:** **The iteration does not necessarily converge to some fixed point under our assumptions.**
> * Our assumptions regarding the loss function include: the smoothness (Assumption 3.1) and compactness (Assumption 4.2) of the manifold that the optimizer converges to; and the smoothness of the loss function (Assumption C.1, C.2). Note that we don't require any assumption such as convexity that will result in a single global optimum.
> * We could not find the notation “x(t)” used in the review besides that in our definition of gradient flow projection (Line 154, Line 202). Here, x(t) is defined as an ODE, which does not have "noisy dynamics". We define the gradient flow projection only when the limit point exists, so this notion is well-defined.
> * If “x(t)” actually means the SDE trajectory “$X(t)$”, then it is true that $X(t)$ may converge to a single point in some cases. However, even this is NOT implied by our assumptions. Some other assumptions are needed to ensure this convergence. Even if we add more assumptions to make $X(t)$ converge to a single point, Adam's trajectory does NOT converge to a single point. This is because $X(t)$ only describes the continuous approximation of Adam projected onto the minimizer manifold. The original trajectory of Adam can be seen as $X(t)$ plus some perturbation of scale $O(\eta^{1/4})$ (Theorem 4.1), which may not decay to zero when $t \to \infty$.
>
> **Q2: Why Assume the Minimizer Lies on a Manifold? This is often based on empirical observation, as said in the paper. However, from a theoretical standpoint, assuming the set of minimizers is a manifold requires strong structural properties.**
>
> **A:** The manifold assumption is still reasonable from the theoretical perspective for the following reasons:
> - **Under an over-parametrized setting, the minimizer manifold provably exists** with mild assumptions. Specifically, with parameter dimension $d$ and dataset size $n \ll d$, define the set of global minimizers as $\Gamma$. Two commonly used assumptions are: 1. all global minimizers interpolate the dataset (i.e. $f_i(\theta) = y_i, \forall i \in [n]$ for all $\theta \in \Gamma$), and 2. the Jabobian matrix $[\nabla f_1 (\theta),\nabla f_2 (\theta),\cdots,\nabla f_n(\theta)]$ is of full rank $n$ for all $\theta \in \Gamma$. With these two assumptions, **the preimage theorem** implies that $\Gamma$ is a $(d-n)$-dimensional manifold. A similar discussion can be found in Section 3.1 of [4].
> - The tendency for minimizers to form connected sets, namely the **"mode connectivity"** phenomenon, has been widely observed in practice [1,3] and has been demonstrated theoretically in some notable settings [2].
> - **The manifold assumption is a standard practice** used by many theoretical papers in the literature [4,5,6,7], which is a close approximation of mode connectivity for the tractability in theoretical analysis.
>
> **Q3: Does $Diag(H)^{1/2}$ mean $Diag(H^{1/2})$ or $(Diag(H))^{1/2}$?**
>
> A: $Diag(H)^{1/2}$ means $(Diag(H))^{1/2}$. That is, first we keep only the diagonal entries in $H$, and then take the square root.
>
> ---
> #### Response to Minor Comments
>
> Also, we thank you for your careful reading and helpful suggestions. We will incorporate all of these corrections into the revised manuscript. Specifically:
> - Line 159: We will adopt the suggested terminology "in the normal space."
> - Line 202: We will add the missing space to correct "preconditionerflow" to "preconditioner flow."
> - Definition 4.3: We will correct the typo ("given" → "Given").
>
> [1] Loss surfaces, mode connectivity, and fast ensembling of dnns.
>
> [2] Understanding warmup-stable-decay learning rates: A river valley loss landscape perspective.
>
> [3] Explaining landscape connectivity of low-cost solutions for multilayer nets.
>
> [4] What Happens after SGD Reaches Zero Loss? --A Mathematical Framework.
>
> [5] Why (and when) does local sgd generalize better than sgd?
>
> [6] Convergence rates for the stochastic gradient descent method for non-convex objective functions.
>
> [7] Understanding gradient descent on the edge of stability in deep learning.

---

> ### Author Response · Authors · 2025-08-03
> **Following up on our rebuttal**
>
> Dear Reviewer GRmZ,
>
> Thank you for acknowledging our rebuttal. We just wanted to briefly follow up on our two main clarifications:
> - We were not entirely certain of your definition of x(t). As detailed in our rebuttal, we analyzed several possible interpretations and showed that none of them imply convergence to a single point.
> - Assuming the minimizers form a manifold is a standard practice in this research area, which is also theoretically grounded.
>
> We hope these points have addressed your main concerns. Please let us know if you have any further questions.

---

> > ### Comment · Reviewer_GRmZ · 2025-08-05
> >
> > Thank you for the rebuttal, the issues are solved. I have raised the score.

---

### Official Review · Reviewer_8TUV · 2025-07-03

**Clarity:** 3
**Significance:** 4
**Originality:** 4
**Rating:** 5
**Confidence:** 2

**Summary:**

The paper theoretically analyzes the implicit bias of Adam. One of the main results shows that in the setting with label noise, Adam and SGD minimize different things: Adam minimizes $\mathrm{tr}(\mathrm{Diag}(H)^{1/2})$ and (as has been previously known) SGD minimizes $\mathrm{tr}(H)$. Furthermore, the authors propose a tunable version of Adam which minimizes $\mathrm{tr}(\mathrm{Diag}(H)^{\lambda})$ for any choice of $\lambda \in (0,1)$. This allows more control over the desired implicit bias of the algorithm.

**Questions:**

* In the experiments, all version of AdamE either behave more or less like SGD or like Adam. Can you narrow down at which $\lambda$ the change happens? Or, if we tried many more intermediate values of $\lambda$, we would see a gradual shift requiring more and more training data as $\lambda$ decreases? If it is sudden, do you have an explanation for this phase transition?
* I assume "Aappendix" in line 302 should say Appendix A?

**Ethical Concerns:**

["NO or VERY MINOR ethics concerns only"]

**Final Justification:**

There seems to be broad agreement among reviewers on the value of the work, the authors engaged with my questions and I am happy to keep my positive score.

**Limitations:**

yes

**Quality:**

4

**Strengths And Weaknesses:**

Unfortunately, I could not check all the math, but the paper appears very interesting to me.

Strengths:
* Adam is considered far less in theoretical studies than GD is despite the fact that Adam is, used more in many practical settings. This paper adds to the understanding of Adam.
* The tools presented in the paper capture other adaptive gradient methods and should provide useful for further works in the future. It is almost more like a framework than a single result.
* The authors make an effort to try to illustrate what the difference in behaviour between SGD and Adam mean intuitively.
* Proving Adam's bias towards sparsity in the presence of label noise is very interesting.

Weakness:
* There are a few assumptions (but I found them mostly mild).
* The analysis only works for specific parameter settings: $\beta_2$ and the learning rate have to follow a specific relation.

---

> ### Author Rebuttal · Authors · 2025-07-31
>
> ### Response to Reviewer 8TUV
>
> Dear reviewer `8TUV`,
>
> We sincerely thank you for your interest in the paper and your appreciation of the slow SDE of Adam. We are pleased to hear that the paper's results, particularly the analysis of Adam's bias towards sparsity in the presence of label noise, are considered “very interesting.” Moreover, we are grateful for the acknowledgment that the tools and framework presented could be valuable for future work in adaptive gradient methods. Below, we address the specific concerns and questions raised in the review.
>
> **Q1: In the experiments, all versions of AdamE either behave more or less like SGD or like Adam. Can you narrow down at which $\lambda$ the change happens? Or, if we tried many more intermediate values of $\lambda$, we would see a gradual shift requiring more and more training data as $\lambda$ decreases? If it is sudden, do you have an explanation for this phase transition?**
>
> **A:** In our experiments, the **transition is sudden, and happens very close to zero, between $\lambda \in (0.01, 0.001)$**. To provide an insight into this, note that **the change of minimizer may not be smooth when we gradually change the norm being regularized**.
>
> Here we provide a toy case for understanding the sudden phase transition caused by the regularization parameter  $\lambda$. Recall that AdamE-$\lambda$  minimizes $L_p$-norm where $p = 1 - \lambda$, so the phase transition happens around $p = 1$. We consider a setting where we have two parameters  $x, y$  and the manifold being  $x+y=1$. A point with minimal  $L_1$  norm on this manifold could be any point on the segment between  $(0, 1)$ and  $(1, 0)$, but for any  $p < 1$ , the  $L_p$  minimizers on the manifold suddenly shrink to only two points, $(0, 1)$ and $(1, 0)$.
>
> **Q2: I assume "Aappendix" in line 302 should say Appendix A?**
>
> **A:** Thank you for pointing it out! After some reordering, this part of the proof actually exists in Appendix G in the supplementary material, and we will fix this in the camera-ready version.
>
> **W1: There are a few assumptions (but I found them mostly mild).**
>
> **A:** We do make a few assumptions, such as the smoothness of the loss function and the existence of a minimizer manifold, which we agree are mild, as acknowledged by the reviewer. We also look forward to further refining and weakening these assumptions in future work.
>
> **W2: The analysis only works for specific parameter settings: $\beta_2$ and the learning rate have to follow a specific relation.**
>
> **A:** Thank you for your careful reading and thoughtful feedback.
>
> - In this work, as the first step towards uncovering the implicit bias of Adam under a general setting, we analyze Adam under the case where $1-\beta_2 = O(\eta^2)$, namely the  $2$-scheme.
> - For any $p$-scheme with $p>2$, the second-order moment will not vary significantly throughout the whole implicit bias phase, so Adam will simply become a pre-conditioned version of SGD, and the conclusion can be simply transferred from that of SGD.
> - Although we did not cover the case where $p<2$, the SDE derived by us is already useful, since it allows for the first time the analysis of Adam's implicit regularizer (Section 5.1) and the anticipation of Adam's advantage/disadvantage over SGD on sparse linear regression/matrix factorization. Subsequent works can build upon ours and derive a tighter bound, which is beyond the scope of this paper.

---

> > ### Comment · Reviewer_8TUV · 2025-08-04
> >
> > Many thanks for the response. I will keep my score.

---

### Decision · Program_Chairs · 2025-09-17

**Decision:**

Accept (poster)

**Comment:**

This paper investigates the implicit bias of Adam and demonstrates that the optimization trajectories of Adam, after reaching the minimizer manifold, can be approximated by a continuous SDE. The reviewer agrees that understanding the implicit bias of Adam is an important problem, and the paper presents solid results. A minor comment is that the requirement relating the parameters $\beta$ and $\eta$ may limit the generalizability of these findings to broader settings. The authors are encouraged to incorporate the reviewer’s constructive feedback and discuss potential limitations in the revised version.